# Analysis Algorithm for Sky Type and Ice Halo Recognition in All-Sky Images

Sylke Boyd, Stephen Sorenson, Shelby Richard, Michelle King, Morton Greenslit

Division for Science and Mathematics, University of Minnesota-Morris, 500 E 4th Street, Morris, MN

*Correspondence to*: Sylke Boyd (sboyd@morris.umn.edu)

**Abstract.** Halo displays, in particular the 22° halo, have been captured in long-time series of images obtained from Total Sky Imagers (TSI) at various Atmospheric Radiation Measurement (ARM) sites. Halo displays form if smooth-faced hexagonal ice crystals are present in the optical path. We describe an image analysis algorithm for long-time series of TSI images which scores images with respect to the presence of 22° halos. Each image is assigned an ice halo score (IHS) for 22° halos, as well

as a photographic sky type (PST), which differentiates cirrostratus (PST-CS), partially cloudy (PST-PCL), cloudy (PST-CLD), or clear (PST-CLR) within a near-solar image analysis area. The colour-resolved radial brightness behaviour of the near-solar region is used to define the discriminant properties used to classify photographic sky type and assign an ice halo score. The scoring is based on the tools of multivariate Gaussian analysis applied to a standardized sun-centred image produced from the raw TSI image, following a series of calibrations, rotation, and coordinate transformation. The algorithm is trained based on a

training sets for each class of images. We present test results on halo observations and photographic sky type for the first four months of the year 2018, for TSI images obtained at the Southern Great Plains (SGP) ARM site. A detailed comparison of visual and algorithm scores for the month of March 2018 shows that the algorithm is about 90% reliable in discriminating the four photographic sky types, and identifies 86% of all visual halos correctly. Numerous instances of halo appearances were identified for the period January through April 2018, with persistence times between 5 and 220 minutes. Varying by month,

we found that between 9% and 22% of cirrostratus skies exhibited a full or partial 22° halo.

## Introduction

Modelling and predicting the Earth's climate is a challenge for physical science, even more so in light of the already observable changes in Earth's climate system (Fasullo and Balmaseda, 2014; Fasullo et al., 2016; IPCC, 2013, 2014). Global circulation

models (GCMs) describe the atmosphere in terms of a radiative dynamic equilibrium. The Earth receives solar shortwave (SW) radiation and discards energy back into space in form of terrestrial long-wave (LW) radiation. The radiation balance of the earth has been subject to much study and discussion (Fasullo and Balmaseda, 2014; Fasullo and Kiehl, 2009; Kandel and Viollier, 2010; Trenberth et al., 2015). Global Circulation Models (GCMs) describe the influence of various parts of the earth system in terms of radiative forcing factors (Kandel and Viollier, 2010; Kollias et al., 2007). Clouds may restrict the SW flux

reaching the surface, but they also influence the LW emissions back into space. While low stratus and cumulus clouds exhibit a net negative radiative forcing, high cirroform clouds are more varied in their radiative response, varying between negative and positive forcing depending on time of day, season, and geographical location (Campbell et al., 2016). The Fifth Assessment Report from the IPCC in 2013 (IPCC, 2013) identified ice and mixed clouds as major contributors to the low confidence level into the aerosol/cloud radiative forcing. The uncertainty in the aerosol/cloud forcing has implications for the confidence in and for the variance of the predictions of global circulation models (Fu et al., 2002; Trenberth et al., 2015). Closing the radiation budget of the Earth hinges on reliable cloud data (Hammer et al., 2017; Schwartz et al., 2014; van Diedenhoven et al., 2015; Waliser et al., 2009). Traditionally, cloud radiative forcing is modelled using a cloud fraction based on sky images (Kennedy et al., 2016; Kollias et al., 2007; Schwartz et al., 2014). Cirrostratus clouds, lacking sharp outlines, pose a challenge to this approach (Schwartz et al., 2014). The uncertainty about the role of cirrus in the global energy balance has been attributed to limited observational data concerning their temporal and spatial distribution, as well as their microphysics (Waliser et al., 2009). Cirroform clouds, at altitudes between 5000-12,000 m, are effective LW absorbers. Cloud particle sizes can range from a few microns to even centimetre sizes (Cziczo and Froyd, 2014; Heymsfield et al., 2013). Methods to probe cirrus cloud particles directly involve aircraft sampling (Heymsfield et al., 2013) and mountainside observations (Hammer et al., 2015). Ground- and satellite-based indirect radar and LIDAR measurements (Hammer et al., 2015; Hong et al., 2016; Tian et al., 2010) give reliable data on altitudes, optical depths, and particle phase. Even combined, these methods leave gaps in the data for spatial and temporal composition of ice clouds. The analysis of halo displays as captured by long-term total sky imagers may provide further insight and allow to close some of the gaps.

Optical scattering behaviour is influenced by the types of ice particles, which may be present in very many forms, including crystalline hexagonal habits in form of plates, pencils and prisms, hollow columns, bullets and bullet rosettes, and amorphous ice pellets, fragments, rimed crystals and others (Bailey and Hallett, 2009; Baran, 2009; Yang et al., 2015). Only ice particles with a simple crystal habit and smooth surfaces can lead to halo displays (Um and McFarquhar, 2015; van Diedenhoven, 2014). Usually, this will be the hexagonal prism habit, which we can find in plates, columns, bullet rosettes, pencil crystals, etc. If no preferred orientation exists, a clear tell-tale sign for their presence is the 22° halo around a light source in the sky, usually sun or moon. More symmetry in the particle orientations will add additional halo display features such as parhelia, upper tangent arc, circumscribed halo, and others (Greenler, 1980; Tape and Moilanen, 2006). As shown in theoretical studies (van Diedenhoven, 2014; Yang et al., 2015), halos form in particular if the ice crystals exhibit smooth surfaces. In that case, the forward scattered intensity is much more pronounced as in cases of rough surfaces, even if a crystal habit is present. If many of the ice particles are amorphous in nature, or did not form under conditions of crystal growth- for example by freezing from super-cooled droplets, or by riming – the forward scattering pattern will be weaker, and similar to what we see for liquid droplets: a white scattering disk surrounding the sun, but no halo. In turn, roughness and asymmetry of ice crystals influence the magnitude of backscattered solar radiation, thus influencing the radiative effect of cirrus clouds (van Diedenhoven, 2016). If the particles in the cirroform cloud are very small, e.g. a few microns (Sassen, 1991), diffraction will lead to a corona. We believe that a systematic observation of the optical scattering properties adds information to our data on cirrus microphysics

and cirrus radiative properties. The authors observed the sky at the University of Minnesota Morris, using an all sky camera, through a five-month period in 2015, and found an abundance of halo features.

There are a few studies pursuing a similar line of inquiry (Forster et al., 2017; Sassen et al., 2003). The study by Sassen et al (Sassen et al., 2003) showed a prevalence of the 22° halo, full in 6% and partial in 37.3% of cirrus periods, based on a ten-year

photographic and LIDAR record of mid-latitude cirrus clouds, also providing data on parhelia, upper tangent arcs, and other halo display features, as well as coronas. The photographic record was taken in Utah, and based on 20-minute observation intervals; cirrus identification was supported by LIDAR. The authors found an interesting variability in halo displays, related to geographical air mass origin, and suggest that optical displays may serve as tracers of the cloud microphysics involved. Forster *et al.* (Forster et al., 2017) used a sun-tracking camera system to observe halo display details over the course of  several

months in Munich, Germany, and a multi-week campaign in the Netherlands in November 2014. A carefully calibrated camera system provided high-resolution images, for which a halo detection algorithm was presented, based on a decision tree and random forest classifiers. Ceilometer data and cloud temperature measurements from radiosonde measurements were used to identify cirrus clouds. The authors report 25% of all cirrus clouds also produced halo displays, in particular in the sky segments located above the sun. The fraction of smooth crystals necessary for halo display appearance is at a minimum 10% for columns,

and 40% for plates, based on an analysis of scattering phase functions for single scattering events (van Diedenhoven, 2014). While this establishes a lower boundary, it is correct to say that the observability of a halo display allows to conclude that smooth crystalline ice particles are present and single-scattering events dominate. The consideration of the percentage of cirrus clouds that display optical halo features allows therefore, upon further study, inferences about the microphysical properties of the cloud. This raises interest in examining existing long-term records of sky images.

Long-term records of sky images have been accumulated in multiple global sites. The Office of Science in the US Department of Energy has maintained Atmospheric Radiation Measurement (ARM) sites. These sites, among other instruments, contain a Total Sky Imager (TSI), and have produced multi-year records of sky images. In this paper, we introduce a computational method to analyse these long-term records for the presence of halo displays in the images. We are introducing an algorithm to analyse long sequences of TSI images. The algorithm produces a time record of near-solar photographic sky type (PST),

differentiated as cirrostratus (PST-CS), partly cloudy (PST-PCL), cloudy (PST-CLD), and clear (PST-CLR) sky types, as well as assign an ice halo score (IHS). The resolution and distortion of the TSI images restricts the halo search to the common 22° halo. Other halo features, such as parhelia, can occasionally be seen in a TSI image, but often are too weak or too small to reliably discriminate them from clouds and or 22° halos. If present they would be classified by this algorithm as part of a 22°halo. Coronas are obscured by the shadow strip, and often also by over-exposure in the near-solar area of the image. The

algorithm offers an efficient method of finding 22° halo incidences, full or partial. Since ARM sites also have collected records of LIDAR and radiometric data, the TSI halo algorithm is intended to be compared to other instrumental records from the same locations and times. This will be addressed in future work.

Section 1 describes the TSI data used in this work. Section 2 presents the details of the image analysis algorithm, including subsections on algorithm goals, image preparation, and sky type and halo scoring. Section 3 applies the algorithm to the TSI data record of the first four months of 2018, and examines effectiveness and types of data available for this interval. Summary and outlook are given in Sect. 4.

## 1  TSI images

Images used in this paper were obtained from Atmospheric Research Measurement (ARM) Climate Research Facilities in three different locations: Eastern North Atlantic (ENA) Graciosa Island, Azores, Portugal; North Slope Alaska (NSA) Central Facility, Barrow AK; and Southern Great Plains (SGP) Central Facility, Lamont, OK (ENA, 2018; OLI, 2018; SGP, 2018). The ranges and dates vary by location, as listed in Table 1. The images were taken with Total Sky Imagers, which consist of a camera directed downward toward a convex mirror to view the whole sky from zenith to horizon. A sun-tracking shadow band is used to block the sun, which covers a strip of sky from zenith to horizon. Images were recorded every 30 seconds. The longest series was taken at the Southern Great Plains (SGP) location, reaching back to July 2000. The images, in JPEG format, have been taken continuously during day time. Aside from night time and polar night, there are some additional gaps in the data, perhaps due to instrument failure or other causes. Camera quality, exposure, mirror reflectance, image resolution, and image orientation varies over time as well as by location. For example, an image from SGP taken in 2018 has a size of 488 by 640 pixels. The short dimension limits the radius of the view circle to at most 240 pixels. A pixel close to the center of the view circle corresponds to an angular sky section 2.8º wide and 0.24º tall. At SGP, the solar position never reaches this point. Close to the horizon, one pixels averages a sky section that is 0.24º wide and 1.24º tall. Best resolution is achieved at zenith angle 45º, in which case every pixel represents a sky region of 0.33º by 0.33º. The perspective distortion is largest for sky segments close to the horizon due to perspective distortions of the sky. We used a sampling of eighty images taken from across the TSI record and across all available years to initiate the training set (ENA, 2018; NSA, 2018; SGP, 2018). This included images visually classified from the images as photographic sky types CS, PCL, CLD, CLR, and halo-bearing. Descriptions of the PST are provided in Table 2. The 80 sample images were used to develop the algorithm and define a suitable set of characteristic properties for STS and IHS. This set will be referred to as seed images since they also initialize the master table described below.

## 2  Algorithm

### 2.1  Goal and Strategy

The algorithm aims to process very large numbers of images, and return information about the presence of 22° halos, as well as the general sky conditions. The program is written in C++ and uses the opencv library for image processing. If given a list of image directories, the algorithm proceeds to sequentially import, process, and score TSI images compared to training sets gleaned from representative images for each scored class. We define four classes of photographic sky types (PSTs), listed in

Table 2, and a halo class. The factors that determine these choices are discussed in Sect. 2.3.1 and 2.3.3. The algorithm assigns a numeric photographic sky type score (PSTS) and a numeric ice halo score (IHS). For all image classes, sets of discriminant image properties have been defined which differ between ten distinct properties for PST classes, and 31 distinct properties for the halo class.

5    Multivariate analysis is one of the standard methods in image analysis, applied in a wide variety of problems. Numerous text books provide introductions to this method in theoretical background (Harris, 1975; Gnanadesikan, 1977), as well as in an application-oriented manner (Alpaydin and Bach, 2014; Flury, 1988). A set of $N_P$ discriminant properties of the image is chosen, selected to be characteristic for a particular sky type or the presence of a halo. Let this set of properties be the observation vector

$$X = \{x_i\}_i^{N_p}$$

(1)

For each class, a training set is created. The training set is a set of $N_t$ observation vectors for images that have been visually assigned to the class. A training set defines an ellipsoidal centroid in the property space of $X$, centred at the mean observation vector

$$M = \{\mu_i\}_{i=1}^{N_P}$$

(2)

$$\mu_i = \frac{1}{N_t} \sum_{k=1}^{N_t} x_{ik}$$

(3)

The centroid's extend is described by the $N_P \times N_P$ covariance matrix

$$\Sigma = \overline{(X - M)(X - M)^T} = \begin{pmatrix} \sigma_{11} & \sigma_{12} & ... \\ \sigma_{21} & \sigma_{22} & ... \\ ... & ... & ... \end{pmatrix}$$

(4)

with elements

$$\sigma_{ij} = \frac{1}{N_t} \sum_{k=1}^{N_t} x_{ik} x_{jk} - \mu_i \mu_j$$

(5)

25    The observation vector of any further image $X'$ will then be referenced with $M$ and $\Sigma$ in form of a multivariate normal distribution

$$F = C_0 exp\left(-\frac{1}{2}(X' - M)^T \Sigma^{-1}(X' - M)\right)$$

(6)

in which the quadratic form in the exponent is known as the square of the Mahalanobis distance in property space. The closer
30    an image places to the centroid of a class, the higher its score Eq. (6) will be. The Mahalonobis distance is expressed in units

of standard deviations, eliminating the influence of the units of the discriminant properties and the need for weights. It is interesting to note that the average Mahalonobis distance for a class is equal to the number of discriminant properties. The pre-factor $C_0$ in Eq. (6) is different for the photographic sky type scores (PSTS) and the ice halo score (IHS) since the dimensionality of the observation vectors for these two class types is different. It is chosen to place the values for $F$ into a convenient number range. The value $F$ for each class of images is akin to a continuous numerical probability that the image is located close to the centroid of this particular class.

The algorithm is outlined in Figure 1, together with the respective references to this text. Both, $M$ and $\Sigma^{-1}$, are computed a priori from the training sets via Eqs. (2) and (4). In order to score a time series of property vectors $X$, one only needs to import $M$ and $\Sigma^{-1}$ for each class once at the start of the analysis run. The training sets for each class of images are started using the set of 80 images described in Sect. 1, and are expanded as needed. This allows to continually train the algorithm toward improvement of scoring. This basic algorithm structure is used on a standardised local sky map, described in 2.2. The details of PSTS and IHS will be described separately below. The code and accessories can be accessed at a GitHub repository (Boyd et al., 2018).

## 2.2 Image preparations and local sky map (LSM)

The goal of the image preparation is to create a local sky map centred at the sun, in easy-to-use coordinates, after a minimal colour calibration, and after extraneous image parts have been masked. The image preparations include the following steps: (1) a colour correction, (2) an alignment calibration, (3) a removal of the perspective distortion, (4) masking and marking of the solar position, and (5) rotation and crop to create a Local Sky Map (LSM). Some sample steps in the image preparation are illustrated in Figure 2. The figure includes the original image, the image after preparation step (4), and the LSM after preparation step (5). The two sample images in Fig. 2 were taken at the Southern Great Plains ARM site in March and April of 2018 (SGP, 2018). One of the images contains a solar 22° halo, the other one is a partly cloudy sky without any halo indications.

Step (1) is a colour correction. Both original images in Fig. 2 have a slightly green tinge, which is typical for images from the TSI at this location, in particular after an instrument update in 2010. This is noticeable in particular if images are compared to earlier TSI data from the same location, and can become a problem for the planned analysis, especially for the use of relative colour values. Since the algorithm is intended for multiple TSI locations and records taken over long time, including device changes, it is necessary to consider the fact that no two camera devices have exactly the same colour response, even if of same type (Ilie and Welch, 2005). The colour calibration used in this algorithm is based on sampling of clear-sky colour channels to define weighed scaling factors for a whole series of images. Every pixel in a TSI image exhibits a value between 0 and 255 for each of the three colour channels blue (B), green (G), and red (R). The colour values represent the intensity of the colour channel registered for the particular pixel, varying between 0 (no intensity) and 255 (brightest possible). In a discoloured series,

measurements of BGR were taken in clear-sky images (indexed PST-CLR), and a scaling factor and weight for each colour channel defined based on this information:

$$
\left.\begin{aligned}
\beta_B &= 1.00 \\
\beta_G &= \frac{G_{ref}}{G_{CLR}} \times \frac{B_{CLR}}{B_{ref}} \\
\beta_R &= \frac{R_{ref}}{R_{CLR}} \times \frac{B_{CLR}}{B_{ref}}
\end{aligned}\right\} \text{ with } \left(B_{ref}, G_{ref}, R_{ref}\right) = (180, 120, 85)
$$

(7)

The reference values are based on colour values for clear sky images from the TSI records listed in Table 1. Near-zenith, clear blue sky provides a reproducible colour reference in all the locations. Once these colour-scaling factors are determined for a series, every image was then tinted by generating an average colour $(\bar{B}, \bar{G}, \bar{R})$ for a small near-zenith sky-sample and applying

$$
\begin{aligned}
B' &= \left[B + \alpha\left(\beta_B\bar{B} - B\right)\right] \\
G' &= \left[G + \alpha\left(\beta_G\bar{G} - G\right)\right] \\
R' &= \left[B + \alpha\left(\beta_R\bar{R} - R\right)\right]
\end{aligned}
$$

(8)

to each colour channel and pixel, respectively, followed by a simple scaling to preserve the total brightness of the pixel $I = \sqrt{B^2 + G^2 + R^2}$. For the series SGP 2018, these factors were $\beta = (0.9, 0.78, 1)$ and $\alpha=0.4$. The coefficient $\alpha$ regulates the strength of the tinting such that $\alpha=0$ leads to no tint, and $\alpha=1$ produces an image of a single colour. This tinting is minimal, and linear colour behaviour is a reasonable assumption.

Step (2) is a stretch-and-shift process that identifies the horizon circle. Occasionally, a slight misalignment of camera and mirror axis leads to an elliptical appearance of the sky image. A calibration is necessary in such cases to stretch the visible horizon ellipse to circular shape, and to centre the horizon circle as close to the zenith as possible. A north-south alignment correction may also have to be applied. Both calibrations will ensure successful identification of the solar position in the next step. These calibrations become necessary if the TSI was not perfectly aligned in the field. They need to be readjusted after any disturbances occurred to the instrument, such as storms, snow, instrument maintenance, etc. Typically, this can be once every few months, or sometimes several times per month. It is important to check the calibrations regularly by sampling across the series whether the solar position was correctly identified after calibration. In addition, the horizon circle is placed at a zenith angle smaller than 90°, often between 85° and 79°, to eliminate the strong view distortion close to the horizon, and in some cases, objects present in the view. As explained earlier, the zenith angle resolution per pixel exceeds 1.2° close to the horizon. The information value for a solar zenith angle (SZA) larger than 80° is diminished. These pixels are excluded from the analysis. Practically, this is a very thin ring cut from the original image but does help eliminate false signals at low sun angles. The current process requires to find these calibrations for a small-sampling of images in a series, and to then apply them to all images in the series.

Step (3) removes the perspective distortion. The projection of the sky onto the plane of an image introduces a perspective distortion, as described in Long et al (Long et al., 2006). A coordinate transformation is performed to represent the sky within the horizon circle in terms of azimuth and zenith angle. The azimuth is the same in both projections. Zenith angle $\theta$ relates to the radial distance $r$ in the original image from the centre of the horizon circle as $r = R \sin \theta$. While R is not determined, image horizon radius $R_H$ and horizon zenith angle $\theta_H$ provide one known point to allow for proportional scaling. The coordinate transformation represents the sky circle in a way in which radial distance from zenith $s_z$ scales with zenith angle $\theta$ as

$$s_z = \frac{R_H}{\sin \theta_H} \times \theta$$

(9)

Long et al. (Long et al., 2006) discuss a further image distortion introduced by the particulars of the optics of the system of convex mirror and camera. The authors give an empirical correction curve for the SZA transformation. This correction is small; it has been omitted in this algorithm. One of the visible effects of this transformation concerns 22° halos: in the original TSI image, a halo appears as a horizontal ellipse; after the transformation it will have a shape closer to a circle.

Step (4) identifies the solar position and masks non-sky details. The position of the sun is marked based on the geographical position of the TSI and the Universal Time (UTC) of the image. Extraneous details, such as the shadow strip, the area outside the horizon circle, the camera, and the camera mount, are masked. The centre panel of Fig. 2 shows the image produced by all these adjustments up to step (4). Since often the position of the sun is detectable in the image, the marked sun position serves to refine the calibrations described above.

In step (5), the standardized local sky map (LSM) is created. A sketch of the layout of the LSM is provided in Figure 3. The LSM provides a standard sky section, centred at the sun, oriented with the horizon at the bottom, and presented in the same units for all possible TSI images (independent on the resolution of the original). Units of measurement in the LSM are closely related to angular degrees, but do not match perfectly due to a zenith-angle dependence of the azimuth arc length. The LSM is generated by rotating and cropping the image from step (4) to approximately within 40° of the sun, with the sun at its centre. The side length of the LSM in pixels scales with the previously determined horizon radius $R_H$ in pixels and the corresponding maximum zenith angle $\theta_H$ in ° as

$$w_{LSM}(pixels) = \frac{R_H(pixels)}{\theta_H(degrees)} \times 40°$$

(10)

Equation (10) provides a unit transformation between pixel positions and LSM units. For a TSI image of size 480×640 pixels, the LSM will have a size of approximately 240×240 pixels. For the earlier, smaller TSI images, the LSM has a size of approximately 140×140 pixels. The unit scaling includes the calibration choices $R_H$ and $\theta_H$, hence there is a slight variation in LSM side lengths. We eliminate the influence of the LSM sizes by performing all algorithm operations in standardized LSM units, which roughly correspond to angles of 1°. In other words, all LSM are equivalent to each other in terms of their LSM

units, but not in terms of pixel positions. At $\theta = 45°$, the arc length of azimuth angle φ is equivalent to the arc length of $\theta$ of same size; however, if $\theta > 45°$ the azimuth arc is stretched, requiring an additional horizontal compression to ensure equivalence of horizontal and vertical angular units. The LSM is divided into quadrants, shown in Figure 3, which are analyzed and classified separately by the algorithm described in the next section.

## 2.3 Computing Photographic Sky Type and Halo Properties

### 2.3.1 Average radial intensity (ARI)

Halos, as sun-centred circles, are creating a brightness signal at a scattering angle of 22°. We found it useful to analyse the radial brightness $I(s)$ with $s$ being the radial distance from the sun in the image plane, similar to the halo detection algorithm by Forster (Forster et al., 2017). The term intensity refers to the colour values of any of the colour channels, and varies between 0 and 255. There is a physical reason for using $I(s)$ in PST and halo assessment. The presence of scattering centres in the atmosphere influences the properties of sky brightness in the near-sun sky section. A very clear atmosphere, for example, exhibits an exponential decline, but with relatively high intensity values in the blue channel due to Rayleigh scattering. In case of cirrostratus, the increased forward scattering of larger particles (in this case ice crystals) leads to a decreased gradient of radial brightness, with more evenly distributed intensities in the red, green, and blue channels. In a partially cloudy sky, we would find sharp variations in $I(s)$, varying with colour channel. An overcast sky, on the other hand, may exhibit no decline in radial brightness, and will generally have low intensity values across all colour channels. A sketch of the LSM is given in Figure 3. The radial intensity $I(s)$ is computed using the colour intensity values of the image (0 to 255), separated by colour channel. The LSM is divided into four quadrants: TR = top right, BR = bottom right, BL = bottom left, TL = top left, analysed separately for quadrant scores, and then recombined for the image scores. The division into quarters allows to accommodate partial halos, low solar positions, and the influence of low clouds in partially obstructing the view to cirrostratus. The algorithm uses various properties of $I(s)$ to assign numeric PSTS and IHS, as detailed below.

The average radial intensity $I(s)$ is computed as an average over pixels at constant radial distance s from the sun. Due to the low resolution of the LSM, and due to some noise in the data, we average $I(s)$ over a circular ribbon with a width of 4 pixels, centred at $s$. Computing $I(s)$ over a thin ribbon addresses issues encountered when averaging over a circle in a coarse square grid, allowing continuity where otherwise pixilation may interrupt the line of the circle. Figure 4 shows the radial intensity of the red channel (R) in the bottom right quadrants of the LSMs featured in Fig. 2. Panel A includes $I(s)$, a linear fit, as well as the running average $\overline{I_6}$, plotted versus radial distance $s$. The running average is taken as the average of $I(s)$ over a width of 6 LSM units centred at $s$:

$$\overline{I_6}(s) = \frac{1}{N} \sum_{s-3LSMunits}^{s+3LSMunits} I(s)$$

(11)

The clear-sky image exhibits a lower red intensity overall than the halo image. The halo presents as a brightness fluctuation at about 21 LSM units. The analysis of $I(s)$ is undertaken in an interval between 15 and 26 LSM units, called the radial analysis interval (RAI). The RAI is marked in Figure 3. A linear fit yields a slope and intercept value used for the STS. We define the radial intensity deviation as

$$\eta(s) = I(s) - \bar{I}_6(s)$$

                                                                                                                              (12)

Panels B in Figure 4 show $\eta(s)$ for both situations. The details of the halo signal in $\eta(s)$ contribute in particular to the computation of the IHS.

### 2.3.2    Photographic Sky Type (PST)

The training sets for the properties of $I(s)$ were started for the set of 80 seed images mentioned in Section 1. Twenty images for each sky type were divided further by sky quadrants, yielding between 60 and 80 property sets for each sky type to initiate the training sets. Some quadrants were eliminated by horizon-near solar positions. The training quadrants were used to apprise the utility of $I(s)$ in making sky type assignments, with focus on the radial analysis interval (RAI) between 15 and 26 LSM units. The ten image properties used to compute the numeric PSTS are listed in Table 3. Also listed are the components of $M$
together with their standard deviations, computed from a later and more complete version of the training sets. The ten image properties include the slope and intercept of the line fit to $I(s)$ for each colour channel, where the slope characterizes a general brightness gradient, and the intercept gives access the overall brightness in the RAI. The line fit alone will not allow to differentiate partially cloudy skies from other sky types. However, the presence of sharply outlined clouds leads to a larger variation in intensity values, even for the same radial distance from the sun. The areal standard deviation (ASD) is an average
of the standard deviation of $I(s)$ for each radial distance s, averaged over all radii separated by colour channel. To set apart clear skies, the average colour ratio (ACR) in the analysis area is computed as

$$ACR = \overline{\frac{B^2}{GR}}$$

                                                                                                                              (13)

In Figure 5, the PST property set is represented graphically, including means, standard deviations, and extreme values as
observed for the completed training set. Clearly, no single property alone will suffice to assign a PST reliably. There is overlap in the extreme ranges. Relations between the colour channels are influential, as well. We are using the mechanism described in Section 2.1, Eqs. (1) through (6). The training sets for each class are collected in a master table, where $M$ and $\Sigma^{-1}$ for each PST are computed. As a new image is processed, its PST property vector $X$ is computed for each sky quadrant. Subsequently, a numeric score is computed for each sky type using Eq. (6). The coefficient $C_0$ in Eq. (6) for the PSTS computation is chosen
as $10^3$ which places a rough separator of order 1 between images that match closely a particular sky type, and those which do not. The raw values of $F_{image}$ in Eq. (6) vary greatly even between similar looking images, hence the PSTS is computed as a relative contribution between 0 and 100% for each sky type and each quadrant. For the PST-CS score this would mean:

$$PSTS\_CS = \frac{F_{CS}}{F_{CS} + F_{PCL} + F_{CLD} + F_{CLR}} \times 100\%$$

(14)

and equivalent for all other PTS classes. A single image quadrant can carry scores of 45% for PST-CS, 35% for PST-PCL, and 20% for PST-CLD. The dominant sky type then is PST-CS for this quadrant, since it contributes the largest score. The

PSTS for the image is assigned as the average over all quadrants. If the raw scores $F$ for all PSTs were smaller than $10^{-8}$ the quadrant is classified as N/A. It simply means that its properties are not close to any of the PST categories. Such conditions may include overexposed quadrants, horizon-near solar positions, a bird sitting on the mirror, and other conditions that produce images very different from the PST sought after. Also classified as N/A are quadrants in which the average radial intensity lies above 253 (overexposure), or contains a large fraction of horizon (bottom quadrants in low sun positions). A one-day sample

of sky type data is shown in Figure 6, for 10 March 2018. The day was chosen for its variability, including periods of each of the PST, as well as clearly visible halo periods. The central panel tracks the PSTS for all photographic sky types through the day, taken for all four LSM quadrants combined. It is important to note that the PST only can be representative of the section of sky near to the sun. Some of the late-day images in Figure 6 contain quadrants that were eliminated due to overexposure. The white scattering disk around the sun near the horizon does not allow for analysis, exemplified in the sample image at

22:53:00 UTC included in Fig. 6. For large portions of the day, the dominant sky types have been classified as PST-CS and PST-PCL, and the images corroborate this. The 14:36:00 image shows a thicker cloud cover, and the algorithm correctly responds by increasing the PST-CLD score. At 21:00:00, the algorithm indicates an increased PST-CLR score, consistent with the visual inspection of the TSI image at the time. Given the simplicity and physical relevance of this photographic sky type assessment, we believe that a radial scattering analysis around the sun has the potential to address some of the challenges that

have been encountered using a simple photographic cloud fraction in radiation modelling (Calbó and Sabburg, 2008; Ghonima et al., 2012; Kollias et al., 2007). The variation in radial intensity gradient as scatterers are present along the optical path can provide an alternative assessment for the presence of cirroform clouds, solving problems of classifying near-solar pixels using a colour ratio and/or intensity value only (Kennedy et al., 2016; N. Long et al., 2006). That will be a direction to discuss and explore in the future.

**2.3.3    Ice halo score (IHS)**

The 22° halo is a signal in the image that can be obscured by many other image features, including low clouds, partial clearings, inhomogeneous cirrostratus, regions of over-exposure, and near-horizon distortions. The appearances of 22° halos span a wide variety of sky conditions, ranging from almost clear skies to overcast altostratus skies, with the majority of halo phenomena appearing in cirrostratus skies. The challenge to extract the halo from such a wide variety of sky conditions is formidable.

While the statistical approach described in Section 2.1 will again form the core of the approach, the challenge shifts to defining a set of suitable discriminating properties of the image. In addition to the properties used in sky type assignment, the halo scoring must seek features in $\eta(s)$, Eq. (12), that are unique in halo images, such as a minimum followed by a maximum at

halo distance from the sun. The absolute values of $\eta(s)$ are dependent on various image conditions. Due to the variety of sky conditions, and variations in calibration and image quality, the values of maximum and minimum alone are not sufficient to reliably conclude the presence of a halo. We have found instances in which $\eta(s)$ does exhibit the halo maximum, but does not dip to negative values first. However, the upslope – crest – downslope sequence is consistently present in all cases of 22° halo.

The halo search is undertaken for a sequence of upslope – crest – downslope in terms of radial positions and range of slopes. All three characteristics present clearly in the derivative of the $\eta(s)$, the radial intensity deviation derivative $\eta'(s)$. This derivative of the discrete series $\eta(s)$ is approximated numerically by a secant methods as

$$\eta'_i \approx \frac{\eta_{i+1} - \eta_{i-1}}{s_{i+1} - s_{i-1}}$$

10 (15)

In Figure 7, both $\eta(s)$ and $\eta'(s)$ are shown for the bottom-right quadrant of the green channel of the halo image in Figure 2. The sequence of radial halo markers is illustrated in Figure 7. The algorithm computes $\eta'(s)$ and seeks the positive maximum and the subsequent negative minimum, plus the radial position of the sign-change between them. This produces a sequence of radial locations $s_{up}$, $s_{max}$, and $s_{down}$ which basically outline the halo bump in width and location. There are often multiple maxima

of $\eta'(s)$ contained in the RAI. A halo image typically has fewer maxima than a non-halo image, but of larger amplitude. Therefore, the number of maxima as well as the upslope value $\eta'_{up}$ and down-slope derivative $\eta'_{down}$ join the set of halo indicators. If multiple maxima are found, the dominant range is used. Lastly, a radial sequence should be consistent across all three colour channels. The resolution of the TSI images only allows to resolve 0.4° to 1.2° with certainty; in addition variations in calibration and SZA do influence deviations from the expected 22° position. The separation of colours observed in a 22°

halo display is not resolved with statistical significance, therefore this was not used as a criterion for halo detection. The standard deviation of all three radial positions across the three colour channels was added to the halo scoring set of properties. We arrive at a set of 31 properties for the computation of the IHS, listed in Table 4, together with their means and standard deviations. The mean value vector M and the inverse covariance matrix $\Sigma^{-1}$ are computed in the master table and then imported by the halo searching algorithm for use in Eq. (6). The coefficient $C_0$ in Eq. (6) is arbitrary. In the IHS computation, a value of

$10^6$ was chosen for $C_0$ which places a rough separator of order one between image quadrants that do have a halo, and those which do not. While the scoring of individual images works very well for true halo images, it does trigger the occasional halo score for images that do not exhibit a halo. This may occur due to inhomogeneities in a broken cloud cover, or other isolated circumstances. These false halo scores often occur on isolated images. We utilize the factor of residence time of a halo to address this. In a 30-s binned series of TSI images, the halo will appear usually in a sequence of subsequent images, often in

the order of minutes or even hours. We added a Gaussian broadening to the time series of halo scores $F_i$, taken at times $t_i$ with a broadening $w$

$$IHS(t) = \sum_{t_i=t-3w}^{t_i=t+3w} F(t_i) \exp\left[-\frac{(t_i - t)^2}{2w^2}\right]$$

(16)

This de-emphasizes isolated instances, and enforces sequences of halo scores, even if they individually exhibit weak signals or gaps. This procedure reduced the false halo identifications significantly. Just as for the PSTS, the training set for the IHS in the master table is being complemented as more images are analysed. The raw halo score $F$ is computed for each of the four quadrants of an individual image, their average is used to assign the raw score for the whole image. The broadening in Eq. (16) was chosen as $w=7$ images throughout, corresponding to 3.5 minutes. In Figure 6, the clear 22° halo between 19:00 and 20:00 UTC produces a strong IHS. There are a few weaker halo signals, and upon inspection of the images we find that these correspond to partial halos (like at 17:07:00), or halos in a more variable sky.

## 3    Results for January through April 2018

We chose the record of the month of March of 2018 at the SGP location for a thorough comparison of algorithm results to visual image inspection, as well as an expansion of the training set. The complete month TSI record, starting at 1 March 2018 0:00:00 UTC and ending at 31 March 2018 23:59:30 UTC, contains 44,057 images. Only 31,398 of were classifiable in terms of their PST. Exclusions occur due to large SZA, overexposure, or low PSTS.

The algorithm and the current training set (starting with the eighty sets discussed above) is used to assign an image IHS and a set of four image PSTS, averaging over the quadrant IHS and PSTS values. Both of these score sets are continuous numerical values, resulting in a time-resolved scoring for all PSTS and IHS values as shown in Figure 6, across the month of March. In order to manage comparison to a visual classification of these images, and to learn how both score sets behave in terms of numerical values, the following two procedural steps are added in the post-processing: (1) For the PST, the sky type with the maximum contribution is taken as the image sky type; (2) an IHS discriminator is used to assign a halo/no halo designator to an image. This IHS discriminator is arbitrary, not part of the image analysis algorithm, and dependent on factors such as $w$ and $C_0$, the quality of the calibration, and the quality and relevance of the training set. The algorithm assigns a continuous IHS to every image as a number varying between $10^{-10}$ and $10^6$, with fluid continuous change in consecutive images. The decision on the value of the discriminator is based on the behaviour of the timeline. Halo images generate a significant peak above a population of low-level peaks. The discriminator is placed to exclude about 75% of the low peaks when analysing for a *count* of halo incidences. Our testing, minimizing false negatives and maximizing correct positives, places it at around 4000 for the month of March.

Visual image classification for so many images poses a considerable challenge, which we approached in form of an iteration. For each of the 31 days of March, an observer assigned sky classifications to segments of the day by inspecting the day series as an animation. This can easily be done by using an image viewer and continuously scrolling through the series. Then, the

day would be subjected to the algorithm. The sections of the record in which visual and algorithm differed were inspected again, at which point either the visual assessment was adjusted, or samples of the misclassified images were added to the training set. Adjustment to visual classifications often occurred at the fringes of a transition. For example, when a sky transitions from cirrostratus to altostratus to stratus, the transitions are not sharp. The observer sets an image as the point in

which the sky moved from PST-CS to PST-CLD, but the criteria in the algorithm would still indicate PST-CS. This can affect up to a hundred images at transition times, which then were reclassified. On the other hand, if a clearly visible halo was missed by the algorithm in form of a low numerical IHS, a couple of new lines were added to the training set, selected from the few hundred quadrant cases in which this particular halo had scored low. The IHS discriminator is not part of the algorithm itself, but follows in the post-processing from the general behaviour of the IHS across the month. It is a tool to allow a comparison,

but not an ultimate answer to halo strength. Halo strength could be assessed by the IHS. After each change to the training set, the algorithm would be repeated, and recalibrations to the visual record, as well as to the training set were made. The process was repeated several times until no more gains in accuracy were observed. The training sets at the end of this process contained between 93 and 188 property records, of which up to 50% were taken from March 2018. Compared to the number 31,398 of classifiable images in March (after exclusion of high-SZA, overexposure, and other), and considering that each of these images

contributes up to four individual property sets, the number of training sets is indeed diminutive. These adjustments were done by SB.

The resulting time lines for PSTS and IHS for the month of March are plotted in Figure 8. Many of the images exhibit strong indicators for multiple PST. The largest PSTS is used to assign a PST to an image. As expected, the high halo scores coincide with strong PST-CS signals. Noteworthy is also, that there are a number of days in which PST-CS does not carry a 22° halo,

indicated by very small IHS values.

In Table 5, visual and algorithm results of the sky type assignments are cross-listed for SGP March 2018. It is worth reminding the reader that PST are assigned only for the radial analysis interval indicated in Figure 3. Table 5A lists the percentage of visually assigned sky types that correspond to the algorithm-assigned PTS; B lists the percentage of algorithm-assignet PTS that also have been identified as a visual sky type. For example, the algorithm correctly identifies 88 % of all visual CS skies

as PST-CS (part A); 86% of the images classified as PST-CS by the algorithm also have been visually classified as CS (part B). PST-CLD is reliably identified by the algorithm. A small percentage (3%) of visual PST-CLD skies trigger a PST-PCL signal, mostly due to inhomogeneities in cloud cover. The algorithm classifies 95% of all visual PST-CLR skies correctly. Differentiating between PST-CS and PST-PCL is successful. However, these two sky types pose some difficulties. For example, 8.5% if visual PST-CS skies scored a PST-CLR signal, and 10% of images classified as PST-CS were visually

assigned a PST-PCL sky type. In these cases we often found that the algorithm assignment might be more persuasive than the visual assignment – a visual assignment is a subjective call, and open to interpretation of the observer. Combined with image distortion and resolution limits, it is quite possible that the visual assignments carry a considerable uncertainty. Some of the visual PST-CS skies, for example, present to the eye as PST-CLR, but reveal the movement of a thin cirrostratus layer if viewed in context of time-development (animation). Similarly, cirrostratus may present as an inhomogeneous layer in

transition skies, triggering a PST-PCL assessment in the algorithm. Low solar positions are prone to larger image distortion, which may lead to misinterpretation. It is worth noting that every image quadrant receives a PSTS for all classes of PST. In cases of mismatch, we often find that the two sky types at conflict both contribute significantly to the PSTS of the image quadrant. If SZA > 68°, no PST assignments were made. Most of the 397 PST-CLR images that presented as PST-CS to the algorithm were taken at very low sun, with a significant over-exposure disk in near-solar position. Table 5 also lists a comparison of visual halo identifications with the algorithm scores. According to this assessment, the algorithm correctly calls 85 % of visual halo images, while not diagnosing 15 % of them. On the other hand, 12 % of all halo signals do not correspond to a halo in the image. One can improve the correct identification rate by lowering the cut-off score, on the cost of an increase in the signal from false identifications. Balancing the false positive and false negatives yields a reliability of about 12 to 14 %. Some of the false negatives arise from altocumulus skies, in which the outlines of cloudlets may trigger halo signals by their distribution and size. These are very difficult to discriminate from visual halo images. Some images were flagged with an IHS by the algorithm, and the presence of a weak halo revealed itself only after secondary and tertiary inspection of the image. Caution is advised in relying heavily on visual classifications of TSI images alone. The visual sky type and halo assignments themselves have an uncertainty due to subjectivity. While it is easy to distinguish a partially cloudy sky from a clear sky, this may become difficult for the difference between thick cirrostratus and stratus. Their visual appearances may be quite similar. Sometimes, an assignment can be made in context of temporal changes. Some clear-appearing skies reveal a thin cirrostratus presence if viewed in a time series instead of in an individual image. It is therefore a future necessity to combine the visual assignments of sky types with LIDAR data for altitude, optical thickness, and depolarization measurements to make an accurate assessment of the efficacy of the PST identification, following closely the processes described by Sassen et al. (Sassen et al., 2003) and Forster et al. (Forster et al., 2017).

We applied the algorithm to the TSI record for the first four months of 2018 for the SGP ARM site. It is worth noting that this paper is not intended to present a complete exploration of the ARM record concerning 22° halos. We are, however, including a demonstration of capacity of the algorithm presented here. Table 6 summarizes our findings. It lists the percentages for the PST by month. A portion of the images has not been assigned with a PSTS. The conditions under which this occurs have been described earlier, and include horizon-near solar positions, images with over-exposure in the RAI, and images for which the raw PSTS for each sky type was numerically too low to be considered a reliable assessment. Therefore, PST percentages refer only to all identified images. January and March exhibited a large fraction of clear skies. February was dominated by cloudy skies, while April registered a high percentage of PST-CS. Only a partial month of images was available for April. Cloud types depend strongly on the synoptic situation. That means that no further conclusions should be made from these data without expanding the data set. The 22° halo statistics in Table 6 lists data on the 22° halo, including duration, number of incidents, and data on partial halos. The partial halo data are based on the individual quadrant IHS for an image, while the image score is used for duration and incidence information. The number of separate halo incidences counts sequences of images for which the IHS did not fall below the cut-off value of 4000. While it is worth noting that the number of incidences lies in the order of magnitude of the number of days in a month, it is certain that the halo instances are not evenly distributed. Figure 8 does

demonstrate this behaviour. However, even on a day of persistent cirrostratus with 22° halo, interruptions of its visibility can occur. Sometimes low stratocumulus may obscure the view of the halo, sometimes the cirrus layer is not homogeneous. This may lead to a large number of separate halo incidences in a short time, while none are counted at other times. The mean duration of a halo incident lies between 16 and 34 minutes, depending on month. We listed the maximum duration found in each month as well. The longest halo display in the time period occurred in April 2018, with nearly 3.5 hours. Mean values are easily skewed by a few long-lasting displays. Figure 9 shows the distribution of 22° halo durations for the four months. The most common duration of a 22° halo lies between 5 and 10 minutes, followed by 10 to 15 minutes. Due to the time-broadening applied via Eq. (16), the display time cannot be resolved below 3 minutes. We consider the fraction of images in which a halo was registering. That marker varied between 3.9% for January and 9.4% for April. In accord with findings in (Sassen et al., 2003), we find a low amount of halo display activity in January. However, this may be influenced by the large SZA in January. The closer the sun to the horizon, the more TSI images have been excluded from the analysis, and the stronger the influence of distortion.

Occasionally, only partial halos will be seen, depending on positioning of the cirroform clouds and on obstruction by low clouds. The division of the LSM into quadrants allows to assess the possibility of fractional halos, as indicated in Table 6. The overwhelming portion of halo incidences shows full or 75% halo. This means that in four or three of the quadrants, the IHS has exceeded its minimum cut-off. Quarter halos have only rarely registered in the algorithm. Many of the half halos can be found for images taken close to sunrise or sunset. That explains their relative frequency in January and February.

We started the project with the goal to find information on cirrostratus composition, in particular with respect to assessments of variation of smooth versus rough crystals. Forster et al (Forster et al., 2017) discuss that the necessary fraction of smooth crystals for a halo appearance lies between 10% and 40%. The bottom part of Table 6 investigates the relation between sky type and 22° halo incidences. The first set of data in the Relations section of Table 6 gives the fraction of each sky type, as it produced a 22° halo incident. For example, in January we found that 9 % of PST-CS were accompanied by a 22° halo. In the data for April, this fraction increased to 22% of PST-CS. We also have registered halos for a portion of PST-PCL, and for PST-CLD. No halos have been registered in any of PST-CLR. The April data are consistent with the observations of Forster et al (Forster et al., 2017) who report a 22° halo for 25% of all cirrus clouds for a 2.5-year photographic record taken in Munich, Germany. Differences exist, however, in that the Forster observations verified ice cloud with LIDAR and IR measurements, while this current record compares to a photographically assigned sky type. We must consider reasons for the PST-PCL and PST-CLD halo incidences. Upon random sampling of these combinations we find the following: The PST-PCL indicator has been assigned to images that have a highly varied cirroform sky, including halo appearances. In a few instances, low clouds triggered the PST-PCL indicator, however, a cirroform layer at higher altitude still contributed a halo score above the threshold. Many of the halo scores in PST-CLD skies belong to images with an overcast appearance, however, most likely belong to a thickening and lowering cirro- or altostratus as often found in warm front approaches. These are not false scores, but conditioned by the limitations of the PST classification. The second set of numbers in Table 6 shows the fraction of all halos

associated with the various PST. In January, 49% of all halo incidences occurred in PST-CS skies, while in March this number was 87%. As for the overall frequency of halo displays, we can refer to Table 6, in which the observed halo frequency for all PST combined is listed. It varies from 3.9% in January 2018 to 9.4% in April 2018. The closest comparison is the number given by Sassen et al. (Sassen et al., 2003) who report a full 22° halo at 6% of the 10-year record of hourly images, while any

halo feature was observed at 37.3 % of time. For such a comparison, Forster et al. (Forster et al., 2017) is cautioning that a statistics like this may strongly depend on the binning interval.

With this image analysis algorithm used on TSI images to identify the PST and the appearance of 22° halos, the next useful and logical step will be to relate these data to other instrument records: LIDAR for altitude, particle density, and particle phase (solid or liquid), photometric measurements to glean information on radiative flux. ARM sites have accumulated such

instrumental data. The algorithm proposed here will make such data investigation possible.

Finally, it is worth discussing the general approach of the TSI algorithm in comparison to the halo detection algorithm developed by Forster et al (Forster et al., 2017). Both algorithms utilize features found in the radial intensity $I(s)$, such as the sequence of minimum – maximum at the expected radial positions in order to find halos in an image. The random forest classifier approach described in (Forster et al., 2017) is a machine learning approach that arrives at a binary conclusion for an

image in form of halo/no halo. Their algorithm was trained on a visually classified set of images in order to construct a suitable decision tree. In addition to 22°halos, the Forster algorithm also identifies parhelia and other halo display features in images taken by a high-resolution, sun-tracking halo camera. The algorithm presented here for TSI data must work with a much less specialized set of images, notably of lower resolution. It does not characterize halos in a binary decision, but rather assigns a continuous ice halo score to an image, in addition to photographic sky type scores for four different types of sky conditions.

Similar to the Forster algorithm, the TSI algorithm also was trained on a visually classified set of images. For the algorithm presented here, further training sets are easily added. Both algorithms have overlap. The TSI algorithm makes extensive use of the radial brightness gradient (slope) for the sky type assignments. The relation of this gradient to the physical presence of scatterers along the optical path makes this an attractive approach.

## 4    Summary

ARM sites have produced long-term records of sky images. We developed an algorithm that assigns sky type and halo scores to long-term series of TSI images with the goal of using these long-term image records to provide supporting information the presence of smooth, hexagonal ice crystals in cirrus clouds from observations of 22° halos. We described this algorithm in this paper, including the image preparation to generate a standardized image section centred at the sun, called the Local Sky Map (LSM). A multivariate analysis of selected LSM properties, as supported by a master table, allows the assignment of scores

with respect to photographic sky type and 22° halo presence in the solar-near section of the sky. In particular, we focus on the properties associated with the radial brightness behaviour around the sun. Physically, the number and type of scattering centres in the atmosphere does influence the radial brightness gradient, thus giving us access to an assessment of cloud type and cloud

cover. The brightness fluctuation associated with the 22° halo provides a further set of properties specific to the presence of a 22° halo. We analyse all four quadrants adjacent to the sun separately, then combine the scores into a raw image score. For the ice halo score, we also apply a Gaussian broadening across the time series. The algorithm has been found to be about 90% in agreement with the visually assigned sky type, and 85% in agreement with the visually identified ice halo score. The application to the first four months of the TSI records from SGP ARM site indicates periods of halo displays, with a most common duration of about 5 to 10 minutes, but lasting up to 3 hours. It allowed to identify the fraction of PST-CS skies that do produce halo displays, as well as find such data for other PST. In the future, the algorithm will be applied to deliver 22° halo data for the long-term TSI records accumulated in various geographical locations of ARM sites, and allows further investigation into correlations with other instrumental records from those sites. In particular, LIDAR data for altitude and optical thickness measurements, as well as depolarization analysis will be a useful combination with this photographic halo display record. It is reasonable to expect that the reference set for sky type determination will improve with the support of LIDAR data. The method described here may be suitable to expand to other types of sky analysis on TSI images.

## Author contribution

Sylke Boyd is the main author of this paper and the code. The four co-authors worked on the algorithm as undergraduate researchers. Stephen Sorenson decided on the use of C++ and opencv3.2 for image manipulation, and initiated the program code. Shelby Richard worked out the details of the radial intensity computation and properties. Michelle King and Morton Greenslit contributed algorithm parts to eliminate optical distortions and low-cloud obstruction, and input management. SR, MK, and MG all contributed to data collection and analysis.

## Acknowledgement

Data were obtained from the Atmospheric Radiation Measurement (ARM) Program sponsored by the U.S. Department of Energy, Office of Science, Office of Biological and Environmental Research, Climate and Environmental Sciences Division. The work was supported by The Undergraduate Research Opportunities Program (UROP) at the University of Minnesota, as well as a grant to the University of Minnesota, Morris from the Howard Hughes Medical Institute through the Precollege and Undergraduate Science Education Program. SB wishes to thank the University of Minnesota-Morris for the generous one-semester release from teaching obligations, allowing the completion of this work.

## Competing Interests

The authors declare that they have no conflict of interest.

## Code availability

Code and accessory files are made available at github under DOI 10.5281/zenodo.8475 (Boyd et al., 2018).

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

**Table 1. TSI data set properties. Seed images for the algorithm were taken from all three locations.**

| Location | Dates and times (UTC) | | Image interval | Resolution (pixels) |
|---|---|---|---|---|
| Southern Great Plains (SGP, 2018) | 2 Jul 2000 0:35:00 | 15 Aug 2011 01:17:30 | 30 s | 288×352 |
| 36° 36′ 18″ N, 97° 29′ 6″ W | 15 Aug 2011 22:17:30 | 19 Apr 2018 01:02:00 | 30 s | 480×640 |
| North Slope of Alaska (NSA, 2018) | 25 Apr 2006 21:44:00 | 2 Nov 2010 21:31:00 | 30 s | 288×352 |
| 71° 19′ 22.8″ N, 156° 36′ 32.4″ W | 9 Mar 2011 01:08:30 | 11 Apr 2018 18:59:30 | 30 s | 480×640 |
| Eastern North Atlantic (ENA, 2018) | 1 Oct 2013 08:13:00 | 28 May 2018 21:04:00 | 30 s | 480×640 |
| 39° 5′ 29.76″ N, 28° 1′ 32.52″ W | | | | |

**Table 2. Descriptions of the photographic sky types (PST)**

| Sky type | | Visual description |
|---|---|---|
| Cirrostratus | PST-CS | Muted blue, no sharp cloud outlines; solar position clearly visible, bright scattering disk or halo may be present; changes are gradual and slow (several minutes) |
| Partly cloudy | PST-PCL | Variable sky with sharply outlined stratocumulus or altocumulus; variations between sky quadrants; sun may be obscured; changes are abrupt and fast (less than two minutes) |
| Cloudy | PST-CLD | Sun is obscured; low brightness; low blue intensity values; stratus, nimbostratus, altostratus, or cumulonimbus; changes occur slowly (order of hours) |
| Clear | PST-CLR | Blue, cloud-free sky; sun clearly visible and no bright scattering disk around it; changes are slow (order of hours) |
| No data | N/A | This may occur at low sun positions for the bottom quadrants of the LSM, or due to overexposure in the near-solar region of the image; it's the default at night. |

**Table 3. Discriminant properties used to classify the photographic sky type. Averages, and standard deviations for the training set of each class are listed. All units based on colour intensity values and LSM units. Number of records for each sky type is indicated in parentheses.**

| PST property | PST-CS (155) | PST-PCL (99) | PST-CLD (93) | PST-CLR (96) |
|---|---|---|---|---|
| Slope $a$ | B  -3.0 ±1.5 | B  -1.6 ±2.2 | B  -0.7 ±1.7 | B  -2.3 ±1.6 |
| | G  -3.2 ±1.7 | G  -1.6 ±2.2 | G  -0.7 ±1.7 | G  -2.8 ±1.6 |
| | R  -3.6 ±1.9 | R  -1.9 ±2.6 | R  -0.8 ±1.8 | R  -2.8 ±1.7 |
| Intercept $b$ | B  276 ±34 | B  248 ±46 | B  193 ±40 | B  248 ±43 |
| | G  271 ±33 | G  240 ±53 | G  195 ±44 | G  233 ±47 |
| | R  255 ±48 | R  228 ±65 | R  179 ±47 | R  184 ±47 |
| ASD[1] | B  13.1 ±5.3 | B  20.5 ±7.0 | B  14.2 ±5.0 | B  15.4 ±5.2 |
| | G  15.0 ±6.0 | G  22.9 ±7.7 | G 15.0 ±5.1 | G 16.3 ±5.3 |
| | R  16.6 ±6.6 | R  25.5 ±8.1 | R  15.8 ±5.6 | R  14.8 ±5.7 |
| ACR[2] | 1.33 ±0.36 | 1.24 ±0.32 | 1.08 ±0.12 | 2.07 ±0.11 |

[1] Areal Standard Deviation; [2]Average Colour Ratio

**Table 4. Discriminant properties used for the ice halo score. Averages and standard deviations for a training set of 188 quadrant records are listed. All units based on colour intensity values and LSM units.**

| IHS property | B | G | R |
|---|---|---|---|
| Slope $a$ | -3.3 ±1.5 | -3.3 ±1.6 | -3.8 ±1.8 |
| Intercept $b$ | 279 ±35 | 278 ±37 | 268 ±45 |
| ASD | 12.6 ±4.7 | 14.8 ±6.0 | 16.2 ±6.4 |
| Maximum upslope $\eta'_{up}$ | 2.1 ±1.3 | 2.1 ±1.4 | 2.5 ±1.6 |
| Maximum downslope $\eta'_{down}$ | -1.6 ±1.0 | -1.6 ±1.0 | -1.8 ±1.1 |
| Upslope location $s_{up}$ | 17.5 ±1.9 | 17.8 ±2.3 | 17.5 ±2.1 |
| Maximum location $s_{max}$ | 18.9 ±1.9 | 19.1 ±2.3 | 18.8 ±2.1 |
| Downslope location $s_{down}$ | 20.0 ±2.1 | 20.2 ±2.4 | 19.9 ±2.2 |
| Number of maxima $n_{max}$ | 2.4 | 2.6 | 2.5 |
| BGR consistency | $\sigma_{BGR}(s_{up}) = 0.8$ | $\sigma_{BGR}(s_{max}) = 0.8$ | $\sigma_{BGR}(s_{down}) = 0.9$ |
| ACR | | 1.2±0.3 | |

**Table 5. Algorithm versus visual classifications for SGP March 2018. Part A shows the percentage of visual assignments corresponding to algorithm assignments; Part B shows the percentage of algorithm assignments and how they distribute among the visual assignments. For example, 88% of all visual CS skies are classified as PST-CS by the algorithm, but only 86% of all algorithm PST-CS skies also identify as visual CS. Agreement combinations in bold. IHS > 4000 to count an algorithm halo.**

| A | Percentage of visually assigned sky type which corresponds to algorithm-assigned PST | | | | | | | |
| | CS | | PCL | | CLD | | CLR | |
| --- | --- | --- | --- | --- | --- | --- | --- | --- |
| | N | % | N | % | N | % | N | % |
| PST-CS | **6675** | **88** | 683 | 11 | 38 | 1 | 397 | 4 |
| PST-PCL | 182 | 2 | **5513** | **86** | 176 | 3 | 191 | 2 |
| PST-CLD | 61 | 1 | 47 | 1 | **6129** | **97** | 0 | 0 |
| PST-CLR | 641 | 8 | 136 | 2 | 0 | 0 | **10529** | **95** |
| N/A | 12597 (40% of all images) | | | | | | | |
| | Percentage of visually assigned halos which corresponds to the algorithm assignment | | | | | | | |
| | 22° halo | | | | No 22° halo | | | |
| | N | | % | | N | | % | |
| 22° halo | **1996** | | **85** | | 272 | | 1 | |
| No 22° halo | 349 | | 15 | | **41409** | | **99** | |

| B | Percentage algorithm-assigned PST which corresponds to a visually assigned sky type | | | | | | | |
| | CS | | PCL | | CLD | | CLR | |
| --- | --- | --- | --- | --- | --- | --- | --- | --- |
| | N | % | N | % | N | % | N | % |
| PST-CS | **6675** | **86** | 683 | 9 | 38 | 0 | 397 | 5 |
| PST-PCL | 182 | 3 | **5513** | **91** | 176 | 3 | 191 | 4 |
| PST-CLD | 61 | 1 | 47 | 1 | **6129** | **98** | 0 | 0 |
| PST-CLR | 641 | 6 | 136 | 1 | 0 | 0 | **10529** | **93** |
| N/A | 12597 (40% of all images) | | | | | | | |
| | Percentage of algorithm-assigned assigned halos which corresponds to a visual assignment | | | | | | | |
| | 22° halo | | | | No 22° halo | | | |
| | N | | % | | N | | % | |
| 22° halo | **1996** | | **88** | | 272 | | 12 | |
| No 22° halo | 349 | | 1 | | **41409** | | **99** | |

**Table 6. PST and 22° halo formations during the months of January through April 2018, SGP. Percentages are with respect to all classifiable images. Times are UTC.**

| | | Jan 2018 | Feb 2018 | Mar 2018 | Apr 2018[1] |
|---|---|---|---|---|---|
| | total number of images | 36632 | 36011 | 44057 | 27741 |
| | Number with valid PST | 21238 | 23604 | 31398 | 20436 |
| | begin date of record | 1 Jan 2018 13:47:00 | 1 Feb 2018 13:36:00 | 1 Mar 2018 0:00:00 | 1 Apr 2018 0:00:00 |
| | end date of record | 31 Jan 2018 23:50:00 | 28 Feb 2018 23:59:30 | 31 Mar 2018 23:59:30 | 19 Apr 2018 1:02:00 |
| PST | PST-CS | 20 % | 18 % | 25 % | 34 % |
| | PST-PCL | 24 % | 24 % | 19 % | 19 % |
| | PST-CLD | 11 % | 33 % | 20 % | 25 % |
| | PST-CLR | 45 % | 25 % | 36 % | 22 % |
| 22° halos | Number of separate halo incidents | 26 | 45 | 34 | 46 |
| | Mean duration | 16 min | 22 min | 34 min | 21 min |
| | Maximum duration | 62 min | 136 min | 171 min | 208 min |
| | Total halo time | 411 min | 998 min | 1160 min | 963 min |
| | % halo instances with | | | | |
| | $^4/_4$ 22° halo | 29 % | 42 % | 77 % | 42 % |
| | ¾ 22° halo | 38 % | 31 % | 13 % | 40 % |
| | ½ 22° halo | 32 % | 25 % | 10 % | 18 % |
| | ¼ 22° halo | 1 % | 1 % | 0 % | 0 % |
| Relations | % halo instances of all sky type instances | | | | |
| | PST-CS | 9 % | 16 % | 18 % | 22 % |
| | PST-PCL | 6 % | 7 % | 6 % | 9 % |
| | PST-CLD | 4 % | 5 % | 10 % | 12 % |
| | PST-CLR | 0 % | 0 % | 0 % | 0 % |
| | All STS | 3.9 % | 8.5 % | 7.4 % | 9.4 % |
| | % sky type of all halo instances | | | | |
| | PST-CS | 49 % | 60 % | 87 % | 78 % |
| | PST-PCL | 42% | 33 % | 9 % | 14 % |
| | PST-CLD | 2 % | 5 % | 3 % | 5% |
| | PST-CLR | 0 % | 0 % | 0 % | 0 % |
| | N/A | 7 % | 2 % | 1 % | 3 % |

[1]incomplete month

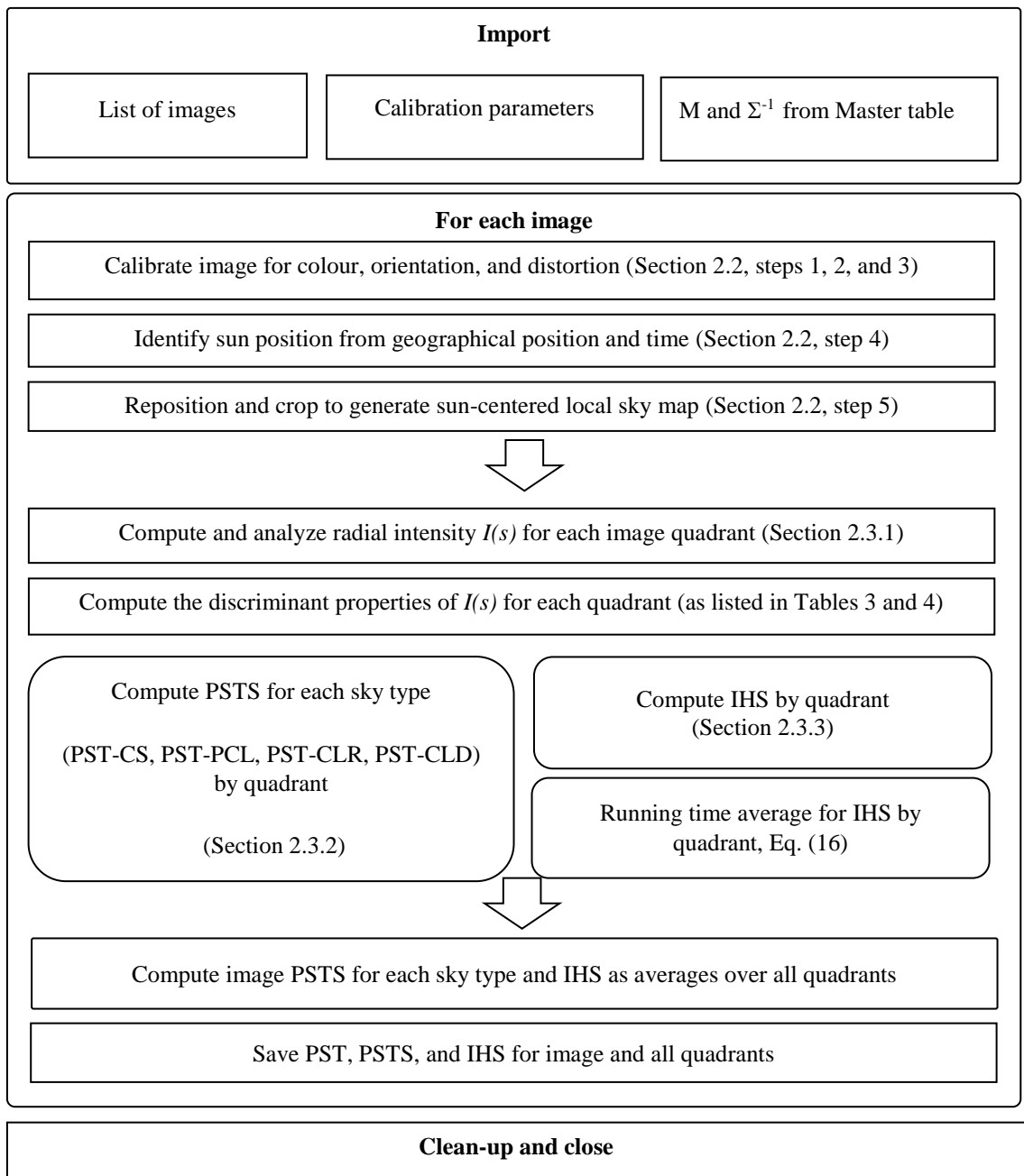

**Figure 1. Flow chart of the algorithm for the analysis of TSI images.**

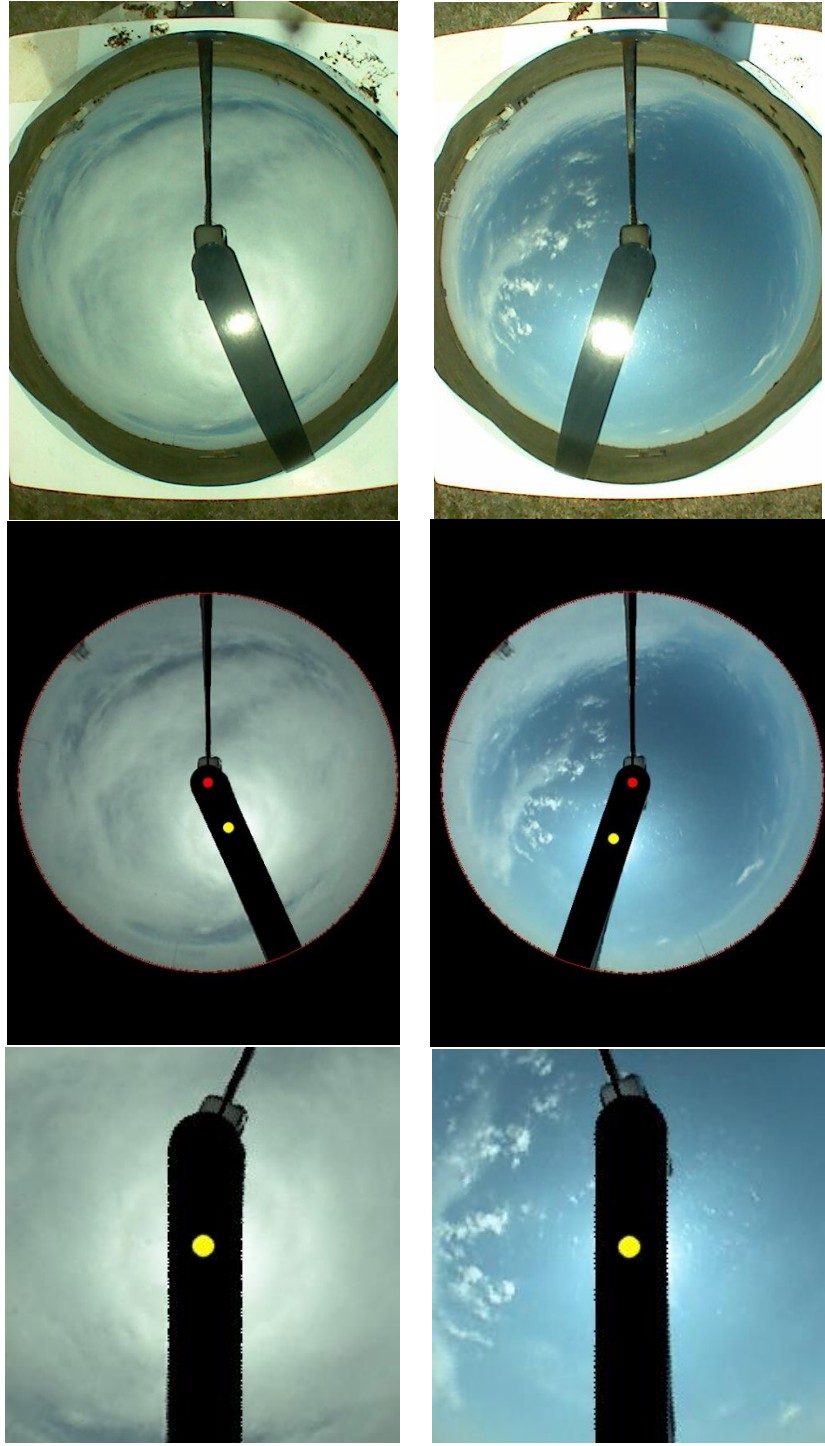

**Figure 2 Two examples for image preparation. The left column develops an image from SGP 17 April 2018 17:45:00 UTC, the right image was taken on SGP 3 April 2018 19:09:30 UTC. Top row: original image; centre row: image after colour correction, distortion removal, masking of horizon and equipment, and sun mark were applied; bottom row: final local sky map with sun at centre and a width of about 80 LSM units.**

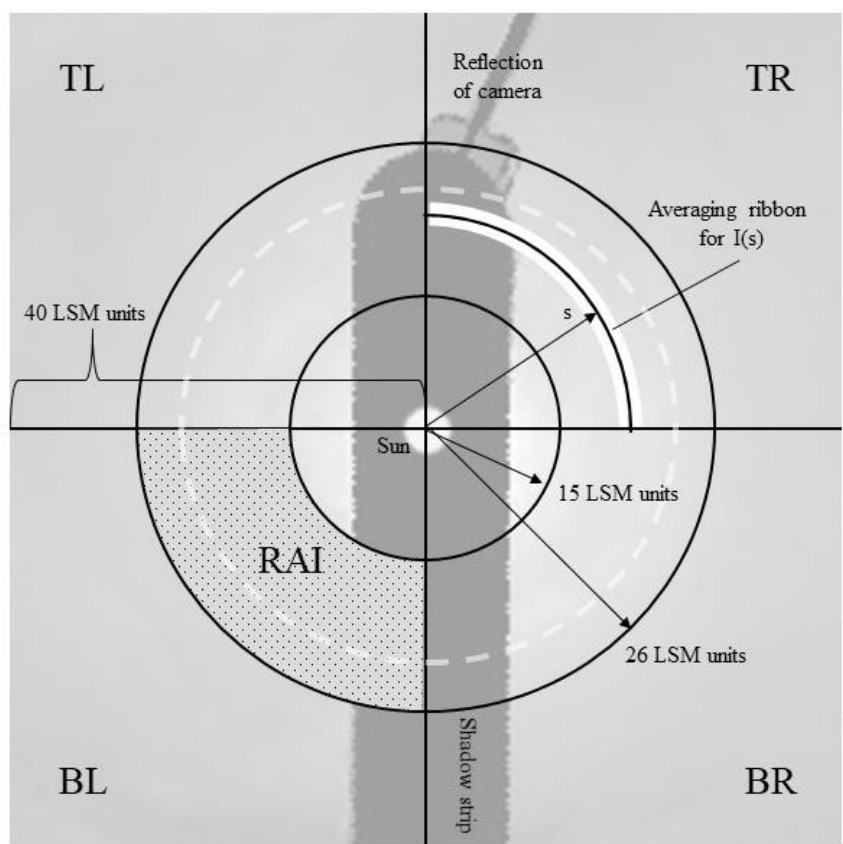

**Figure 3. Layout of the local sky map (LSM). The LSM is divided into four quadrants, named according to their position as TR – top right, BR – bottom right, BL – bottom left, and TL – top left. The RAI is the Radial Analysis Interval for which STS and IHS properties are evaluated. The approximate position of the halo maximum is sketched in light gray. Shadow strip and camera are excluded from analysis.**

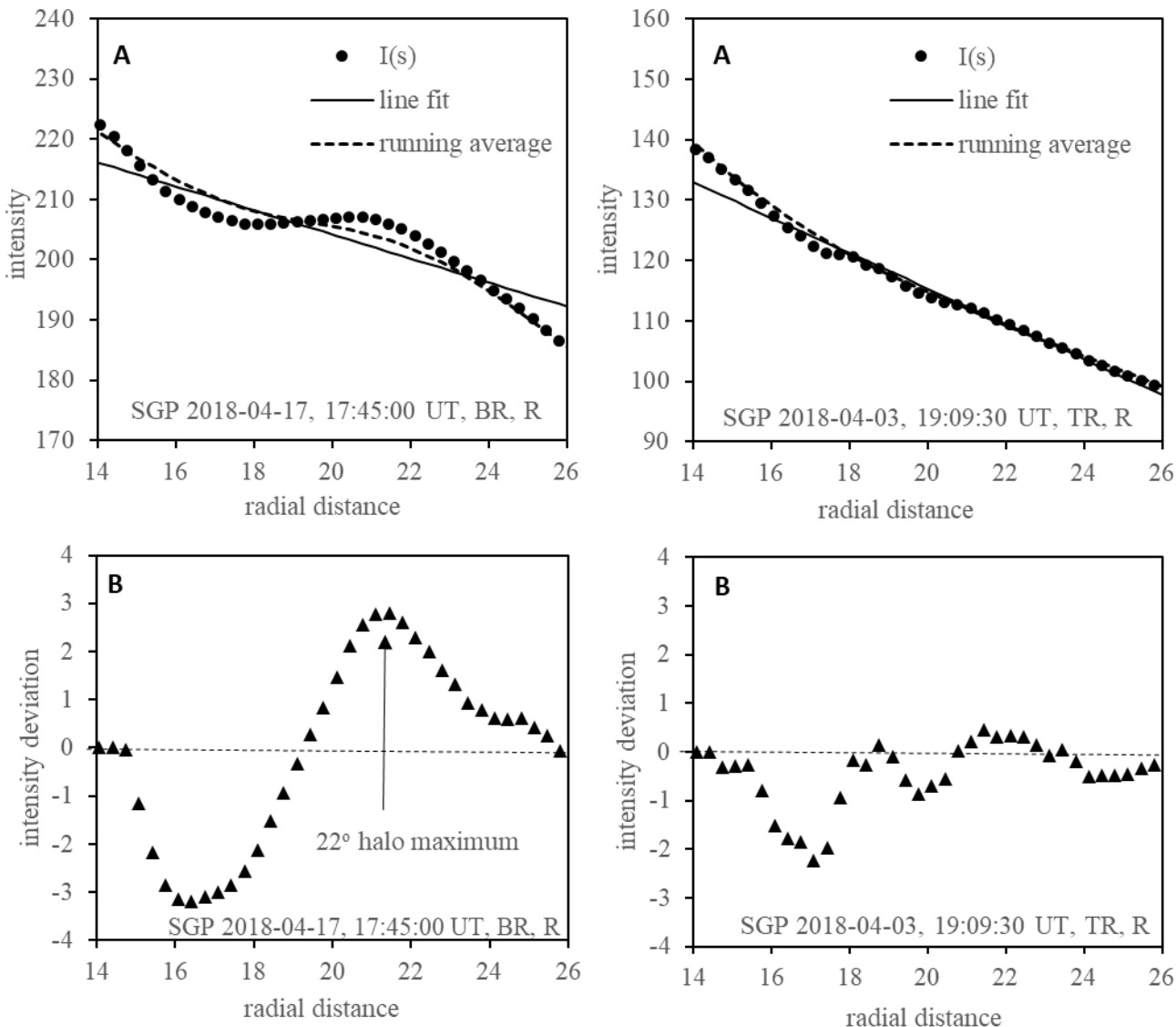

**Figure 4. Average radial intensity of the red channel is shown versus radial distance s, measured in LSM units, for the two images of Fig. 2, halo at left. Panel (A) includes the average intensity $I(s)$, a linear fit, and the running average $\bar{I}_6(s)$ as averaged over a width of 6 LSM units. (B) shows the radial intensity deviation $\eta(s)$. The halo signal is visible as a minimum at 17 LSM units, followed by a maximum at 21 LSM units in the left column.**

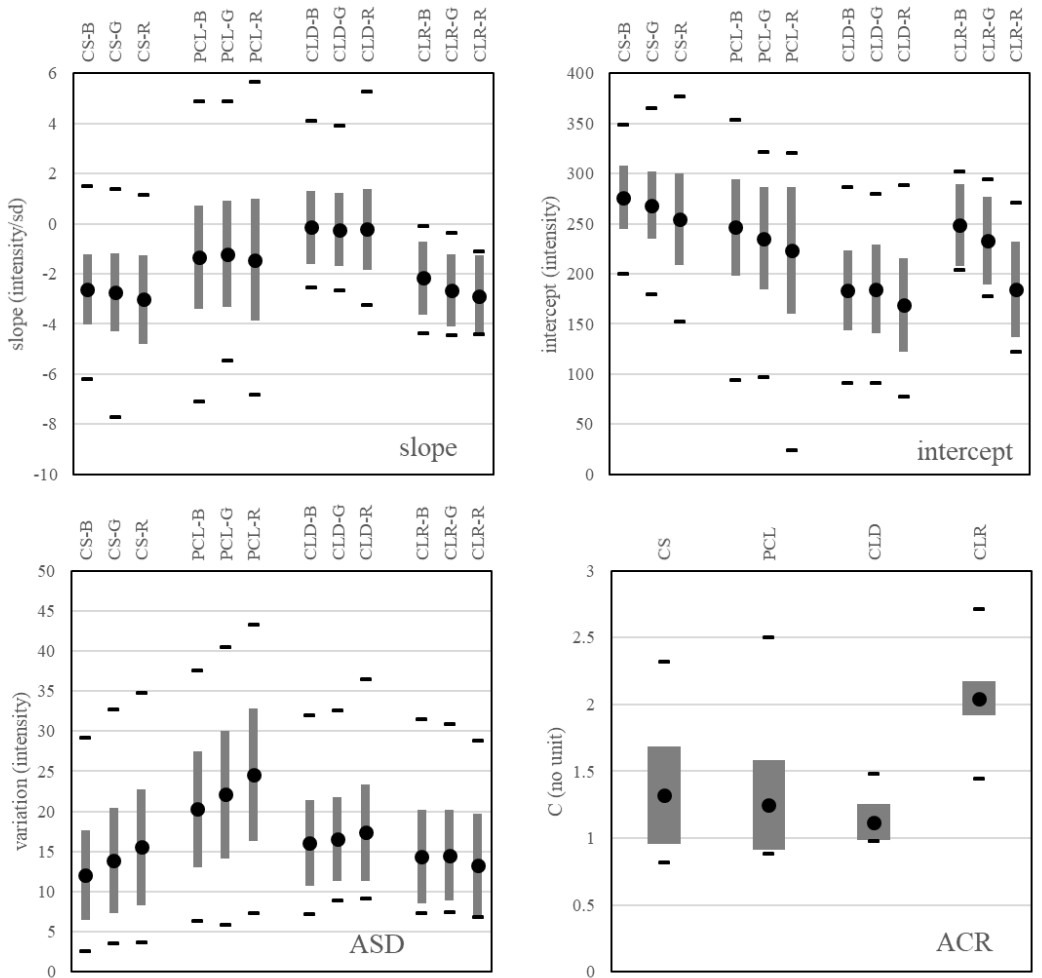

**Figure 5. Photographic sky type properties. Slope and intercept (top row) for the radial fit; areal standard deviation (ASD) of brightness (bottom left); average colour ratio (ACR) (bottom right). Sky types were assigned visually. Black circles indicate the mean, grey boxes the range of the first standard deviation, black bars limit the extreme values found in the master table.**

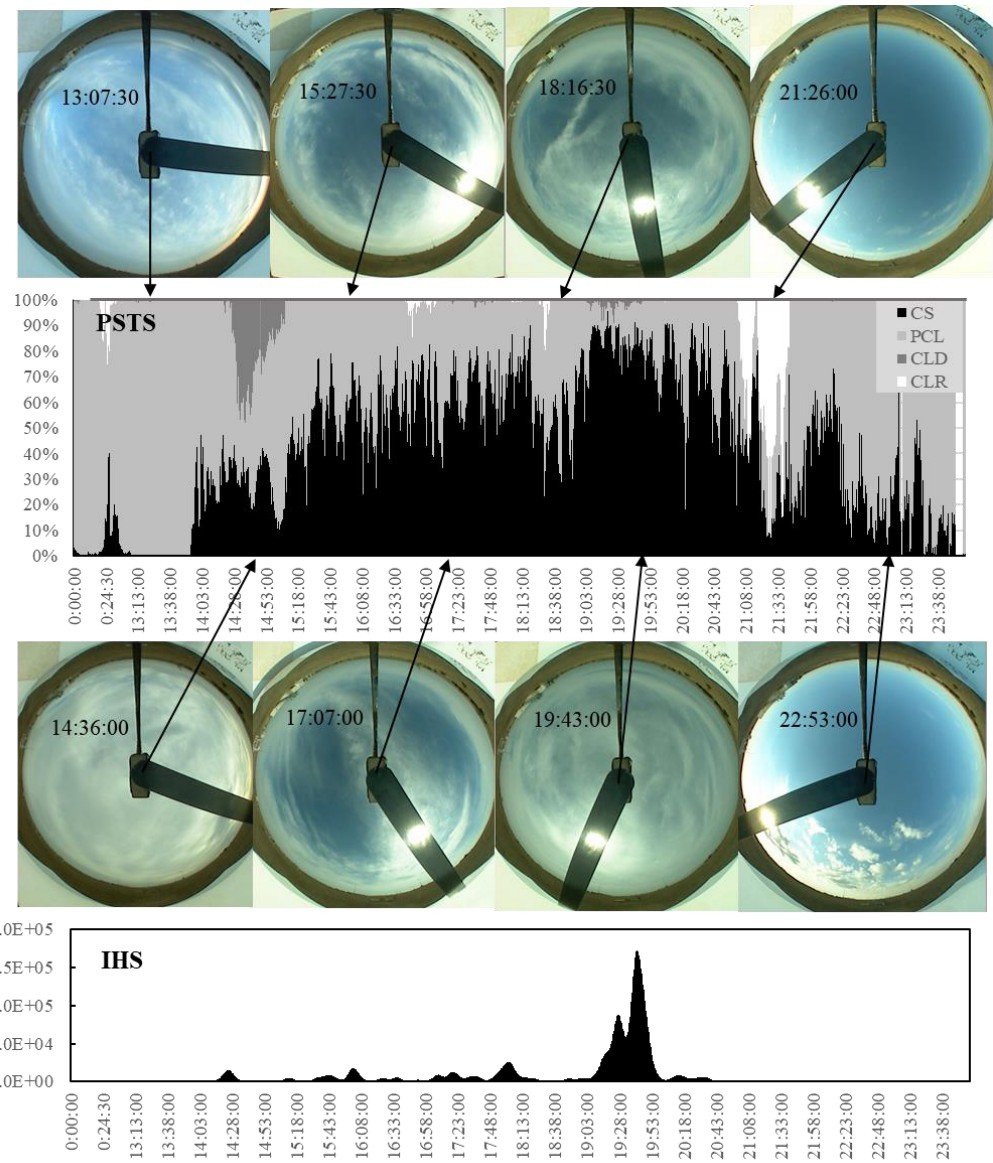

**Figure 6. One-day example for PSTS and IHS (SGP March 10, 2018). Sample TSI images are included. The middle panel shows PSTS versus time of day (N/A excluded). Bottom panel shows the IHS versus time; w=3.5 min. All times in UTC.**

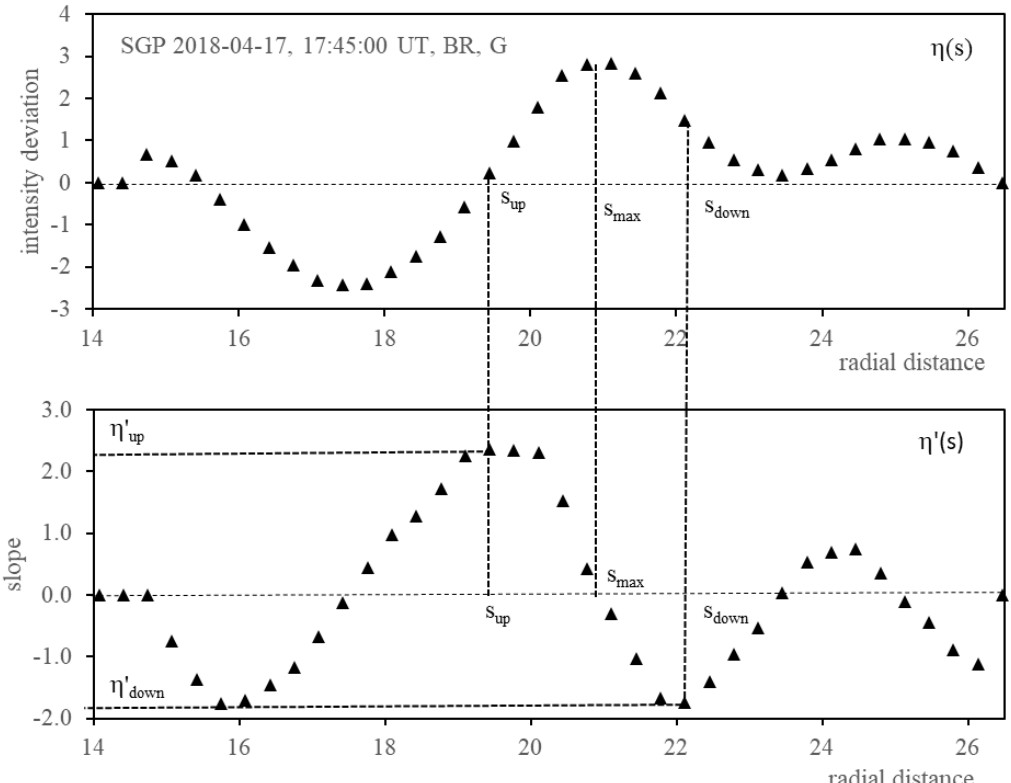

**Figure 7. Radial markers used in halo scoring. The data belong to the green channel of the TSI image from SGP, April 17, 2018, see Fig. 2. The top panel shows the radial intensity deviation $\eta(s)$; the bottom panel shows its derivative $\eta'(s)$. Units are colour value units (0 to 255) for the intensity, and LSM units for the radial distance. The sequence of radial locations used in halo scoring is indicated, as well as the interpretation of the up- and down-slope markers.**

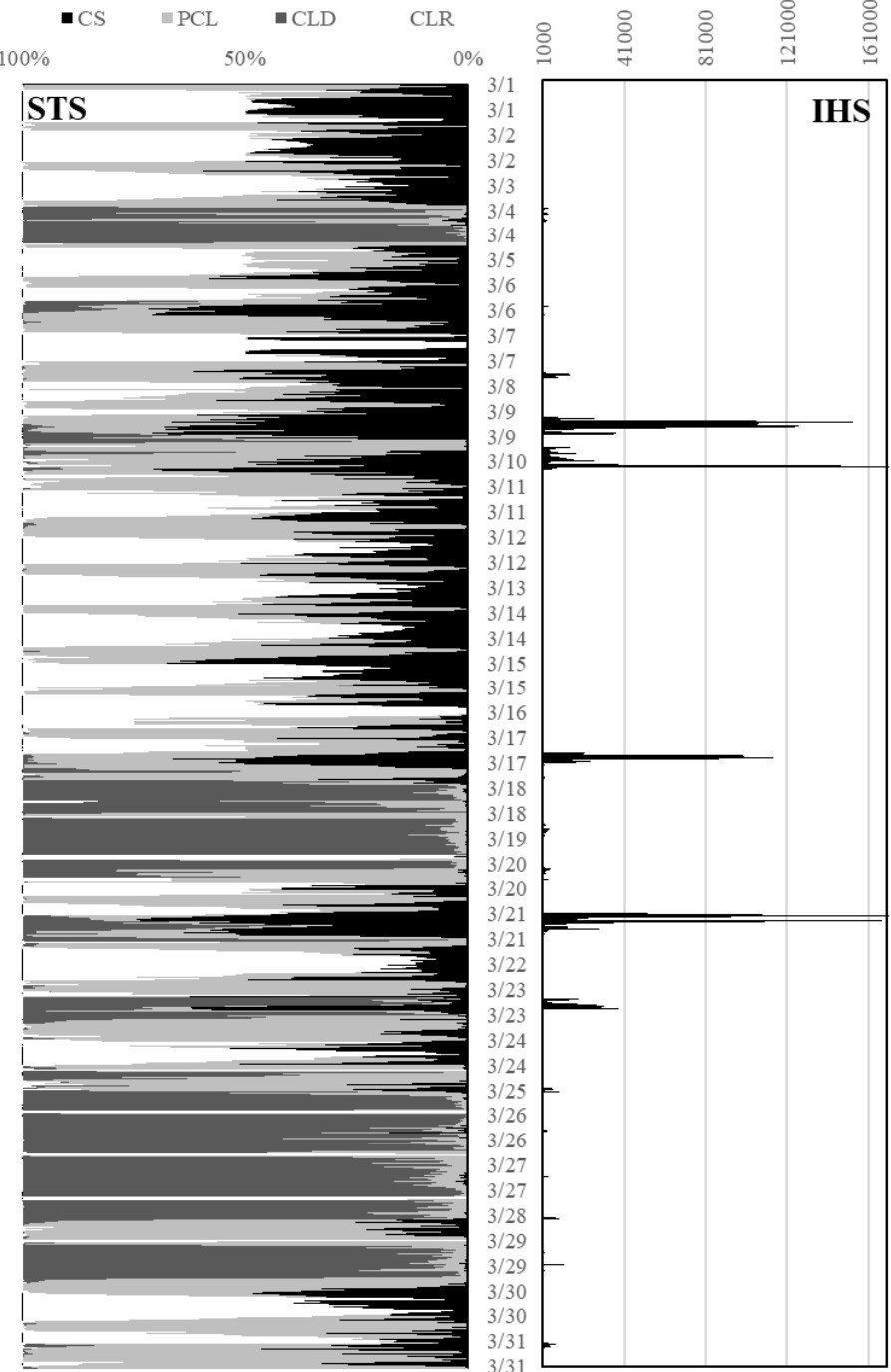

**Figure 8. PSTS and IHS versus time for TSI images from SGP March 2018. Left panel shows the PSTS: PST-CS – black, PST-PCL – light grey, PST-CLD – dark grey, PST-CLR – white. Right panel: IHS broadening w=3.5 minutes.**

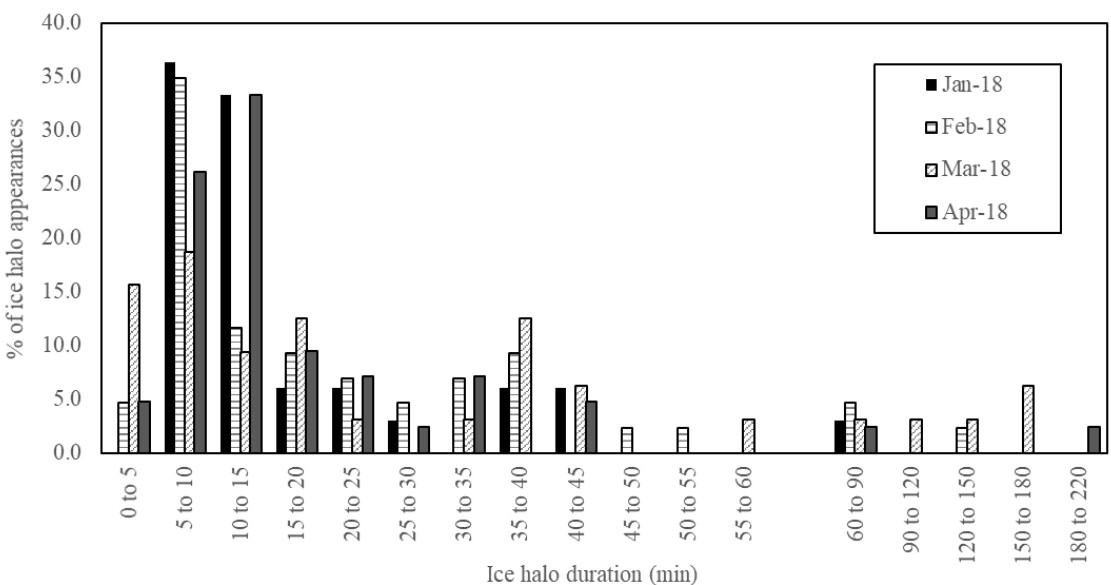

**Figure 9. Distribution of observed 22° halo durations for the first four months of 2018 at SGP ARM site.**