# Peer review of "Analysis Algorithm for Sky Type and Ice Halo Recognition in All-Sky Images"

_Atmospheric Measurement Techniques, 2018_

## Referee Comment (RC1) · Anonymous Referee #1 · 16 Jan 2019

This paper describes an algorithm to infer the type of sky and the presence of halos produced by pristine ice crystals from all sky imagery. In my view, the algorithm is unique and innovative. The paper presents an evaluation of the algorithm for a limited dataset. Its application to more ARM TSI data would provide a very valuable dataset.

The paper is reasonably well written, although there are quite a few technical issues that are listed below. Also, the introduction needs to be improved, as the relevancy of halo detection is largely missing and many of the references are not the most relevant ones.

Below are my comments that need to be addressed in the revised paper before it can be accepted for publication.

**Introduction**

General: The introduction is missing one important motivation for detecting halos. One main reason why detection of halos is relevant is that the pristine crystals that produce them have scattering phase functions with less pronounced backscattering than those of the amorphous or roughened crystals that do not produce halos (Yang et al 2015). As crystal roughness or distortion increase, their phase functions are characterized by decreasing halo features (van Diedenhoven 2014) and decreasing asymmetry parameters (van Diedenhoven et al. 2014; Yang et al 2015). Ice particle surface roughening has a significant influence on the global cloud radiative effect (Yi et al. 2013). Some text along these lines, with the relevant references needs to be included in the introduction.

Page 2, lines 7 and 11: Replace Knobelspiesse et al., 2015 with Waliser et al. (2009).

Page 2, line 12: Here it is stated that "Cloud particle sizes can range from 0.1 microns to a few millimetres (Cziczo and Froyd, 2014)." I think few microns to a few millimetres (or even centimeters) is more realistic. Also, I suggest to replace the reference with Heymsfield et al. (2013).

Page 2, line 12: Replace the reference to Delene, 2011 with Heymsfield et al. (2013)

Line 15: add Hong et al. (2016) to the reference on lidar/radar.

Line 20: I suggest to replace all references here with Bailey and Hallet (2009); Baran (2009) and Yang et al (2015).

Line 21: It is not very clear what is meant with "observable symmetric scattering patterns", but it seems that "halo displays" may be more appropriate. Also note in the sentence that smooth crystal surfaces are needed for halos. Please add references to Um and McFarquhar (2015) and van Diedenhoven (2014)

Line 24: In reference to the "additional ice halo features" cite the book of Tape and Moilanen (2006).

[Figure]

Line 26: Add "forward" before "scattering"

On line 27, discussing the corona, refer to Sassen (1991). Also a more realistic size for corona producing ice crystals is "a few microns".

Line 31: The presentation at the Gordon Research Conference on Radiation and Climate in 2015 cannot be considered a published result, so please remove the reference. In any case, I thought this sentence was very confusing as I thought this was referencing to the results presented in this paper. I suggest to remove this part.

Line 32: The reference to Seefeldner is incorrect and should be Forster et al. 2017.

**Section 1**

Page 4: line 14: I suggest to replace "start the master table" with "train the algorithm", as the master table is not introduced yet.

**Section 2.1**

Page 5, line 12: A value for pre-factor $C_0$ is given, but the X is not defined yet. I suggest to give a value for $C_0$ later.

Page 5: equation 3: Do the absolute values of the elements in x need to be similar so they are weighted equally? Please explain in the text.

**Section 2.2**

Page 5;, line 26: What are the units for BGR? Is that one byte?

Page 6: line 13: Add "to" between "and" and "then".

Page 6 (and elsewhere): Use equation numbers for each equation and expressions throughout the paper.

Page 6: line 5, what is "B" here? In the previous it references to Blue, but the expression is used for all colors. Please clarify in the text.

Page 6: line 7: Remove the brackets and make the text explaining alpha into a proper sentence.

Figure 2: Define RAI, TL, TR, BL, BR in caption.

**Section 2.3**

Page 7: line 6: What are the units of $I(s)$?

Page 7: line 12: Replace "A cloudy sky" with "An overcast sky".

Page 7: line 19: Use italics for "$s$"

Figure 3 (and 6): Please add proper x-axis labels. There is an "s" in the corner. Please spell out "radial distance" and place in the center. I also suggest to add the labels to the bottom of the bottom figures and add a dotted line indicating the zero deviation.

Page 8: line 11: replace CLR with "clear".

Page 8: line 16: It is noted that the "mechanism described in section 2.1" is used. Be explicit about the properties discussed here are inserted in X? Also, I suggest to provide the value of $C_0$ here.

Figure 4: Please add y-axis labels. AST in the caption should be ASD.

Figure 5: Black arrows are used to match the images to the timeline, but these arrows are not visible on the black indicating CS. I propose pointing the arrows of the top images to the top of the timeline plot, so that they are visible.

Page 9: line 30: It is mentioned that "if a radial sequence is found in one colour channel, it should be found in the same locations in all colour channels". Should the angular difference between colors of the halo not be taken into account? The red part of the halo is closer to the sun.

**Section 3**

Page 11, line 23: Some caution is rightfully raised about the visual classification. Some

more information on the method would be helpful to include in the text (here or above at the start of section 3). For example, who was doing the classifications? Is that one person, all of the authors, other people? Also, my guess is that the person (or persons?) evaluating the images are not doing this blindly, so they might already be biased towards the classification of the algorithm.

Page 13, line 3-4: For clarity add "in Table 6" after the "second set of numbers". I suggest to remove the part ", which may be a little easier to interpret".

Page 13: It might be good to discuss the results somewhat more in comparison to Forster et al. 2017.

**References** (please adjust the format to the one used in the manuscript)

Bailey MP, Hallett J (2009) A comprehensive habit diagram for atmospheric ice crystals: confirmation from the laboratory, AIRS II, and other field studies. J Atmos Sci 66:2888–2899. https://doi.org/10.1175/2009JAS2883.1

Baran AJ (2009) A review of the light scattering properties of cirrus. J Quant Spectrosc Radiat Transfer 110:1239–1260. https://doi.org/10.1016/j.jqsrt.2009.02.026

Heymsfield, A. J., C. G. Schmitt, and A. R. Bansemer, 2013: Ice cloud particle size distributions and pressure-dependent terminal velocities from in situ observations at temperatures from 0° to -86°C. Journal of the Atmospheric Sciences, 70, 4123-4154, doi:10.1175/JAS-D-12-0124.1.

Hong, Y., Liu, G., Li, J. -L. F. (2016). Assessing the Radiative Effects of Global Ice Clouds Based on CloudSat and CALIPSO Measurements. Journal Of Climate.

Waliser, D.E., J.-L.F. Li, C.P. Woods, R.T. Austin, J. Bacmeister, J. Chern, A. Del Genio, J.H. Jiang, Z. Kuang, H. Meng, P. Minnis, S. Platnick, W.B. Rossow, G.L. Stephens, S. Sun-Mack, W. Tao, A.M. Tompkins, D.G. Vane, C. Walker, and D. Wu, 2009: Cloud ice: A climate model challenge with signs and expectations of progress. J. Geophys. Res., 114, D00A21, doi:10.1029/2008JD010015.

Sassen, K, "Corona-producing cirrus cloud properties derived from polarization lidar and photographic analyses," Appl. Opt. 30, 3421-3428 (1991)

Tape W. and Moilanen J., (2006) Atmospheric Halos and the Search for Angle X, vol 58, American Geophysical Union, Washington, DC. https://doi.org/10.1029/SP058

Um J, McFarquhar G.M., (2015) Formation of atmospheric halos and applicability of geometric optics for calculating single-scattering properties of hexagonal ice crystals: impacts of aspect ratio and ice crystal size. J Quant Spectrosc Radiat Transfer 165:134–152. https://doi.org/10. 1016/j.jqsrt.2015.07.001

van Diedenhoven B, Ackerman A, Cairns B, Fridlind A (2014) A flexible parameterization for shortwave optical properties of ice crystals. J Atmos Sci 71:1763–1782. https://doi.org/10. 1175/JAS-D-13-0205.1

Yang P, Liou K-N, Bi L, Liu C, Yi B, Baum BA (2015) On the radiative properties of ice clouds: light scattering, remote sensing, and radiation parameterization. Adv Atmos Sci 32:32–63. https://doi.org/10.1007/s00376-014-0011-z

Yi B, Yang P, Baum BA, L'Ecuyer T, Oreopoulos L, Mlawer EJ, Heymsfield AJ, Liou K-N (2013) Influence of ice particle surface roughening on the global cloud radiative effect. J Atmos Sci 70:2794–2807. https://doi.org/10.1175/JAS-D-13-020.1

---

## Author Comment (AC1) · 25 Jan 2019

Author's Response to Comments by Reviewer 1 The authors would like to thank reviewer 1 for the thorough and helpful comments to our manuscript "Analysis Algorithm for Sky Type and Ice Halo Recognition in All-Sky Images". We have incorporated many of the suggestions. The paper has improved significantly thanks to the thorough attention to detail given by reviewer 1. We are much obliged, and extend our gratitude. Below, we outline the details on changes made to the manuscript in response to the comments.

General: The introduction is missing one important motivation for detecting halos. One main reason why detection of halos is relevant is that the pristine crystals that produce

them have scattering phase functions with less pronounced backscattering than those of the amorphous or roughened crystals that do not produce halos (Yang et al 2015). As crystal roughness or distortion increase, their phase functions are characterized by decreasing halo features (van Diedenhoven 2014) and decreasing asymmetry parameters (van Diedenhoven et al. 2014; Yang et al 2015). Ice particle surface roughening has a significant influence on the global cloud radiative effect (Yi et al. 2013). Some text along these lines, with the relevant references needs to be included in the introduction.

This is a very good point to make, thank you! Text was inserted in the Introduction to address this concern (page 2, lines 25 to 35) "As shown in theoretical studies (van Diedenhoven, 2014; Yang et al., 2015), halos form in particular if the ice crystals exhibit smooth surfaces. In that case, the forward scattered intensity is much more pronounced as in cases of rough surfaces, even if a crystal habit is present. If many of the ice particles are amorphous in nature, or did not form under conditions of crystal growth- for example by freezing from super-cooled droplets, or by riming – the forward scattering pattern will be weaker, and similar to what we see for liquid droplets: a white scattering disk surrounding the sun, but no halo. In turn, roughness and asymmetry of ice crystals influence the magnitude of backscattered solar radiation, thus influencing the radiative effect of cirrus clouds (van Diedenhoven, 2016). If the particles in the cirroform cloud are very small, e.g. a few microns (Sassen, 1991), diffraction will lead to a corona. Hence, we believe that a systematic observation of the optical scattering properties adds information to our data on cirrus composition and cirrus radiative properties."

Page 2, lines 7 and 11: Replace Knobelspiesse et al., 2015 with Waliser et al. (2009).

Reference to Waliser et al. (2009) was added in line 7. The reference to Knobelspiesse et al, 2015. had already been changed after the editor's request. In line 11, we replaced the reference to Knobelspiesse et al., 2015 with the suggested reference.

Page 2, line 12: Here it is stated that "Cloud particle sizes can range from 0.1 microns

to a few millimetres (Cziczo and Froyd, 2014)." I think few microns to a few millimetres (or even centimeters) is more realistic. Also, I suggest to replace the reference with Heymsfield et al. (2013).

Correction to "few microns to even centimeter sizes" was incorporated, and a reference to Heymsfield et al (2013) was added. The Cziczo and Froyd article is a review paper, reviewing particle size distributions from various measurements.

Page 2, line 12: Replace the reference to Delene, 2011 with Heymsfield et al. (2013) Correction was incorporated.

Line 15: add Hong et al. (2016) to the reference on lidar/radar. Reference was reviewed and added.

Line 20: I suggest to replace all references here with Bailey and Hallet (2009); Baran (2009) and Yang et al (2015).

The references in line 20 have been corrected as suggested.

Line 21: It is not very clear what is meant with "observable symmetric scattering patterns", but it seems that "halo displays" may be more appropriate. Also note in the sentence that smooth crystal surfaces are needed for halos. Please add references to Um and McFarquhar (2015) and van Diedenhoven (2014)

The sentence has been rephrased to "Only ice particles with a simple crystal habit and smooth surfaces can lead to halo displays", and the suggested references have been added.

Line 24: In reference to the "additional ice halo features" cite the book of Tape and Moilanen (2006).

The citation was inserted.

Line 26: Add "forward" before "scattering"

[Figure]

The word was added to the sentence.

On line 27, discussing the corona, refer to Sassen (1991). Also a more realistic size for corona producing ice crystals is "a few microns".

The wording was changed according to the suggestion. The proposed reference was inserted.

Line 31: The presentation at the Gordon Research Conference on Radiation and Climate in 2015 cannot be considered a published result, so please remove the reference. In any case, I thought this sentence was very confusing as I thought this was referencing to the results presented in this paper. I suggest to remove this part.

The sentence and reference was removed.

Line 32: The reference to Seefeldner is incorrect and should be Forster et al. 2017.

The previous change removed this reference.

Section 1 Page 4: line 14: I suggest to replace "start the master table" with "train the algorithm", as the master table is not introduced yet.

(now line 10) The correction was inserted.

Section 2.1 Page 5, line 12: A value for pre-factor C0 is given, but the X is not defined yet. I suggest to give a value for C0 later.

We replaced the sentence with "The pre-factor C0 in Eqn.(6) is chosen later to place the values for F into a convenient number range." Renumbering the equations resulted in the shift in Eqn number.

Page 5: equation 3: Do the absolute values of the elements in x need to be similar so they are weighted equally? Please explain in the text.

We inserted the following sentence in the hope this would clarify the issue: "For the image properties we chose in STS and IHS computation, the elements of Ximage lie

within one order of magnitude of each other. Hence, no weighing became necessary for this application." Renumbering the equations moved former Eqn (3) to Eqn (6).

Section 2.2 Page 5;, line 26: What are the units for BGR? Is that one byte?

The color values in the jpg have no defined unit, but scale with the receive intensity of the light. We hope to have clarified this by inserting the following statement: "Every pixel in a TSI image exhibits a value between 0 and 255 for each of the three colour channels blue (B), green (G), and red (R). The colour values represent the intensity of the colour channel registered for the particular pixel, varying between 0 (no intensity) and 255 (brightest possible)."

Page 6: line 13: Add "to" between "and" and "then".

Correction was applied.

Page 6 (and elsewhere): Use equation numbers for each equation and expressions throughout the paper.

We renumbered the equations, and included all of them.

Page 6: line 5, what is "B" here? In the previous it references to Blue, but the expression is used for all colors. Please clarify in the text.

Equation (8) was modified to include the tinting procedure as done for every color channel in each pixel.

Page 6: line 7: Remove the brackets and make the text explaining alpha into a proper sentence.

We included this correction: "The coefficient ïĄą regulates the strength of the tinting such that ïĄą=0 leads to no tint, and ïĄą=1 produces an image of a single colour." (now page 6, line 13/14)

Figure 2: Define RAI, TL, TR, BL, BR in caption.

The caption was modified to include "The LSM is divided into four quadrants, named according to their position as TR – top right, BR – bottom right, BL – bottom left, and TL – top left. The RAI is the Radial Analysis Interval for which STS and IHS properties are evaluated."

Section 2.3 Page 7: line 6: What are the units of I(s)?

We added the following sentence to the text: "The term intensity refers to the colour values of any of the colour channels, and varies between 0 and 255."

Page 7: line 12: Replace "A cloudy sky" with "An overcast sky".

Now page 7, line 20: the correction has been made.

Page 7: line 19: Use italics for "s"

This has been corrected after the editorial request, perhaps. I am not sure what it refers to? Figure 3 (and 6): Please add proper x-axis labels. There is an "s" in the corner. Please spell out "radial distance" and place in the center. I also suggest to add the labels to the bottom of the bottom figures and add a dotted line indicating the zero deviation. The suggested changes to the figures have been implemented.

Page 8: line 11: replace CLR with "clear".

Correction was included, now on line 21.

Page 8: line 16: It is noted that the "mechanism described in section 2.1" is used. Be explicit about the properties discussed here are inserted in X? Also, I suggest to provide the value of C0 here.

We hope that the following modification explains the process better (now page 8 lines 27 to 33): "We are using the mechanism described in section Error! Reference source not found., Eqns (1) through (6). The continually refined master table defines a mean value vector M, see Eqn (2), and inverse covariance matrix ïĄŞ-1, see Eqn (4), for each sky type. The mean values for M are given in Error! Reference source not found.,

together with their standard deviations for the training set of images. As a new image is processed, its STS property vector X, Eqn (1), is computed for each sky quadrant. Subsequently, a score is computed for each sky type using Eqn. (6). A value of 105 was used for C0 which places a rough separator of order 1 between images that match closely a particular sky type, and those which do not."

Figure 4: Please add y-axis labels. AST in the caption should be ASD.

Corrections were made as suggested.

Figure 5: Black arrows are used to match the images to the timeline, but these arrows are not visible on the black indicating CS. I propose pointing the arrows of the top images to the top of the timeline plot, so that they are visible.

The adjustments to the figure have been made as suggested, and an error in the caption was corrected.

Page 9: line 30: It is mentioned that "if a radial sequence is found in one colour channel, it should be found in the same locations in all colour channels". Should the angular difference between colors of the halo not be taken into account? The red part of the halo is closer to the sun.

There are multiple factors that influence the inability of the algorithm to resolve the color channels. The first is the small size of the TSI images. Even under good lighting conditions, the angular resolution is limited to 0.3ïĆř, particularly near the horizon. When the perspective distortion is removed, a further reduction in resolution is introduced, again particularly affecting near-horizon solar positions at zenith angles above 45ïĆř. That affects the majority of images. The third influence lies in uncertainties in the solar position. Even though a series of calibrations can address any minor misalignments in North-south line, zenith position, mirror-camera alignment, shadow strip position etc, they are manually assigned, and introduce another uncertainty that affects the fine-angle resolution. The wording was changed to "Lastly, a radial sequence should be

consistent across all three colour channels,..." to avoid potential confusion.

Section 3 Page 11, line 23: Some caution is rightfully raised about the visual classification. Somemore information on the method would be helpful to include in the text (here or above at the start of section 3). For example, who was doing the classifications? Is that one person, all of the authors, other people? Also, my guess is that the person (or persons?) evaluating the images are not doing this blindly, so they might already be biased towards the classification of the algorithm.

The training of the algorithm is a give-and-take, with the goal to maximize agreement. Visual inspection is not perfect, and neither is the algorithm. We modified page 11, lines 8 through 19 to try to address some of the reviewer's concerns: "For each of the 31 days of March, an observer assigned sky classifications to segments of the day by inspecting the day series as an animation. This can easily be done by using an image viewer and continuously scrolling through the series. Then, the day would be subjected to the algorithm. The sections of the record in which visual and algorithm differed were inspected again, at which point either the visual assessment was adjusted, or the misclassified images were included in the Master table in order to train the algorithm toward better recognition. Adjustment to visual classifications often occurred at the fringes of a transition. For example, when a sky transitions from cirrostratus to altostratus to stratus, the transitions are not sharp. The observer sets an image as the point in which the sky moved from CS to CLD, but the criteria in the algorithm would still indicate CS. This can affect up to a hundred images at transition times, which then were reclassified. On the other hand, if a clearly visible halo was missed by the algorithm, this would be a case for adding new property lines to the Master table in order to capture the particular conditions. After each change in the Master table, the algorithm would be repeated, and recalibrations to the visual record, as well as to the Master table itself were made. The process was repeated several times until no more gains in accuracy were observed. These adjustments were done by SB."

Page 13, line 3-4: For clarity add "in Table 6" after the "second set of numbers". I

suggest to remove the part ", which may be a little easier to interpret".

The text has been adjusted accordingly, now page 13, line 29.

Page 13: It might be good to discuss the results somewhat more in comparison to Forster et al. 2017.

Absolutely. Since submitting this paper, long-year analyses of the TSI record have been undertaken, which show some really interesting seasonal variations in the halo appearances in CS skies. We find maximum halo fraction in CS of 20-25% of all CS skies in March and April, consistently through several years of records. We presented that on a poster at the AMS which can be found online if there is interest. This will give a much better basis for comparison than the four-month record included in this paper. We included some more language on page 13 lines 22 to 25 and 30 to address this concern. "in particular with respect to assessments of variation of smooth versus rough crystals. Forster et al (Forster et al., 2017) discuss that the necessary fraction of smooth crystals for a halo appearance lies between 10% and 40%. The authors observe a 22ïĆř halo for 25% of all cirrus clouds for a 2.5-year photographic record taken in Munich, Germany." And "This is certainly consistent with the observations of Forster et al (Forster et al., 2017)."

Please also note the supplement to this comment:
https://www.atmos-meas-tech-discuss.net/amt-2018-401/amt-2018-401-AC1-supplement.pdf

**Supplement:**

[revised manuscript text omitted]

---

## Referee Comment (RC2) · Anonymous Referee #2 · 18 Feb 2019

The value of this study lies, in my opinion, in assessing the feasibility of using TSI images for analysing halo displays to gather information about ice crystal properties. This study is similar to previous studies (e.g. Sassen et al. 2003, Forster et al. 2017), yet holds some valuable contributions since it expands the method to TSI observations, eliminating the need of a sun-tracking camera. However, this work is lacking a discussion comparing the presented method and especially its results to the abovementioned studies. Advantages and limitations of this undoubtedly widely applicable method must be discussed.

**Major remarks:**

1. As visible in Fig. 5, identifying a 22° halo in TSI images might be challenging (even visually) due to a relatively coarse resolution, stray light, and over-exposed image regions. On the upside, TSI cameras are widely used, hence providing a large dataset from several geographic locations which is very attractive for long-term intercomparison studies.

2. The method used to calculate the 22° halo score has considerable overlap with Forster et al. 2017: features are determined in order to discriminate images containing a 22° halo from images which don't. It would be valuable to discuss the slightly differing choice of these features compared to Forster et al. Please discuss the impact of image resolution/FOV of TSI images on the choice of features. The presented study determines the threshold for labelling the images as "22° halo" and "no 22° halo" manually in contrast to Forster et al., who utilized a machine learning method. Please discuss the merits of the different approach used here.

3. How exactly was the algorithm trained? In order to assess its ability to correctly assign the labels "22° halo" and "no 22° halo" as well as the 4 different sky types, it is common practice to test the trained algorithm against *independent* images, which were excluded from the training data. Please describe how exactly this algorithm was tested.

4. Over-exposed image regions are mentioned several times (e.g. P11, L17). Please discuss their impact on the image classification. How was over-exposure treated in general? How would you assess the influence of over-exposed pixels to the detection of 22° halos?

5. How was ensured that parhelia were not accidentally misclassified as 22° halos at low elevations of the sun?

6. The definition and choice of the four sky types should be explained in the text. Halo displays can form in cirrocumulus and optically thick cirrus clouds as well. It is mentioned several times throughout the manuscript that the sky type classification of the images is used to infer information about the "presence of smooth crystalline habits among the cloud particles" (e.g. P13, L6). To answer that question it would be necessary to differentiate between ice clouds and other sky types including clear sky, as in Sassen et al. 2003 and Forster et al. 2017. Thus, the choice of sky types in this study, seems to be not ideal and aims more at differentiating cloud cover ("clear" vs. "cloudy" vs. "partially cloudy", cf. P11, L4). The definition of "cirrostratus" seems to be limited to optically thin, homogeneous cirrus. However, ice clouds and thus halo displays could also be connected with a "cloudy" as well as "partially cloudy" sky type (cf. P9, L10/11) or

even "clear" for very thin cirrus (cf. P11, L26-29). Please re-assess the choice of sky type classes regarding the interpretation of the results.

7. Finally, "long-term data / image records" is mentioned several times (e.g. P1, L6; P3, L19; P13, L13) and Table 1 refers to multiple datasets spanning up to about 7 years of data. I see this as a potential major advantage of this study. However, the study evaluates only images of one ARM site (Southern Great Plains) from January through April 2018. Please describe only the data that was actually used (cf. Tab. 1 and the statement on P2, L30/31). If the algorithm is applicable to large long-time datasets, why wasn't this exploited?

8. The introduction could be tailored more towards implications of unknown cirrus optical and microphysical properties, especially ice crystal shape (and orientation), on the Earth's radiation budget and satellite remote sensing of cirrus clouds. The selection of literature should be revised in this context, with an emphasis on primary literature, especially on the formation and frequency of halo displays (e.g. Minnaert (1937), Tricker (1970), Greenler (1980), Tape (1994), Tape and Moilanen (2006)), as well as ice crystal microphysical properties (e.g. Magono and Lee (1966), Bailey and Hallett (2009)). For example, as a reference for the various ice crystal sizes (P2, L13) and shapes (P2 L20), literature on in situ observations would be more suitable. Delene 2011 and Ewald et al. 2013, for example, don't seem to be the primary literature to support the statement.

**Specific comments on the manuscript:**
**Introduction**
- P2, L16-17: "All of these methods are restricted to a particular time…It is clear that no single method has all the composition information", please clarify. This statement seems to be inherent of any kind of measurement. What exactly should be pointed out here? How do TSI observations solve this problem?
- P2, L24: "More symmetry in the particle orientations will add additional ice halo features." Which additional features? Please cite corresponding literature.
- P3, L11: "The fraction of smooth crystals necessary for ice halo appearance is 10% for columns, and 40% for plates (van Diedenhoven, 2014)". It should be added that these results represent a lower threshold and are based on analysis of scattering phase functions. Therefore, they are not directly applicable to observations of 22° halos in the atmosphere including multiple scattering.
- P3, L12-14: "The consideration of the percentage of cirrus clouds that display optical halo features allows a direct conclusion with respect to the fraction of crystalline habit in the cloud, and, upon further study, about the microphysical conditions in the cloud." The fraction of hexagonal crystals in cirrus clouds cannot be directly inferred from the frequency of visible halo displays. Beside the single scattering properties, which van Diedenhoven 2014 investigated, multiple scattering has to be considered (cf. Forster et al. 2017).
- P3, L28: "refinement of the algorithm goals" is unclear in this context, please elaborate.

**1 TSI images**
- P4, L11: Please provide a range for the angular resolution, specifically for the camera (SGP) and time period used in this study.

**2 Algorithm**
**2.1 Goal and Strategy**
- P5, L7: What is $C_0$ ? → Normalization constant? Explanation in L12 should be moved here.

**2.2 Image preparations and local sky map (LSM)**
- P5, L19: "Some sample steps…" Why not all? Which steps are not shown?
- P5, L26: What is the reason for this the colour balance drift?
- P6, L3: Please specify "from across all TSI records available to the authors". Is this a good reference if the TSI white balance generally drifts? Another possible method would be to calibrate against a white or gray point in the images (e.g. bright cumulus clouds).
- P6, L4: The method how the TSI images were corrected using the scaling factors is not quite clear. It seems like only the blue channel (B) is corrected? How is the normalization of the brightness between 0 and 255 ensured?
- Please define R, G, B. Is this the brightness of the respective color channel?
- P6, L9: "The second step identifies the horizon circle, stretches the visible horizon ellipse…", circle or ellipse? L21 states that the coordinate transformation corrects for deviations from a circle for the 22° halo. And thus also for the horizon?
- P6, L10: "A north-south alignment correction may also have to be applied." Was it applied? If not, the position of the sun and the 22° halo will be shifted. Please discuss.
- P6, L11: "In addition, the horizon is chosen at a zenith angle smaller than 90°, often between 85° and 79°…" How often? Which threshold was used in the other cases? Does it affect the Local Sky Map anyway?
- P6, L19-21: How exactly was the image distortion investigated? Please support this statement by numbers.
- P6, L24: Which "extraneous details" are masked? Please specify.
- P6, L26: What are 40 sky degrees?
- P6, L26: "Units of measurements in the LSM…". Why not simply use pixels? Or zenith and azimuth angles in degrees?
- P6, L28/29: Do you refer to image distortion? "requiring an additional horizontal compression", please explain the procedure. "The algorithm is robust enough to allow this scaling by solar position alone, without loss of efficacy". This should be discussed together with the results.
- Figure 2: Please include a figure showing the LSM as an overlay to the TSI image with 22° halo of Fig. 1 in addition. It would be very helpful to see which portion of the image is actually used for the analysis of the 22° halo when it comes to interpreting the results.

**2.3 Computing Sky Type and Halo Properties**
**2.3.1 Average radial intensity (ARI)**
- P7, L5/6: "We found it useful…", as in previous publications (e.g. Forster et al. 2017). It is indeed practical to use the radial brightness distribution since for randomly oriented ice crystals (causing the 22° halo) the scattering phase function varies only along the scattering angle.
- P7, L14/15: move this explanation of the LSM to section 2.2

- P7, L19/20: How does a radial average over 4 pixels affect the visibility of the 22° halo? Is it necessary? Does the angular resolution of 0.4° to 0.7°, as stated in L25, still hold after averaging?
- P7, L3: Please define "a"
- P7, L24: Please indicate the position of the 22° halo in Fig. 3
- P7, L27: What is 15-26 LSM units in degrees? Where is the 22° halo in terms of LSM units? → Could be visible in an additional figure with an overlay of the LSM onto the TSI image with 22° halo of Fig. 1 (as suggested above)

**2.3.2 Sky type score (STS)**
- P8, L3: Please provide the exact number of images/image segments that were used for training, cf. P10, L22: 44026 images?
- P8, L16: How about introducing the metrics defined in section 2.1 here? In my opinion, the procedure is much easier to understand after the "properties" are explained. In section 2.1 it would be sufficient to explain that a multivariate analysis is performed based on image features/properties. The TSI images are then classified by comparing these features to reference values in a look-up table.
- P8, L17: "continually refined master table" → Please explain this procedure.
- P8, L18/19: As suggested above it would be more convenient for the reader to define Eq. 3 here.
- P8, L19: Please provide the range of values expected for $F_{image}$ (in case of "22° halo" and "no 22° halo"). This might already be interesting to note on P5, L7.
- P8, L25: How was the threshold of $10^{-8}$ chosen? Is it simply outside of the range of F? What kind of images yield this result? → explained later on P12, L1-3. Should be already mentioned here.
- P8, L29: "taken for the combined sky" → "for all 4 LSM quadrants"?
- P9, L9: Please explain the challenges that can be addressed by the "radial scattering analysis" and how

**2.2.3 Ice halo score (IHS)**
- P9, L9: The 22° halo is formed by ice crystals in high-level cirrus clouds. So it is visible wherever cirrus clouds are present and not obstructed by low-level water clouds. The sentence as it stands now gives the wrong impression that the 22° halo is overlaid over low-level clouds. Please correct the sentence accordingly.
- P9, L10/11: The fact that 22° halos are present in images classified as CLD and CLR provides more information about the definition of these categories and the selection of criteria rather than about the formation of the 22° halo. Please adjust the formulation of the sentence to avoid misunderstanding.
- P9, L16: "variations in calibration" The image calibration should not vary across the images. The authors probably want to refer to the north-south mis-alignment of the camera and the coarse angular resolution which can pose a problem in identifying exact position of the 22° halo peak.

- P9, L31: According to theory, the 22° halo peak should not be at the "same" location for the red and blue colour channel, but shifted. Is this feature used for the detection of 22° halos?
- P10, L11: Please define "w" here, instead of L15.
- Figure 5: Please provide values for the IHS at the y-axis of the lower panel
- Is the IHS calculated for each quadrant separately? (How) are they combined to classify the image? → info on P10, L22 should be stated here as well as in section 2.3.2.

**3 Results for January through April 2018**
- P10, L28: The values for C= $10^6$ and w=3.5 (w=4 was defined in L15!) should be mentioned earlier, where the respective equations were defined. Eq. 2 should be Eq. 5?
- P10, L19: It is not be surprising that "high halo scores coincide with strong CS signals", however it can be considered a confirmation that the image features used to train the algorithm were reasonably selected.
- P10, L31 through P11, L2: The determination of a "cut-off" or threshold value "to assign an image with a label of halo/no halo" results from training the algorithm. The same way as the threshold of 50% for the sky type. In both cases the threshold is "arbitrary" to some extent, but should be chosen to minimize either false positive or false negative classifications. This is correctly stated later on P11, L19, but should be mentioned earlier.
- Table 5: The difference between "%vis" and "%alg" is not quite clear. It seems that "%vis" provides an assessment of the visual image classification? This might be confusing for the reader. The interesting quantities here are the fraction of correctly and incorrectly classified images by the algorithm, compared to the ground truth (visual classification). Note that IHS > 4000 in the caption, but IHS > 3500 in the text!
- P11, L4: "A small percentage of visual CLD skies trigger a PCL signal, mostly due to inhomogeneities in cloud cover." Please provide a number for the percentage. Does CLD mean completely overcast? Or do the inhomogeneities here correspond to small clear-sky patches?
- P11, L4/5: Please provide a number stating how successful the classification of CLR is.
- P11, L27: If some CLR images were labeled as "22° halo" why is the fraction of halo instances of CLR all sky type 0% in Tab. 5?
- P11, L23-31: The discussion of the challenges of visual classification of TSI images is very interesting, especially for other publications relying on this. As correctly mentioned, additional Lidar observations together with a temperature threshold e.g. from radiosonde data are useful to improve the classification (cf. Sassen et al. 2003 and Forster et al. 2017). Please add the respective citation also on P13, L7-9 and P13, L30.
- P12, L7: Please explain "various dimensions of the record".
- On P10, L22 it was stated that "An image IHS and STS are assigned as the average over all scoring quadrants." How were the results for the individual quadrants obtained in Tab. 6?
- It should be noted that due to the shadow band a "full 22° halo" actually misses its top and bottom.

**4 Summary**
- P13, L24: 86% vs 85% on P11, L18!

- P13, L27/28: "The algorithm now will be applied to deliver ice halo data for the long-term TSI records accumulated in various geographical locations of ARM sites" Please replace by "In the future, the algorithm will be applied…" to avoid the misunderstanding that this was performed in the present study.

**Please consider the following remarks to further improve the quality of the manuscript:**
The use of technical terms in the manuscript should be revised. In several instances a more commonly used expression exists, which should be used instead where applicable. For example:
- "Ice halo", I would suggest using the term "halo display", which is most commonly used in the literature. Please replace "ice halo" by "22° halo" wherever this specific type of halo display is referred to, e.g. P1, L18 and P12, L10. IHS could be changed to HS22 or simply HS, when it is clear that it is only applied to the 22° halo.
- "look-up table" might be a more commonly use term than "external expandable master table". It is not clear what "expandable" and "external" means in this context? Most tables are expandable.
- P4, L14: Please explain the "master table" and "seed images". Try to use technical terms where possible. The term "master table" is more common in the context of databases. Here the term "look-up table" might be a better choice. A more suitable word for "seed images" would be "training images/data".
- "composition information" (e.g. P2, L17), please specify: does the term refer to microphysical properties, optical properties or cloud phase.
- P1, L14: "standardized", "calibrations": Please describe the specific methods rather than using general terms. "Calibrations" → "colour correction" (this seems to be the only calibration performed).
- P3, L9: "radio probes" → "radiosonde measurements"
- P3, L14: Please replace the term "conditions" by a more specific technical term, e.g. "microphysical properties"
- P3, L24: Please specify "photometric data".
- P3, L29: "effectiveness and types of data" is unclear.
- P4, L5: Please specify "ranges and dates".
- P2, L21: "Only ice particles with a simple crystal habit can lead to observable symmetric scattering patterns". Please specify "simple". Is the message: Ice crystals with a regular hexagonal structure and smooth faces form halo displays?
- P5, L18; P13, L15: "image preparation" → "image processing".
- P5, L18: Please specify "easy-to-use coordinates"→ spherical coordinates? Please specify "minimal colour calibration"→ "colour/white balance correction"
- P5, L26: "colour drift" → better "colour balance"
- P9, L9: "Partial clearings" → "clear-sky regions"
- P10, L22-26 could be summarized as "the algorithm was trained".
- Figure 7 basically shows the "training data set"
- P13, L14: "…information on cirrus composition…" → more specific: "…information on the presence of smooth, hexagonal ice crystals in cirrus clouds from observations of 22° halos…".

**Typos and suggestions for improvement:**

- P3, L19: "We are introducing an algorithm that will read, standardize, and analyse…" Most algorithms read in data and process it in some way before analysing it. Please be more specific or simply say "We are introducing an algorithm to analyse TSI observations regarding the near-solar sky type,…"
- P3, L25: "halo algorithm" -> please use a more descriptive term. "combined and correlated", better: "compared"?
- P4, L6: "…the whole sky **from zenith** to horizon."
- P4, L6: Better: "A sun-tracking shadow band is used to block the sun, which covers a strip of sky from zenith to horizon" (to emphasize that the reason for the shadow band is the sun)
- P4, L8: "JPEG" (acronym for "Joint Photographic Experts Group")
- P4, L12-13: This seems to be a standard routine and can be omitted.
- P4, L27: "The region is centred at the vector of mean values […] ".
- P5, L21: "…the other **one**…"
- P6, L14: "plain" → "plane"
- P6, L15: "…a coordinate transformation is performed to represent the sky in terms of azimuth and zenith angles." → "…a coordinate transformation is performed to represent the image pixels in spherical coordinates"
- P6, L24: "adjustments" → "corrections"
- P7, L5: "radius of 22°" → better: "scattering angle of 22°"
- P7, L5/6: "intensity behaviour" → "brightness".
- P7, L7: "scattering centres" → "scattering particles". "new to the line-of-sight to the sun" and "in the near-sun sky section" can be omitted for clarity. This statement is true for the whole sky.
- P7, L8: "a very fast initial radial decline…followed by a relatively low gradient" → "exponential decline"
- P7, L10: due to Rayleigh scattering.
- P7, L10/11: the increased forward scattering of larger particles (in this case ice crystals) leads to a decreased gradient of the radial brightness
- P8, L2: Consider starting with the explanation of the properties of I(s). It will make the rest of the section much easier to follow.
- P8, L8: "…gives access **to** the overall brightness…"
- P9, L1: "22:53:00 **UTC**"
- P12, L9: "instance**s"**
- P12, L20: "The closer…, **the** more…"
- P13, L22: "raw image score"→ "STS or IHS"
- P11, L5: "CS and PCL are very successful, but exhibit some difficulties." → better: "Differentiating between CS and PCL…"?
- P13, L16: "A multivariate analysis of selected LSM properties, as supported by a continually developed master table, allows the assignment…" → "A multivariate analysis of selected LSM features, stored in a look-up table, allows the assignment…"

---

## Author Comment (AC2) · 20 Mar 2019

Re: amt-2018-401-RC1

**Author's Response to Comments by Reviewer 2**

I would like to thank reviewer 2 for the time taken to read, understand, and thoughtfully comment on our manuscript. The reviewers comments have been very detailed and targeted, and have helped to improve the manuscript significantly. It is the hope of the authors that this means the reviewer does not consider the work presented here as hopeless, but was mostly concerned with the clarity of presentation.

Below, I will address the concerns expressed in the reviewer's comments and indicate where and which actions were taken to improve the manuscript in response. The response follows the outline of RC2 closely, and is organized as required by the editor

(1) comments from Referees,

(2) author's response,

(3) author's changes in manuscript for each applicable comment.

The adjustments in response to reviewer 1 are highlighted in yellow, while the adjustments included for reviewer 2 are highlighted in gray in the attached manuscript.

Thank you again for your time and valuable input.

Sylke Boyd

Begin response:

**Major remarks:**

(1) As visible in Fig. 5, identifying a 22 · halo in TSI images might be challenging (even visually) due to a relatively coarse resolution, stray light, and over-exposed image regions. On the upside, TSI cameras are widely used, hence providing a large dataset from several geographic locations which is very attractive for long-term intercomparison studies.

(2) I do agree that there are limitations to the resolution of TSI images. These limitations are discussed in the manuscript line 10, page 4 (revised according to recommendations from reviewer 2). It is not difficult to visually identify halos in a TSI series if one considers subsequent images in context. For example, cloud features will move, while 22° halos stay stationary with respect to the sun. That is why the algorithm includes the Gaussian time broadening in equation 14. Clearly, clarification is in order. Section 1 has been expanded accordingly, see in the detailed responses below.

(1) The method used to calculate the 22° halo score has considerable overlap with Forster et al. 2017: features are determined in order to discriminate images containing a 22° halo from images which don't. It would be valuable to discuss the slightly differing choice of these features compared to Forster et al. Please discuss the impact of image resolution/FOV of TSI images on the choice of features. The presented study determines the threshold for labelling the images as "22° halo" and "no 22° halo" manually in contrast to Forster et al., who utilized a machine learning method. Please discuss the merits of the different approach used here.

(2) There are multiple differences and similarities.

In common:

- Using the radial intensity, and searching for a series of properties of that radial intensity in order to identify halos. Both methods find use in the sequence of a minimum, followed by a maximum in radial intensity.
- The use of a training set, and evaluation in a set of images not used for training.
- The success rates in identifying 22° halos are similar. Forster algorithm predicts 97.3 +/- 1.9% accuracy of halo in the top segment, 88.5+/- 7.1% accuracy of halo in the bottom segment of the solar surroundings. While we have not inspected the accuracy by sky quadrant, and while our algorithm does not produce a binary decision, the IHS can be used to produce a binary decision if so desired. In section 3, we introduced a decision threshold for the IHS, and we have found that 88% of our algorithm's halo identifications indeed correspond to a visual halo. That is a success rate comparable to Forster. Both algorithms are secure in decisions of "no halo"

Differences:

- Objective: Forster et al constructed a high-resolution camera with precision control for positioning, allowing a much more precise and highly resolved imaging of the sky surrounding the sun. In consequence, the algorithm developed by Forster et al is able to resolve color dispersion in halo display, in addition to specifically search for halo features such as parhelia, upper tangential arcs, etc. Therefore, the outcomes are different. The TSI images targeted in this manuscript do not allow the search for parhelia, nor for color dispersion, due to limitations in resolution. Our objective is to extract not only halo information, but also sky type information.
- Resolution: employing a precision camera allows complete control over angular variables gleaned from the images. This precision in measuring angles is then used in the algorithm of Forster in searching for halo minimum and maximum in precise locations. That is not the case for the TSI algorithm in this manuscript. The limitations of resolution, and the variety of imperfections in alignments and perspective resolution made it necessary to work out a method that is self-consistent in its units, but not necessarily mapping to objective angles.
- Halo characterization process: Forster et al uses a decision tree constructed from random forest classifiers. It is a classification scheme that is correctly characterized as a machine learning algorithm. The criteria governing the final decision tree are derived from a training set of images. The Forster algorithm arrives at a binary classification

halo/no halo. Instead of a decision tree, we employed multivariate analysis and investigated Mahalanobis distances to regions of interest in the space of properties. Defining these regions (means and covariance matrix) is a continued process, which certainly can be called training. The result, however, is not a binary decision, rather a halo score along a continuous scale, which rates how much an image resembles the reference halo images. It is incorrect to characterize this as "labelling the images as "22⬚ halo" and "no 22⬚ halo" manually ".

The continuous halo score is assigned to an image by the algorithm. If, upon further data analysis, one attempts to use it for a yes/no decision then the application of a threshold becomes necessary. But that is not part of the algorithm itself. In section 3, the manuscript discusses applications of the algorithm, together with a thorough test of its capabilities. That is where a decision threshold is introduced.

- The TSI algorithm introduced here analyses sky type in addition to searching for ice halo features.

I would like to add, that the concern for the considerable overlap is justified. It is interesting, that Forster and the authors of this manuscript arrived at the very same points to characterize halos independently. We (SB and MK) met Linda Forster at the 2015 Gordon Conference on Radiation and Climate, introducing these exact two algorithms. The difference in publication date lies in the difference between what can be done in a pure research environment, and what can be done in an institution where research involves only undergraduate students, including the continued and repeated training and time constraints present here. However, that should not have any bearing on the manuscript. I added a paragraph to section 3.

(3)page 16, line1 ff Finally, it is worth discussing the general approach of the TSI algorithm in comparison to the halo detection algorithm developed by Forster et al (Forster et al., 2017). Both algorithms utilize features found in the radial intensity $I(s)$, such as the sequence of minimum – maximum at the expected radial positions in order to find halos in an image. The random forest classifier approach described in (Forster et al., 2017) is a machine learning approach that arrives at a binary conclusion for an image in form of halo/no halo. Their algorithm was trained on a visually classified set of images in order to construct a suitable decision tree. In addition to 22°halos, the Forster algorithm also identifies parhelia and other ice halo features in images taken by a high-resolution, sun-tracking halo camera. The algorithm presented here for TSI data must work with a much less specialized set of images, notably of lower resolution. It does not characterize halos in a binary decision, but rather assigns a continuous ice halo score to an image, in addition to sky type scores for four different types of sky conditions. Similar to the Forster algorithm, the TSI algorithm also was trained on a visually classified set of images. Further training is easy to incorporate via a master table which provides means and covariance matrices to the algorithm. Both algorithm have overlap. The TSI algorithm makes extensive use of the radial brightness gradient (slope) for the sky type assignments. The relation of this gradient to the physical presence of scatterers along the optical path makes this an attractive approach.

(1) How exactly was the algorithm trained? In order to assess its ability to correctly assign the labels "22·halo" and "no 22·halo" as well as the 4 different sky types, it is common practice to test the trained algorithm against *independent* images, which were excluded from the training data. Please describe how exactly this algorithm was tested.

(2) Exactly. That is why we used 80 seed images to define criteria, and to "seed" the master table. This is mentioned in section 1, page 4 line 14, in the sentence "We used eighty seed images taken from across the TSI record and across all available years to train the algorithm (ENA, 2018; NSA, 2018; SGP, 2018). This included images visually identified as CS, PCL, CLD, CLR, and halo-bearing. The seed samples were used to develop the algorithm and define a suitable set of characteristic properties for STS and IHS. " The description of the training method is contained in section 3, page 11, line 8ff. This section was expanded significantly in response to reviewer 1. Here is the new section to address this concern:

(3) Page 11, line13 ff: For each of the 31 days of March, an observer assigned sky classifications to segments of the day by inspecting the day series as an animation. This can easily be done by using an image viewer and continuously scrolling through the series. Then, the day would be subjected to the algorithm. The sections of the record in which visual and algorithm differed were inspected again, at which point either the visual assessment was adjusted, or the misclassified images were included in the Master table in order to train the algorithm toward better recognition. Adjustment to visual classifications often occurred at the fringes of a transition. For example, when a sky transitions from cirrostratus to altostratus to stratus, the transitions are not sharp. The observer sets an image as the point in which the sky moved from CS to CLD, but the criteria in the algorithm would still indicate CS. This can affect up to a hundred images at transition times, which then were reclassified. On the other hand, if a clearly visible halo was missed by the algorithm, this would be a case for adding new property lines to the Master table in order to capture the particular conditions. After each change in the Master table, the algorithm would be repeated, and recalibrations to the visual record, as well as to the Master table itself were made. The process was repeated several times until no more gains in accuracy were observed. These adjustments were done by SB.
* * *
(1) Over-exposed image regions are mentioned several times (e.g. P11, L17). Please discuss their impact on the image classification. How was over-exposure treated in general? How would you assess the influence of over-exposed pixels to the detection of 22·halos?

(2) Table 2 contains a fifth sky classification: N/A. Overexposure is easily identified: if the average radial color values in the analysis region are above a threshold (used 253) in each color channel then the computation of any further properties is compromised. This particular sky quadrant is excluded from skytype assignments as well as halo analysis. It may be worth to remember that the analysis area lies between 15 and 25 LSM units in the local sky map, which excludes the sun itself. As a practical matter, overexposure is a signal produced often in images taken at solar

positions near the horizon (discussed in section 2.3.2, page 9, line 31). It is influenced by the perspective correction, which lowers resolution significantly under these conditions. Perhaps, the inclusion of the following statement can help clarify this in the manuscript.

(3) Pg 9, line 13ff: It simply means that its properties are not close to any of the sky type categories. Also classified as N/A are quadrants in which the average radial intensity lies above 253 (overexposure), or contains a large fraction of horizon (bottom quadrants in low sun positions).
* * *
(1) How was ensured that parhelia were not accidentally misclassified as 22 · halos at low elevations of the sun?

(2) Low sun positions, which are more prone to lead to parhelia appearances, are victim to low resolution due to perspective distortion. The resolution vertically falls below1.2°. I have not been able to visually and reliably discriminate parhelia in any TSI image. An algorithm specifically for parhelia was therefore not attempted. With the separation into quadrants, any existing parhelia would form right on the boundary between top and bottom quadrant , and basically average into the radial intensity of this quadrant. The algorithm does mark N/A for the bottom quadrants at low solar positions, since the local sky map contains mostly horizon for those quadrants. The top quadrants, if not overexposed, may give halo signals. But again – parhelia can not be visually distinguished in those images.
* * *
(1) The definition and choice of the four sky types should be explained in the text. Halo displays can form in cirrocumulus and optically thick cirrus clouds as well. It is mentioned several times throughout the manuscript that the sky type classification of the images is used to infer information about the "presence of smooth crystalline habits among the cloud particles" (e.g. P13, L6). To answer that question it would be necessary to differentiate between ice clouds and other sky types including clear sky, as in Sassen et al. 2003 and Forster et al. 2017. Thus, the choice of sky types in this study, seems to be not ideal and aims more at differentiating cloud cover ("clear" vs. "cloudy" vs. "partially cloudy", cf. P11, L4). The definition of "cirrostratus" seems to be limited to optically thin, homogeneous cirrus. However, ice clouds and thus halo displays could also be connected with a "cloudy" as well as "partially cloudy" sky type (cf. P9, L10/11) or even "clear" for very thin cirrus (cf. P11, L26-29). Please re-assess the choice of sky type classes regarding the interpretation of the results.

(2) The reviewer's comments in this point seem to be concerned with a bias in the halo search, introduced by an assumption of cirrostratus necessary to detect a halo? Proceeding with this assumption.

Please understand that STS and IHS are assigned independently from each other, and use differing sets of criteria. Even if some of the criteria are similar, the master training yields different averages and covariances. The slopes for CS and for IHS are numerically different (and

all other common properties are as well) as can be seen from comparison of tables 3 (STS properties) and 4 (IHS properties). As a matter of fact, we do find ice halos in both PCL and CLD skies, as explained in section 3, page 14, lines 25ff. Section 2.3.2 describes the selection of sky types to be scored. Since we focus on the areas of the sky in which a 22° halo would be located, the sky type really only characterizes the radial analysis area indicated in Figure 2, not the complete visual sky. The beginning of section 2.3.1 explicitly explains why the radial intensity is the key for the chosen sky types, in that the average gradient contains a measure of number and density of scatterers in that sky area.

I would like to add, that the reviewer statement

"It is mentioned several times throughout the manuscript that the sky type classification of the images is used to infer information about the "presence of smooth crystalline habits among the cloud particles" (e.g. P13, L6)."

indicates a misunderstanding. The whole sentence (now pg15 line 25) reads

"One of the conclusion to be made from the relation between STS and IHS concerns the confidence in the presence of smooth crystalline habits among the cloud particles, as shown only in a one-fifth fraction of all cirrostratus."

It is not the STS alone that can give information about the type of ice crystals, but the combination of STS and IHS. That is an important distinction. It means that if cirrostratus is present, we can conclude with confidence that halo-generating habits are present if a halo is detected, and that occurs for about 20% of all cirrostratus occurrences in the data set analyzed in this manuscript.
* * *
(1) Finally, "long-term data / image records" is mentioned several times (e.g. P1, L6; P3, L19; P13, L13) and Table 1 refers to multiple datasets spanning up to about 7 years of data. I see this as a potential major advantage of this study. However, the study evaluates only images of one ARM site (Southern Great Plains) from January through April 2018. Please describe only the data that was actually used (cf. Tab. 1 and the statement on P2, L30/31). If the algorithm is applicable to large long-time datasets, why wasn't this exploited?

(2) The reviewer is correct. The longest data set is actually 18 years long. The manuscript under review here is a method paper, in which an algorithm is introduced in detail, together with reasonably long test data to demonstrate effectiveness and limitations. This is laid out in the Introduction, now page 3, line23ff.

The explicit analysis of the long-term records in different geographical locations will require a separate paper, to be expanded and supported by LIDAR data. The summary has language to that strategy. In addition, this work is done at an undergraduate institution, with undergraduate research assistants, including all the time and training restrictions this brings. Data collection and analysis will take time. Some of the additional findings have already been published at the 99[th] AMS meeting (https://ams.confex.com/ams/2019Annual/meetingapp.cgi/Paper/351343 ), and more is going to be ready by the end of summer 2019.

The long-term data records are included in this current manuscript since the set of 80 seed images is taken evenly from all three locations and different times. Section 2 already discusses this. I changed the table caption to include this fact as well, hoping to improve clarity.

(3) **Table 1. TSI data set properties. Seed images for the algorithm were taken from all three locations.**

(1) The introduction could be tailored more towards implications of unknown cirrus optical and microphysical properties, especially ice crystal shape (and orientation), on the Earth's radiation budget and satellite remote sensing of cirrus clouds. The selection of literature should be revised in this context, with an emphasis on primary literature, especially on the formation and frequency of halo displays (e.g. Minnaert (1937), Tricker (1970), Greenler (1980), Tape (1994), Tape and Moilanen (2006)), as well as ice crystal microphysical properties (e.g. Magono and Lee (1966), Bailey and Hallett (2009)). For example, as a reference for the various ice crystal sizes (P2, L13) and shapes (P2 L20), literature on in situ observations would be more suitable. Delene 2011 and Ewald et al. 2013, for example, don't seem to be the primary literature to support the statement.

(2) Reviewer 1 had similar comments about the introduction. In response, the set of references as well as the wording has been revised considerably. I hope this also addresses some of the concerns expressed by reviewer 2.That

**Specific comments on the manuscript:**
**Introduction**

(1) P2, L16-17: "All of these methods are restricted to a particular time…It is clear that no single method has all the composition information", please clarify. This statement seems to be inherent of any kind of measurement. What exactly should be pointed out here? How do TSI observations solve this problem?

(2) Great point about the measurements. What should be said here is that despite the existence of so many approaches to cirrus measurements, the composition information has gaps. TSI analysis for ice halos may provide another support that can fill some of these gaps. I replaced the sentence with this:

(3) Pg 2 ln 16ff: Even combined, these methods leave gaps in our knowledge of spatial and temporal composition of ice clouds. The analysis of ice halos as captured by long-term total sky imagers may provide further insight and allow to close some of the gaps.

(1) P2, L24: "More symmetry in the particle orientations will add additional ice halofeatures." Which additional features? Please cite corresponding literature.

(2) Information and reference was added.

(3) Pg 2, ln 25: More symmetry in the particle orientations will add additional ice halo features such as parhelia, upper tangent arc, circumscribed halo, and others (Greenler, 1980; Tape and Moilanen, 2006)
* * *
(1) P3, L11: "The fraction of smooth crystals necessary for ice halo appearance is 10% for columns, and 40% for plates (van Diedenhoven, 2014)". It should be added that these results represent a lower threshold and are based on analysis of scattering phase functions. Therefore, they are not directly applicable to observations of 22 · halos in the atmosphere including multiple scattering.

(2) Language to this extend has been included. I would like to add that the prevalence of multiple scattering will generally lead to a dissolution of the halo display. We see that in thickening altostratus clouds, when a halo perhaps still shows in some images, but it is "washed" out by increasing attenuation. So, at the very least one can conclude two things from a halo display: (a) a minimum of 10-40% smooth enough crystalline ice particles, and (b) dominance of single scattering. Unfortunately, I do not think that the latter has been quantified yet.

(3) Pg 3, 14ff: The fraction of smooth crystals necessary for ice halo appearance is at a minimum10% for columns, and 40% for plates, based on an analysis of scattering phase functions for single scattering events (van Diedenhoven, 2014). While this establishes a lower boundary, it is correct to say that the observability of an ice halo allows to conclude that smooth crystalline ice particles are present and single-scattering events dominate. The consideration of the percentage of cirrus clouds that display optical halo features allows therefore, upon further study, inferences about the microphysical conditions in the cloud.
* * *
(1) P3, L12-14: "The consideration of the percentage of cirrus clouds that display optical halo features allows a direct conclusion with respect to the fraction of crystalline habit in the cloud, and, upon further study, about the microphysical conditions in the cloud." The fraction of hexagonal crystals in cirrus clouds cannot be directly inferred from the frequency of visible halo displays. Beside the single scattering properties, which van Diedenhoven 2014 investigated, multiple scattering has to be considered (cf. Forster et al. 2017).

(2) Addressed in previous.

(1) P3, L28: "refinement of the algorithm goals" is unclear in this context, please elaborate.

(2) Changed wording.

(3) Pg 3, ln 32: Section **Error! Reference source not found.** presents the details of the image analysis algorithm, including subsections on algorithm goals, image preparation, and sky type and halo scoring.

**1 TSI images □**

(1) P4, L11: Please provide a range for the angular resolution, specifically for the camera (SGP) and time period used in this study.

(2) An image from SGP in 2018 has a size of 488 by 640 pixels. The short dimension limits the radius of the view circle, let's say it is 240 pixels. A pixel close to the center corresponds to an angular sky section 2.8º wide and 0.24º tall. In the TSI series analyzed here, the solar position never reaches this point. Close to the horizon, one pixels averages a sky section that is 0.24º wide and 1.24º tall. Best resolution is achieved at zenith angle 45º, in which case every pixel represents a sky region of 0.33º by 0.33º. Language indicating this has been included in section 1.

(3) Pg 4, ln 14ff:  For example, an image from SGP taken in 2018 has a size of 488 by 640 pixels. The short dimension limits the radius of the view circle to at most 240 pixels. A pixel close to the center of the view circle corresponds to an angular sky section 2.8º wide and 0.24º tall. At SGP, the solar position never reaches this point. Close to the horizon, one pixels averages a sky section that is 0.24º wide and 1.24º tall. Best resolution is achieved at zenith angle 45º, in which case every pixel represents a sky region of 0.33º by 0.33º. The image distortion is largest for sky segments close to the horizon due to perspective distortions in the mirror image of the sky.

**2 Algorithm**

**2.1 Goal and Strategy**

(1) P5, L7: What is C0 ? Normalization constant? Explanation in L12 should be moved here.

(2) Reviewer 1 had a similar comment. In response, the following wording has been applied:

(3) Pg 5 ln 27: … improvement of scoring. The pre-factor $C_0$ in Equation (6) is chosen later to place the values for F into a convenient number range. This basic alg..
* * *
**2.2 Image preparations and local sky map (LSM)**

(2) This particular section seems to be written in a confusing manner, as I conclude from the many questions targeting similar issues. I have re-organized it to first list the calibration steps in order and then describe the details of each of the steps.

(3) pg 6 ln2 ff: The image preparations include the following steps: (1) a colour correction, (2) an alignment calibration, (3) a removal of the perspective distortion, (4) masking and marking of the solar position, and (5) rotation and crop to create a Local Sky Map (LSM). Some sample steps in the image preparation are illustrated in **Error! Reference source not found.**. The figure includes the original image, the image after step 4, and the LSM after step 5.

(1) P5, L19: "Some sample steps…" Why not all? Which steps are not shown?
(2) Addressed as highlighted above.
* * *
(1) P5, L26: What is the reason for this the colour balance drift?

(2) The color response of every camera sensor is different, even for cameras of exactly the same type. No two TSI cameras report exactly the same color values for clear blue sky, for example. In addition, camera sensors age with use. Most CCD and CMOS sensors will change their sensitivity over time, perhaps due to defect accumulation as they are exposed hundreds of thousands of times, sometimes to extreme sun (shadow-strip malfunctions do occur and are present in the TSI record), and as the cameras exist in extreme temperature conditions. Revised the statement and inserted reference.

(3) Pg 6, ln 13: Since the algorithm is intended for multiple TSI locations and records taken over long time, including device changes, it is necessary to consider the fact that no two camera devices have exactly the same colour response, even if of same type (Ilie and Welch, 2005).

(1) P6, L3: Please specify "from across all TSI records available to the authors". Is this a good reference if the TSI white balance generally drifts? Another possible method would be to calibrate against a white or gray point in the images (e.g. bright cumulus clouds).

(2) I changed the sentence for clarity. Clear sky near zenith is a good reference. The drift is very slow, and a check-back is needed less than once a year, unless the actual device changed. A device change is a bigger influence on colour changes than the aging of the sensors, although these also influence the tinting. One can not colour-check against white, since the relationships between the different colour channels gets lost if all channels are saturated. Gray values exist in nature in so many different contexts and variations that finding a standard is not easily possible. However, a clear high-pressure sky, near zenith and not close to the sun itself provides a reproducible reference.

(3) Pg 6, ln 22ff: The reference values are based on colour values for clear sky images from the TSI records listed in Table 1. Near-zenith, clear blue sky provides a reproducible colour reference in all the locations.
* * *
(1) P6, L4: The method how the TSI images were corrected using the scaling factors is not quite clear. It seems like only the blue channel (B) is corrected? How is the normalization of the brightness between 0 and 255 ensured?

(2) The paragraph discussing the tint adjustment has experienced editing, based on both reviewer's comments, and reads now:

(3) Once these colour-scaling factors are determined for a series, every image was then tinted by generating an average colour $(\bar{B}, \bar{G}, \bar{R})$ for a small near-zenith sky-sample and applying

$$B' = \left[B + \alpha\left(\beta_B\bar{B} - B\right)\right]$$
$$G' = \left[G + \alpha\left(\beta_G\bar{G} - G\right)\right]$$
$$R' = \left[B + \alpha\left(\beta_R\bar{R} - R\right)\right]$$

(4)          (8)

to each colour channel and pixel, respectively, followed by a simple scaling to preserve the total brightness of the pixel $I = \sqrt{B^2 + G^2 + R^2}$. For the series SGP 2018, these factors were β = (0.9, 0.78, 1) and $\alpha$=0.4. The coefficient $\alpha$ regulates the strength of the tinting such that $\alpha$=0 leads to no tint, and $\alpha$=1 produces an image of a single colour. This tinting is minimal, and linear colour behaviour is a reasonable assumption.
* * *
(1) Please define R, G, B. Is this the brightness of the respective color channel?

(2) Correct. Language to this extend was inserted on page 6 line 15.

(3) Pg 6, line 15: Every pixel in a TSI image exhibits a value between 0 and 255 for each of the three colour channels blue (B), green (G), and red (R).  The colour values represent the intensity of the
* * *
(1) P6, L9: "The second step identifies the horizon circle, stretches the visible horizon ellipse…", circle or ellipse? L21 states that the coordinate transformation corrects for deviations from a circle for the 22⬚ halo. And thus also for the horizon?

(2) While the mirror is circular, a slight misalignment of mirror and camera axis can make the circle appear stretched into an ellipse. This is detrimental to the plan to identify the solar position from time and coordinates, and must be corrected. The paragraph for step (2) was reworded thus:

(3) Pg 7 line 2 ff: Step (2) is a stretch-and-shift process that identifies the horizon circle. Occasionally, a slight misalignment of camera and mirror axis leads to an elliptical appearance of the sky image. A calibration is necessary in such cases to stretch the visible horizon ellipse to circular shape, and to centre the horizon circle as close to the zenith as possible. A north-south alignment correction may also have to be applied. Both calibrations will ensure successful identification of the solar position in the next step. These calibrations become necessary if the TSI was not perfectly aligned in the field and need to be readjusted after any disturbances occurred to the instrument, such as storms, snow, instrument maintenance, etc. Typically, this can be once every few months, or sometimes several times per month. It is important to check the calibrations regularly by sampling whether the solar position was correctly identified after calibration.
* * *
(1) P6, L10: "A north-south alignment correction may also have to be applied." Was it applied? If not, the position of the sun and the 22⬚ halo will be shifted. Please discuss.

(2) That was explained in the previous comment and revision. If an alignment was necessary it was applied. If it was not necessary, then it was not applied. It changes, as described above, sometimes weekly, sometimes only once or twice a year. It has no bearing on the halo and sun position if done correctly. These operations only serve to provide a good reference frame to determine the sun position in the images.
* * *
(1) P6, L11: "In addition, the horizon is chosen at a zenith angle smaller than 90⬚, often between 85⬚ and 79⬚…" How often? Which threshold was used in the other cases? Does it affect the Local Sky Map anyway?

(2) These are details that have no bearing on the construction of the LSM, only on the position of the horizon circle, outside of which everything will be masked. As described above in answer to the resolution question, a pixel located in a TSI image close to the horizon (zenith angle 79⁰) covers a radial angular extend of more than 1.2 degree worth of zenith angle. The vertical resolution is so low that there is basically no informational value along the rim of the horizon. Blocking out these pixels does not influence the working of the algorithm. The text in the paragraph has been changed to:

(3) Pg 7, ln 10: In addition, the horizon circle is placed at a zenith angle smaller than 90°, often between 85° and 79°, to eliminate the strong view distortion close to the horizon, and in some cases, objects present in the view. As explained earlier, the zenith angle resolution per pixel exceeds 1.2° close to the horizon. The information value for zenith angles larger than 80° is diminished. These pixels should be excluded from the analysis. Practically, this is a very thin ring cut from the original image but does help eliminate false signals from low sun angles. The current process requires to find these calibrations for a  sampling of images in a series, and to then apply them to all images in the series.
* * *
(1) P6, L19-21: How exactly was the image distortion investigated? Please support this statement by numbers.

(2) While indeed, the influence of the mirror on perspective was tested, inserting a section on optical transformations on a spherical mirrors, and the respective numerical solution, plus conclusion that it does not have an influence … it just seems excessive. Instead, we'll remove the reference to the mirror distortion, and refer to previous publications (Long et al) on perspective treatment in TSI images.

(3) Page 7, line17ff: Step (3) removes the perspective distortion. The projection of the sky onto the plain of an image introduces a perspective distortion, as described in Long et al (Long et al., 2006). A coordinate transformation is performed to represent the sky in terms of azimuth and zenith angles. The azimuth is the same in both projections. Zenith angle θ relates to the radial distance $r$ in the original image from the centre of the horizon circle as $r = R \sin \theta$. While R is not determined, image horizon radius $R_H$ and horizon zenith angle $\theta_H$ provide one known point to allow for proportional scaling. The coordinate transformation represents the sky circle in a way in which radial distance from zenith $s_z$ scales with zenith angular coordinate θ as

$$s_z = \frac{R_H}{\sin \theta_H} \times \theta$$

 One of the visible effects of this transformation concerns 22° halos: in the original TSI image, a halo appears as a horizontal ellipse; after the transformation it will have a shape closer to a circle.
* * *
(1) P6, L24: Which "extraneous details" are masked? Please specify.
(2) As below:

(3) Pg 7, ln 22: Extraneous details, such as the shadow strip, the area outside the horizon circle, the camera, and the camera mount, are masked.
* * *
(1) P6, L26: What are 40 sky degrees?

P6, L26: "Units of measurements in the LSM…". Why not simply use pixels? Or zenith and azimuth angles in degrees?

P6, L28/29: Do you refer to image distortion? "requiring an additional horizontal compression", please explain the procedure. "The algorithm is robust enough to allow this scaling by solar position alone, without loss of efficacy". This should be discussed together with the results.

(2) I will address these three comments together, since all of them relate to a fundamental misunderstanding of what LSM units are. I hope that the rewrite does help clarify the questions.

(3) Pg 7, 31ff In step (5), the standardized local sky map (LSM) is created. A sketch of the layout of the LSM is provided in Figure 1. The LSM provides a standard sky section, centred at the sun, oriented with the horizon at the bottom, and presented in the same units for all possible TSI images (independent on the resolution of the original). Units of measurement in the LSM are closely related to angular degrees, but do not match perfectly due to a zenith-angle dependence of the azimuth arc length. The LSM is generated by rotating and cropping the image from step (4) to approximately within 40° of the sun, with the sun at its centre. The side length of the LSM in pixels scales with the previously determined horizon radius $R_H$ in pixels and the corresponding maximum zenith angle $\theta_H$ in ° as

$$w_{LSM}(pixels) = \frac{R_H(pixels)}{\theta_H(degrees)} \times 40°$$

(9)

Equation (9) provides a unit transformation between pixel positions and LSM units. For a TSI image of size 480×640 pixels, the LSM will have a size of approximately 240×240 pixels. For the earlier, smaller TSI images, the LSM has a size of approximately 140×140 pixels. The unit scaling includes the calibration choices $R_H$ and $\theta_H$, hence there is a slight variation in LSM pixel sizes. We eliminate the influence of the varying pixel sizes by performing all algorithm operations in standardized LSM units, which roughly correspond to angles of 1°. In other words, all LSM are equivalent to each other in terms of their LSM units, but not in terms of pixel positions. At θ=45°, the arc length of azimuth angle φ is equivalent to the arc length of θ of same size; however, if θ>45° the azimuth arc is stretched, requiring an additional horizontal compression to ensure equivalence of horizontal and vertical angular units.  The LSM is divided into quadrants, shown in Figure 2, which are analysed and classified separately by the algorithm described in the next section.
* * *
(1) Figure 2: Please include a figure showing the LSM as an overlay to the TSI image with 22 · halo of Fig. 1 in addition. It would be very helpful to see which portion of the image is actually used for the analysis of the 22 · halo when it comes to interpreting the results.

(2) In fact, the LSM in figure 2 IS an overlay of the exact thing proposed here. The purpose of figure 2 IS to show which portions of the image are used for analysis, and to support the definition of the variables used. The caption was edited in response to reviewer 1. The halo appears at 21

LSM units, as can be ascertained from the text as well as from the data in figure 3. I included a sketch of the halo in the figure in the hope this makes this easier to understand.

(3)

[Figure]

**Figure 1. Layout of the local sky map (LSM). The LSM is divided into four quadrants, named according to their position as TR – top right, BR – bottom right, BL – bottom left, and TL – top left. The RAI is the Radial Analysis Interval for which STS and IHS properties are evaluated. The approximate position of the halo maximum is sketched in light grey. Shadow strip and camera are excluded from analysis.**

**2.3 Computing Sky Type and Halo Properties**
**2.3.1 Average radial intensity (ARI)**

(1) · P7, L5/6: "We found it useful…", as in previous publications (e.g. Forster et al. 2017). It is indeed practical to use the radial brightness distribution since for randomly oriented ice crystals (causing the 22· halo) the scattering phase function varies only along the scattering angle.

(2) I met Linda Forster at the Gordon Conference on Radiation and Climate in 2015. We both presented posters on halo identification in images, and both independently have used the radial brightness gradient as access. I mention this since the reviewer repeatedly appears to allude that this approach must be referenced to Forster et al. while in truth it is a common-sense approach, independently used in our algorithms. This comment requires no change in the text at this position. Reference to the work of Forster has been made in various other locations throughout the manuscript.

(1) P7, L14/15: move this explanation of the LSM to section 2.2

(2) The comment "The LSM is divided into four quadrants: TR = top right, BR = bottom right, BL = bottom left, TL = top left, analysed separately, and then recombined for the image scores." Has now moved to pg 8 line 31. Since language referring to the quadrants has been inserted into section2, I will leave this in place to allow the discussion that follows in the text.

(1) P7, L19/20: How does a radial average over 4 pixels affect the visibility of the 22
   · halo? Is it necessary? Does the angular resolution of 0.4 · to 0.7 · , as stated
   in L25, still hold after averaging?

(2) This is an excellent question, and answering it helps to improve the text. Earlier, the text was
   expanded to include more detailed information on the resolution. The resolution varies across
   the sky, with 0.4° close to zenith angles of 45°, but only 1.3° close to zenith in declination, and
   1.3° in azimuth close to horizon. The averaging mostly addresses the noise of averaging a circle
   in a coarse square grid. By allowing a band instead of a sharp line, a continuous circular band is
   averaged, instead of a broken series of squares (pixels) that align somewhere close to a perfect
   circular line. The averaging does not diminish the appearance of the halo signal (see figure 3,
   panel B on the left). However, it improves the smoothness of the curve $\eta(s)$ which in turn
   makes it easier to write an algorithm to find maxima, minima, slopes along an imperfect series
   of data. Perhaps, the insertion of the following line provides clarification:

(3) Pg 9, ln 4 ff: ..Due to the low resolution of the LSM, and due to some noise in the data, we average $I(s)$
   over a circular ribbon with a width of 4 pixels, centred at $s$. Computing $I(s)$ over a thin ribbon addresses
   issues encountered when averaging over a circle in a coarse square grid, allowing continuity where
   otherwise pixilation may interrupt the line of the circle. Figure 2 shows..
* * *
(1) P7, L3: Please define "a"
(2) I changed the presentation of the equation for the running average instead. The "a" is not a
   parameter of consequence, only indexes the terms included in the running average.
(3) Pg 9 , ln 10   $\overline{I_6}(s) = \frac{1}{N}\sum_{s-3LSMunits}^{s+3LSMunits} I(s)$

(10)
* * *
(1) P7, L24: Please indicate the position of the 22 · halo in Fig. 3 · P7, L27:

(2) Change in figure and caption for figure 3 was done.

(3) **Figure 2 Average radial intensity of the red channel is shown versus radial distance s, measured in LSM units,
   for the two images of** Error! Reference source not found.**, halo at left. Panel (A) includes the average intensity $I(s)$,
   a linear fit, and the running average $\overline{I}_6(s)$ as averaged over a width of 6 LSM units. (B) shows the radial
   intensity deviation $\eta(s)$. The halo signal is visible as a minimum at 17 LSM units, followed by a maximum at 21
   LSM units in the left column.**

(1) What is 15-26 LSM units in degrees? Where is the 22˙ halo in terms of LSM units? ˙ Could be visible in an additional figure with an overlay of the LSM onto the TSI image with 22˙ halo of Fig. 1 (as suggested above)

(2) As addressed above. LSM units are roughly equivalent to sky degrees, but not perfectly. Halos appear between 21 and 22 LSM units. Colours can not be resolved.

**2.3.2 Sky type score (STS)**

(1) P8, L3: Please provide the exact number of images/image segments that were used for training, cf. P10, L22: 44026 images?

(2) This information is already contained in the text of the section. The formulation of STS properties is based on 80 seed images.

The number the reviewer cites refers to the number of records contained in SGP March 2018, later used to test and train the algorithm. However, this particular position in the text does not address this later testing and training.

(3) Properties of *I(s)* were computed for the set of 80 seed images mentioned in section **Error! Reference source not found.**. Twenty images for each sky type were divided further by sky quadrants, yielding between 60 and 80 property sets for each sky type to seed the master table. Some quadrants were eliminated by horizon-near solar positions.

(1) P8, L16: How about introducing the metrics defined in section 2.1 here? In my opinion, the procedure is much easier to understand after the "properties" are explained. In section 2.1 it would be sufficient to explain that a multivariate analysis is performed based on image features/properties. The TSI images are then classified by comparing these features to reference values in a look-up table.

(2) Thank you for this suggestion. We have considered this before. However, since there are two separate sections making use of the same approach, only differing in details, we decided at the time to present this in the form you read. The proposed re-arrangement may be considered again if the journal decides to move forward with publication.

The master table does not really fit the term "look-up" table. The means and covariances are read by the program exactly once, at the beginning. The term "look-up" table implies repeated referencing to a database, and that is not what is occurring here.

(1) **P8, L17: "continually refined master table"** ☐ **Please explain this procedure.**

(2) This is done later, in section 3, and does not need to be said here. Eliminated the wording.

(3) **Pg10, ln6 The  master table defines a mean value vector *M*,**

(1) P8, L18/19: As suggested above it would be more convenient for the reader to define Eq.3 here.

(2) For now, I will leave the structure as is, and reconsider if asked for a final version of the manuscript. Again, Equation (3) (now 2-6) is used in multiple independent formulations of the same algorithm. It really only needs to be included once, and then referenced.
* * *
(1) P8, L19: Please provide the range of values expected for $F_{image}$ (in case of "22 · halo" and "no 22 · halo"). This might already be interesting to note on P5, L7.

(2) This comment refers to a position in the section on sky type scores, not halo scores. No halo decisions are made here. The values for F are arbitrary, due to the choice of C. The relative values matter for decisions on sky type, as already explained in the text pg10 lns10-20.
* * *
(1) P8, L25: How was the threshold of $10_{-8}$ chosen? Is it simply outside of the range of F? What kind of images yield this result? · explained later on P12, L1-3. Should be already mentioned here.

(2) Inserted additional sentence.

(3) **Pg10, line 18.** Such conditions may include overexposed images, horizon-near solar positions, a bird sitting on the mirror, and other conditions that produce images very different from the sky types sought after.
* * *
(1) P8, L29: "taken for the combined sky" · "for all 4 LSM quadrants"?

(2) Corrected.

(3) Pg10 line23  taken for all 4 LSM quadrants combined.
* * *
(1) P9, L9: Please explain the challenges that can be addressed by the "radial scattering analysis" and how

(2) Inserted sentence

(3) Page 11, line 1-3 The variation in radial intensity gradient as scatterers are present along the optical path can provide an alternative assessment for the presence of cirroform clouds, solving problems of classifying near-solar pixels using a colour ratio and/or intensity value only (Kennedy et al., 2016; N. Long et al., 2006).
* * *
**2.2.3 Ice halo score (IHS)**

(1) · P9, L9: The 22 · halo is formed by ice crystals in high-level cirrus clouds. So it is visible wherever cirrus clouds are present and not obstructed by low-level water clouds. The sentence as it stands now gives the wrong impression that the 22 · halo is overlaid over low-level clouds. Please correct the sentence accordingly.

(2) Corrected. Thanks.

(3) Pg11, line6: The 22° halo is a signal in the image that can be obscured by many other image features, including low clouds, partial clearings, inhomogeneous cirrostratus, regions of over-exposure, and near-horizon distortions.
* * *
(1) P9, L10/11: The fact that 22· halos are present in images classified as CLD and CLR provides more information about the definition of these categories and the selection of criteria rather than about the formation of the 22· halo. Please adjust the formulation of the sentence to avoid misunderstanding.

(2) This sentence is supposed to lay out the challenge of isolating the halo in a variety of sky conditions. Eliminated reference to algorithm sky types.

(3) Pg 11, line 7. The appearances of ice halos span a wide variety of sky conditions, ranging from almost clear skies to overcast altostratus skies, with the majority of halo phenomena appearing in cirrostratus skies.
* * *
(1) P9, L16: "variations in calibration" The image calibration should not vary across the images. The authors probably want to refer to the north-south mis-alignment of the camera and the coarse angular resolution which can pose a problem in identifying exact position of the 22· halo peak.

(2) True, no variation in the short term, however, as described in the rewritten section calibrations, sometimes recalibrations are necessary. The rewrite of section 2.1 addressed this.
* * *
(1) P9, L31: According to theory, the 22· halo peak should not be at the "same" location for the red and blue colour channel, but shifted. Is this feature used for the detection of 22· halos?

(2) No, this feature was not usable in these images. Resolution does not allow to distinguish the peak locations in a statistically reliable manner. In table 4, the peak locations are given together with their standard deviations. No significant difference can be gleaned from the color channels. A sentence was added.

(3) Pg12, line 1: The separation of colours observed in an ice halo display is not resolved with statistical significance in the TSI images, therefore this was not used as a criterion for halo detection.
* * *
(1) P10, L11: Please define "w" here, instead of L15.

(2) Inserted:

(3) Pg12, line 10: We added a Gaussian broadening to the time series of halo scores $F_i$, taken at times $t_i$ with a broadening $w$
* * *
(1) Figure 5: Please provide values for the IHS at the y-axis of the lower panel
(2) Figure was revised accordingly.
* * *
(1) Is the IHS calculated for each quadrant separately? (How) are they combined to classify the image?  ·  info on P10, L22 should be stated here as well as in section 2.3.2.
(2) This sentence was already contained in the text. It moved to the reference to figure 5.
(3) Pg12, line14: The raw halo score $F$ is computed for the four quadrants of an individual image, their sum is used to assign the raw score for the whole image.
* * *
**3 Results for January through April 2018**

(1)  ·  P10, L28: The values for C= $10_6$ and w=3.5 (w=4 was defined in L15!) should be mentioned earlier, where the respective equations were defined. Eq. 2 should be Eq. 5?

(2) Both, C0 and w are arbitrarily chosen, and are passed as a parameter as befits the question. The reference to w=4 images is specific for the day data in figure 5. For the evaluation in section 3, w=3.5 minutes. This limits the time resolution for halo appearances to 3.5 minutes, but smooths out false halo singals encountered in the record for that month. The equation references have been corrected in the renumbering of equations.
* * *
(1) P10, L19: It is not be surprising that "high halo scores coincide with strong CS signals", however it can be considered a confirmation that the image features used to train the algorithm were reasonably selected.

(2) Yes.
* * *
(1) P10, L31 through P11, L2: The determination of a "cut-off" or threshold value "to assign an image with a label of halo/no halo" results from training the algorithm. The same way as the threshold of 50% for the sky type. In both cases the threshold is "arbitrary" to some extent, but should be chosen to minimize either false positive or false negative classifications. This is correctly stated later on P11, L19, but should be mentioned earlier.

(2) Clarifying phrase inserted. Also corrected the limit value to the one finally used in the computations.

(3) Page 13, line 14: Our testing, minimizing false negatives and maximizing correct positives, places it at around 4000 for the month of March.
* * *
(1) Table 5: The difference between "%vis" and "%alg" is not quite clear. It seems that "%vis" provides an assessment of the visual image classification? This might be confusing for the reader. The interesting quantities here are the fraction of correctly and incorrectly classified images by the algorithm, compared to the ground truth (visual classification). Note that IHS > 4000 in the caption, but IHS > 3500 in the text!

(2) The caption contains language explaining "%vis" and "%alg". It is still confusing, even to me. Adding an example to the caption. Number was corrected in text.

(3) Caption for table 5: Table 2. STS and IHS test results for SGP March 2018. Visual assignments were made iteratively in step with the algorithm results as described in section 3. Given are the percentages of images of visual type that have been assigned an algorithm type (%vis), and the percentages of the algorithm type that correspond to a visual type (%alg). For example, 88% of all visual CS skies are also classified as CS by the algorithm, but only 86% of all algorithm CS skies also identify as CS if inspected visually. Agreement combinations in bold. IHS > 4000 to count an algorithm halo.
* * *
(1) P11, L4: "A small percentage of visual CLD skies trigger a PCL signal, mostly due to inhomogeneities in cloud cover." Please provide a number for the percentage. Does CLD mean completely overcast? Or do the inhomogeneities here correspond to small clearsky patches?

(2) CLD means that the radial analysis area exhibits properties close to an overcast sky (positive or zero gradient, low values of intercept, color ratio near 1, etc). Since the sky types are assigned only in the radial analysis area of each quadrant, this is not a statement about the whole sky. The percentage is given in table 5, which it discussed in this spot in the text. The inhomogeneities refer to differences in grayness across the analysis area, triggering a high dispersion in intensity values.

(3) Pg13, line 15: In Table 2, visual and algorithm results of the sky type assignments are cross-listed. It is worth reminding the reader that sky types are assigned only for the radial analysis interval indicated in Figure 1. Cloudy skies are reliably identified by the algorithm. A small percentage (3%) of visual CLD skies trigger a PCL signal, mostly due to inhomogeneities in cloud cover.
* * *
(1)  · P11, L4/5: Please provide a number stating how successful the classification of CLR is.

(2) As given in table 5. Inserted number also in text.

(3) Pg 13, line 18: The algorithm classifies 95% of all visual CLR skies correctly.
* * *
(1) P11, L27: If some CLR images were labeled as "22 · halo" why is the fraction of halo instances of CLR all sky type 0% in Tab. 5?

(2) This comment is confusing. Neither table 5 nor the location in the text refer to the stated issues. I will proceed under the following assumptions (1) the reviewer means table 6, (2) the reviewer refers to a now revised second sentence of sections 2.3.3 (?).
Table 6 lists that no halos where assigned in CLR skies, and that no CLR skies registered a halo signal. The second sentence in 2.3.3 was changed as listed above, to say that skies that appear clear can show a halo.
* * *
(1) P11, L23-31: The discussion of the challenges of visual classification of TSI images is very interesting, especially for other publications relying on this. As correctly mentioned, additional Lidar observations together with a temperature threshold e.g. from radiosonde data are useful to improve the classification (cf. Sassen et al. 2003 and Forster et al. 2017). Please add the respective citation also on P13, L7-9 and P13, L30.
(2) References added.
(3) Page 14, line 8ff: It is therefore a future necessity to combine the visual assignments of sky types with LIDAR data for altitude, optical thickness, and depolarization measurements to make an accurate assessment of the efficacy of the halo detection, following closely the processes described by Sassen et al. (Sassen et al., 2003) and Forster et al. (Forster et al., 2017).
* * *
(1) P12, L7: Please explain "various dimensions of the record".
On P10, L22 it was stated that "An image IHS and STS are assigned as the average over all scoring quadrants." How were the results for the individual quadrants obtained in Tab. 6?

(2) Changed sentence to:

(3) Pg 14, line 20: The ice halo statistics in **Error! Reference source not found.** lists data on ice halo statistics, including duration, number of incidents, and data on partial halos. The partial halo data are based on the individual quadrant IHS for an image, while the image score is used for duration and incidence information.
* * *
(1) It should be noted that due to the shadow band a "full 22 · halo" actually misses its top and bottom.
(2) I changed the designator in the table to 4/4 halo, to make clearer the connection to the number of halo-scoring quadrants.

(3) Table 6: 4⁄4 22° halo instead of "full 22° halo"
* * *
**4 Summary**

(1) · P13, L24: 86% vs 85% on P11, L18!
(2) Corrected the number
(3) Pg 16, line 13 The algorithm has been found to be about 90% in agreement with the visually assigned sky type, and 85% in agreement with the visually identified ice halo score.
* * *
(1) P13, L27/28: "The algorithm now will be applied to deliver ice halo data for the longterm TSI records accumulated in various geographical locations of ARM sites" Please replace by "In the future, the algorithm will be applied…" to avoid the misunderstanding that this was performed in the present study.
(2) Done.
(3) Page 16, line 16. In the future, the algorithm will be applied to deliver ice halo data for the long-term TSI records accumulated in various geographical locations of ARM sites, and allows further investigation into correlations with ...
* * *
**Please consider the following remarks to further improve the quality of the manuscript:**

The use of technical terms in the manuscript should be revised. In several instances a more commonly used expression exists, which should be used instead where applicable. For example:

(1) "Ice halo", I would suggest using the term "halo display", which is most commonly used in the literature. Please replace "ice halo" by "22· halo" wherever this specific type of halo display is referred to, e.g. P1, L18 and P12, L10.
IHS could be changed to HS22 or simply HS, when it is clear that it is only applied to the 22· halo.

(2) I changed multiple instances where clarity was improved. The IHS designator arises from the software implementation of the algorithm, and will remain.
* * *
(1) "look-up table" might be a more commonly use term than "external expandable master table". It is not clear what "expandable" and "external" means in this context? Most tables are expandable.
(2) The term "look-up" table implies continued reference to a data base or external file while the algorithm is working. That is not the case. The terms expandable and external " already fell in the revisions. I think the term "master table" is closest to a reference file that produces the means and covariance matrices, and is read exactly once during algorithm execution.
* * *
…

(2) Many of the suggested improvements in this series were incorporated in the revised manuscript.

**Typos and suggestions for improvement:**

(2) All implemented as proposed, with the exception of "P8,L2 Consider starting with the explanation of the properties of I(s). It will make the rest of the section much easier to follow." Will reconsider if final version is requested.

---

## Referee Report (RR1)

The authors have significantly improved the clarity and readability of the manuscript by incorporating the comments from both previous reviews. As the authors emphasized in their reply to the last comment, this manuscript is supposed to focus on the method and a detailed analysis of the TSI long-term dataset is envisaged for future publications. To give due consideration to this purpose of the manuscript, it is necessary to further relate the presented method to the existing literature. For example:

- Gnanadesikan, R. (1977) Methods for Statistical Data Analysis of Multivariate Observations, Wiley. ISBN 0-471-30845-5 (p. 83–86)
- Alpaydin, Ethem (2010). Introduction to Machine Learning. MIT Press. ISBN 978-0-262-01243-0 (chapter 5, 10)

Please ensure that already established techniques are described using the proper technical terms where applicable. I would argue that the presented method to detect 22deg halos on TSI images is indeed a binary classification (as opposed to the statement on P16, L7) but based on statistical (multivariate) analysis rather than machine learning. However, there is considerable overlap between both fields (statistical analysis vs. machine learning), especially considering training and testing of the algorithm as well as the related technical terms.

From my point of view, the following 3 major points have to be addressed before the manuscript can be published:

(The following comments refer to the revised manuscript and the authors' comments (AC) on the previous review, which are highlighted in blue.)

- 1. **Training and testing**: state-of-the-art techniques exist for "training" and "testing" a classification algorithm (see e.g. Alpaydin 2010). The most important requirement is using a new dataset for testing the algorithm which was not used for training. The algorithm presented here seems to use the same data from March 2018 for both training and testing (cf. P13, L1-3). Please revise the manuscript and, if necessary, the presented method accordingly.
- 2. Linear classification: the method of assigning a "sky type" or "ice halo score" presented in this study seems to be very similar to *Fisher's linear discriminant* (Gnanadesikan, p. 83-86) or *Linear Discriminant Analysis* (LDA) (Alpaydin, chapters 5 and 10). Both use feature vectors weighted by the Mahalanobis distance and a threshold to assign new data to one of the classes (linear classification). Please discuss and add citations where appropriate.

Moreover, please revise the manuscript using the correct technical terms which can be found in the literature (e.g. Alpaydin 2010): e.g. "(expandable) master table" probably refers to "training dataset", which contains "feature vectors".

3. Sky type classification: the TSI images in this study were separated into the categories "Cirrostratus" and three levels of cloud fraction "Cloudy" (CLD), "Partly Cloudy" (PCL), and "Clear" (CLR), which were defined by their visual appearance. This method is different compared to previous studies, which used Lidar observations and a temperature threshold to identify ice clouds (Sassen et al. 2003 and Forster et al. 2017). Using a different method, makes it very difficult to compare the results (P15, L18-21). As stated in the previous review, I see the potential of this study especially in comparing the results to previous studies (and different locations). Therefore, the same criteria should be used to define the basic population of "cirrus clouds".

Furthermore, the choice of sky types does not seem to be very suitable for the described goal of this study: "With the goal of using these long-term image records to provide supporting information [to] the presence of smooth, hexagonal ice crystals in cirrus clouds from observations of 22deg halos, we developed an algorithm that assigns sky type and halo scores to long-term series of TSI images" (P16, L15-17).

Although the majority of 22deg halos coincides with "CS", a significant amount (44% for Jan and 38% for Feb) coincides with "Partly cloudy" and "Cloudy" skies (cf. Tab. 6 "% sky type of all halo instances"). So the sky type categories used here are apparently not a good indicator of whether the present clouds are able to produce a 22deg halo and are therefore not suitable for drawing conclusions about ice crystal microphysics in halo-bearing clouds in general.

It is mentioned several times throughout the manuscript that the sky type classification of the images is used to infer information about the "presence of smooth crystalline habits among the cloud particles" (e.g. P15, L28-30). To answer this question, it would be necessary to identify ice clouds and separate them from other sky types including clear sky, as it was done in Sassen et al. 2003 and Forster et al. 2017.

Nevertheless, it is possible of course to draw conclusions from the frequency of 22deg halos in "CS" skies, but it has to be stated explicitly. In the citation above (P16, L15ff), the words "cirrus clouds" would have to be replaced by "CS", for example. Cirrostratus is only a subcategory of Cirrus, as e.g. Cirrocumulus.

Please address these concerns and revise the manuscript by accurately describing which sky type the results actually refer to when interpreting the results, drawing conclusions, and comparing them with other studies. Please explain in the manuscript the reasons for choosing these specific sky type categories and their merit for the goal of the study.

In the following, please find specific remarks to each of the four points summarized above:

**Specific remarks**

**1. Training and testing:**

a. P13, L1-3: "The sections of the record in which visual and algorithm differed were inspected again, at which point either the visual assessment was adjusted, or the misclassified images were included in the Master table in order to train the algorithm toward better recognition."

Apparently, the March dataset is used for tuning the algorithm, i.e. for finding the classification threshold. This is not equivalent to testing. For the latter, the trained and tuned algorithm is tested against completely new data and should not be

updated simultaneously. In order to avoid a bias in the assessment of the final classification quality of the algorithm, the final trained version should be applied to a dataset which was not used for training (cf. literature for state-of-the-art techniques). This implies that the training data set actually contains 80 + 44,026 images. Please address this point.

- b. P16, L11: "Further training is easy to incorporate via a master table which provides means and covariance matrices to the algorithm."
  "Training" the algorithm presented in this study means finding a threshold that best separates the two classes. Adding new feature vectors to the "master table" is usually referred to as generating training data.
- c. P13, L17: "Upon inspection of the numerical values for IHS, it becomes clear that `a cut-off is needed to assign an image with a label of halo/no halo. This cut-off value is arbitrary and dependent on factors such as *w* and *C*0, as well as the quality of the calibration. **Our testing places it at around 4000 for the month of March**." Which values were used for the other months? To my understanding, training the algorithm should result in *one* threshold value which will be applied to the whole TSI dataset. The sentence highlighted above gives the impression that a separate threshold is determined for each month. In that case, it will require a lot of work tuning the algorithm for each month separately for this large dataset. Please clarify.
- d. AC: Both, CO and w are arbitrarily chosen, and are passed as a parameter as befits the question. The reference to w=4 images is specific for the day data in figure 5. For the evaluation in section 3, w=3.5 minutes. This limits the time resolution for halo appearances to 3.5 minutes, but smooths out false halo singals encountered in the record for that month. The equation references have been corrected in the renumbering of equations.

This should be explained in the manuscript since it affects the results for the mean duration of 22deg halos in Tab. 6. and the histogram in Fig. 8.

If a different value for w is used for each day, the first bin (0-5 min) in Fig. 8 will be mainly subjected to the this choice (should actually be 4-5 min?). Why is the choice of w changed? It should be constant throughout the analysis.

e. P15, L2: "Due to the time-broadening applied via Eqn (16), the display time cannot be resolved below 3 minutes."
P12, L20: "The broadening w in Eqn (16) was chosen as 4 images for this example, which means the Gaussian half width corresponds to 2 minutes."
See previous comment. Please double-check, is it 2, 3, 3.5, or 4 minutes?

**2. Linear classification:**

a. P10, L22 "An image IHS and STS are assigned as the average over all scoring quadrants." How were the results calculated for each individual quadrant in Tab. 6?
 By a linear combination as for the Linear Discriminant Analysis?

b. P16, L7: "The algorithm presented here for TSI data [...] does not characterize halos in a binary decision, but rather assigns a continuous ice halo score to an image [...]" The presented algorithm does classify halos in a binary decision after computing the score. This is true for other classification algorithms as well, e.g. for the random forest classifier. Please correct this statement.

**3. Sky type classification:**

a. P15, L18-21: "For example, in January we found that 9 % of all cirrostratus skies were accompanied by a 22deg halo. In the data for April, this fraction increased to 22% of all cirrostratus skies. We also have registered halos for a portion of partly cloudy skies, and for cloudy skies. No halos have been registered in any of the clear skies. This is certainly consistent with the observations of Forster et al (Forster et al., 2017)."

The reference of the last sentence "this is certainly consistent with the observations of Forster et al" is not quite clear. Does it refer to "No halos have been registered in any of the clear skies" or 22% of all cirrostratus skies show halos? If the comparison refers to the fraction of cirrostratus skies accompanied by a 22deg halo, this statement is not correct. Sassen et al. 2003 and Forster et al. 2017 use Lidar data (and a temperature threshold) to identify clouds dominated by ice crystals. So even though the resulting numbers are similar for April, the population is different.

b. A quantity that could be directly compared is the overall frequency of all 22deg halos with >=1/4. Could you provide a number?

**Minor comments:**

• P15, L28-30: "One of the conclusion to be made from the relation between STS and IHS concerns the confidence in the presence of smooth crystalline habits among the cloud particles, as shown only in a one-fifth fraction of all cirrostratus."

Please clarify this sentence: what is the conclusion here? The average fraction of 22deg in CS sky types amounts to 16.25%, i.e. rather 1/6 than 1/5.

• AC: "[...]I have not been able to visually and reliably discriminate parhelia in any TSI image. An algorithm specifically for parhelia was therefore not attempted. With the separation into quadrants, any existing parhelia would form right on the boundary between top and bottom quadrant, and basically average into the radial intensity of this quadrant. [...] The top quadrants, if not overexposed, may give halo signals. But again – parhelia can not be visually distinguished in those images.

It would be worth adding a short sentence to the manuscript, describing that 22deg parhelia could, in principle be mis-classified as 22deg halo, but due to the coarse image resolution and low brightness, they have not been detected so far in the TSI images.

- P7, L17 ff: Was the image distortion accounted for in addition to the coordinate transformation in Eq. 9? If it was assumed negligible, this should be stated in the text and expected errors could be cited from Long et al. (cf. Fig. 5)
- Long et al. appears twice in the references.
- P7, L17: plain -> plane
- P12, L27: 44,026 images vs. Tab. 6 44,057 images
- Table 5: It seems that the attempt was made here to combine two tables into one. I would suggest limiting the table to the assessment of the algorithm compared to the visual inspection, which has to be assumed as "ground truth" here. That means the table should express only the second part of the sentence: "86 % of all algorithm CS skies also identify as CS if inspected visually". The first part of the sentence would focus on assessing the visual classification of the images against the algorithm, which is not the primary interest here. If the authors consider both results equally important, I would suggest separating the results into 2 tables.

|           | Visual |     |     |     |
|-----------|--------|-----|-----|-----|
| Algorithm | CS     | PCL | CLD | CLR |
| CS        | 86%    | 3%  | 1%  | 6%  |
| PCL       |        | 91% |     |     |
| CLD       |        |     | 98% |     |
| CLR       |        |     |     | 93% |

For example:

---

## Author Response (AR2)

**Author Response to request for major revisions**

We thank the reviewer and editor for their thoughtful input into improvements for this manuscript. In particular, going back to textbooks in multivariate analysis was helpful in improving vocabulary use in the manuscript.

Below, we provide details and justifications for changes made to the manuscript in response to the concerns. The line numbers in the reviewer's text refer to the version of the manuscript submitted to AMT on April 11 2019. Reviewer's comments are in blue italics, author's responses in black.

This resubmission is accompanied by a revised pdf version of the manuscript, as well as a version that tracks the changes made.
* * *
*Reviewer:*

*To give due consideration to this purpose of the manuscript, it is necessary to further relate the presented method to the existing literature. For example:*

*• Gnanadesikan, R. (1977) Methods for Statistical Data Analysis of Multivariate*

*Observations, Wiley. ISBN 0-471-30845-5 (p. 83–86)*

*• Alpaydin, Ethem (2010). Introduction to Machine Learning. MIT Press. ISBN 978-0-262-*

*01243-0 (chapter 5, 10)*

*Please ensure that already established techniques are described using the proper technical terms where applicable. I would argue that the presented method to detect 22deg halos on TSI images is indeed a binary classification (as opposed to the statement on P16, L7) but based on statistical (multivariate) analysis rather than machine learning. However, there is considerable overlap between both fields (statistical analysis vs. machine learning), especially considering training and testing of the algorithm as well as the related technical terms.*

*From my point of view, the following 3 major points have to be addressed before the manuscript*

*can be published:*

*(The following comments refer to the revised manuscript and the authors' comments (AC) on the previous review, which are highlighted in blue.)*

1. ***Training and testing****: state-of-the-art techniques exist for "training" and "testing" a classification algorithm (see e.g. Alpaydin 2010). The most important requirement is using a new dataset for testing the algorithm which was not used for training. The algorithm presented here seems to use the same data from March 2018 for both training and testing (cf. P13, L1-3). Please revise the manuscript and, if necessary, the presented method accordingly.*

AC: This impression of the reviewer is not reflective of the procedure used. Only a diminutive sample of observation vectors (order of $10^2$) are taken from the $10^5$ available observations vectors in March, less than 0.1%, which leaves >99.9% of all images in March as a testing set. The text has been amended as described below in the more detailed comments to better present this.

*2. **Linear classification**: the method of assigning a "sky type" or "ice halo score" presented in this study seems to be very similar to Fisher's linear discriminant (Gnanadesikan, p. 83-86) or Linear Discriminant Analysis (LDA) (Alpaydin, chapters 5 and 10). Both use feature vectors weighted by the Mahalanobis distance and a threshold to assign new data to one of the classes (linear classification). Please discuss and add citations where appropriate.*

*Moreover, please revise the manuscript using the correct technical terms which can be found in the literature (e.g. Alpaydin 2010): e.g. "(expandable) master table" probably refers to "training dataset", which contains "feature vectors".*

AC:

There are quite a few textbooks on multivariate normal analysis, reaching back to the 1970s. The method was one of the first computer-based image analysis methods, used in facial recognition and expanded to a multitude of image classification problems, leading to a slew of publications on the method in particular contexts. A few references to theoretical and applied text books were inserted. Most applicable for this work is the reference to Flury and Riedwyl. As to nomenclature, the wording proposed by the reviewer is used in some of the references, not in others. We inserted the word "training set" where appropriate, and clarified the references to training sets as being specific for a class of images, all collected in a master table. We also use the word "class" for each targeted image property for which an image is analyzed, in addition to "observation vector" for a single property set.

The described method does not represent a linear discriminant analysis. LDA would add another layer to include the definition of hyperplanes that divide the property space into cells for each class of images, and deliver yes/no answers for each image class investigated (4 photographic sky types, plus ice halo presence). The centroids of the classes have too much overlap for this to be a good approach. It is not a method we pursued. Rather, the described method assigns a continuous numeric score for each class, basically a probability density. For the PSTS, ten image properties are used. Comparing the four PSTS classes, the dominant skytype is assigned by the one with the highest probability – but that does not mean the other scores are valueless. Any post-processing decisions that set threshold values are not part of the algorithm itself.

Much of the method description has been revised and reworded as appropriate.

*3. **Sky type classification**: the TSI images in this study were separated into the categories "Cirrostratus" and three levels of cloud fraction "Cloudy" (CLD), "Partly Cloudy" (PCL), and "Clear" (CLR), which were defined by their visual appearance. This method is different compared to previous studies, which used Lidar observations and a temperature threshold to identify ice clouds (Sassen et al. 2003 and Forster et al. 2017). Using a different method, makes it very difficult to compare the results (P15, L18-21). As stated in the previous review, I see the potential of this study especially in comparing the results to previous*

*studies (and different locations). Therefore, the same criteria should be used to define the basic population of "cirrus clouds".*

*Furthermore, the choice of sky types does not seem to be very suitable for the described goal of this study: "With the goal of using these long-term image records to provide supporting information [to] the presence of smooth, hexagonal ice crystals in cirrus clouds from observations of 22deg halos , we developed an algorithm that assigns sky type and halo scores to long-term series of TSI images" (P16, L15-17).*

*Although the majority of 22deg halos coincides with "CS", a significant amount (44% for Jan and 38% for Feb) coincides with "Partly cloudy" and "Cloudy" skies (cf. Tab. 6 "% sky type of all halo instances"). So the sky type categories used here are apparently not a good indicator of whether the present clouds are able to produce a 22deg halo and are therefore not suitable for drawing conclusions about ice crystal microphysics in halo-bearing clouds in general.*

*It is mentioned several times throughout the manuscript that the sky type classification of the images is used to infer information about the "presence of smooth crystalline habits among the cloud particles" (e.g. P15, L28-30). To answer this question, it would be necessary to identify ice clouds and separate them from other sky types including clear sky, as it was done in Sassen et al. 2003 and Forster et al. 2017.*

*Nevertheless, it is possible of course to draw conclusions from the frequency of 22deg halos in "CS" skies, but it has to be stated explicitly. In the citation above (P16, L15ff), the words "cirrus clouds" would have to be replaced by "CS", for example. Cirrostratus is only a subcategory of Cirrus, as e.g. Cirrocumulus.*

*Please address these concerns and revise the manuscript by accurately describing which sky type the results actually refer to when interpreting the results, drawing conclusions, and comparing them with other studies. Please explain in the manuscript the reasons for choosing these specific sky type categories and their merit for the goal of the study.*

AC response:

The authors understand the concern about the sky type choices, in particular the concern that this manuscript does not include a coordination with LIDAR, IR, or other instrumental records that might support the assignments in particular of cirrostratus. The manuscript includes multiple statements that this algorithm must be complemented by other instrumental records to reach the stated goal of contributing to cirrus microphysics understanding, but perhaps it is prudent to be more specific in nomenclature. We describe an image analysis algorithm for TSI data, which uses the color-resolved radial brightness gradient and its accessories to assign a sky type based on data solely from the analysed area in the photographic record. The type of information in the brightness gradient is extensively described in section 2.3. It allows to distinguish four types of sky conditions clearly, namely types that closely resemble visually any of these: CS, CLD, PCL, and CLR. The reviewer makes the correct point that this is not sufficient for a conclusive call about the cloud types present at the image time, since neither altitude nor temperature information is included. In a paper that describes a technique to analyse photographic images, it is then prudent to specifically name these sky type assignments "photographic sky types" (PST). It seems to be a comfortable choice, since it is factually correct, based on the available

record, self-contained in the stated algorithm, and will be helpful to consider at the time when other instrumental records may conflict with the PST.

*In the following, please find specific remarks to each of the four points summarized above:*

***Specific remarks***

*1. **Training and testing**:*

*a. P13, L1-3: "The sections of the record in which visual and algorithm differed were inspected again, at which point either the visual assessment was adjusted, or the misclassified images were included in the Master table in order to train the algorithm toward better recognition." Apparently, the March dataset is used for tuning the algorithm, i.e. for finding the classification threshold. This is not equivalent to testing. For the latter, the trained and tuned algorithm is tested against completely new data and should not be updated simultaneously. In order to avoid a bias in the assessment of the final classification quality of the algorithm, the final trained version should be applied to a dataset which was not used for training (cf. literature for state-of-the-art techniques). This implies that the training data set actually contains 80 + 44,026 images. Please address this point.*

AC:

The text in Section 3 that describes the iterative training process has been revised to clarify the points misunderstood here. Out of the 44,057 images in March, only 31,398 are classifiable in terms of s PST and IHS. Out of these, each image can contribute up to four training sets (4 if all quadrant qualify, 2 if the sun is near to the horizon). This means, the month provides about 100,000 potential property sets to be used in training the algorithm. As you inspect tables 3 and 4, you will notice that the actual final training set for this manuscript contains between 93 and 188 training records for each scored class. So, to assume that the 100,000 quadrants of the images in March are all part of the training set is incorrect. In fact, the samples taken to train the algorithm are diminutive compared to the number of sets on which the algorithm was tested. Perhaps, the new wording clarifies this.

*b. P16, L11: "Further training is easy to incorporate via a master table which provides means and covariance matrices to the algorithm." "Training" the algorithm presented in this study means finding a threshold that best separates the two classes. Adding new feature vectors to the "master table" is usually referred to as generating training data.*

AC: This passage has been amended already while working on other parts of this response.

*c. P13, L17: "Upon inspection of the numerical values for IHS, it becomes clear that `a cut-off is needed to assign an image with a label of halo/no halo. This cut-off value is arbitrary and dependent on factors such as w and C0, as well as the quality of the calibration. **Our testing places it at around 4000 for the month of March**." Which values were used for the other months? To my understanding, training the algorithm should result in one threshold value which will be applied to the whole TSI dataset. The sentence highlighted above gives the impression that a separate threshold is determined for each month.*

*In that case, it will require a lot of work tuning the algorithm for each month separately for this large dataset. Please clarify.*

AC: Understood. The threshold is not part of the algorithm. The algorithm assigns a continuous IHS to every quadrant, and the average to every image, as a number that can be below $10^{-10}$ or above $10^5$, with fluid continuous change in consecutive images. The decision on where to place a cutoff is based on the behavior of the timeline. Halo images place a significant peak above a "forest" of low-level peaks. The discriminator is placed to exclude about 75% of the low-level peaks when data gathering for a *count* of halo incidences. The text has been amended to clarify this. Doing the actual long-term analysis does require a lot of work in the calibration alone, and in the final use of the PSTS and IHS output. That is part of the reason to present only four months in this manuscript.

*d. AC: Both, C0 and w are arbitrarily chosen, and are passed as a parameter as befits the question. The reference to w=4 images is specific for the day data in figure 5. For the evaluation in section 3, w=3.5 minutes. This limits the time resolution for halo appearances to 3.5 minutes, but smooths out false halo singals encountered in the record for that month. The equation references have been corrected in the renumbering of equations. This should be explained in the manuscript since it affects the results for the mean duration of 22deg halos in Tab. 6. and the histogram in Fig. 8. If a different value for w is used for each day, the first bin (0-5 min) in Fig. 8 will be mainly subjected to the this choice (should actually be 4-5 min?). Why is the choice of w changed? It should be constant throughout the analysis.*

*e. P15, L2: "Due to the time-broadening applied via Eqn (16), the display time cannot be resolved below 3 minutes." P12, L20: "The broadening w in Eqn (16) was chosen as 4 images for this example,which means the Gaussian half width corresponds to 2 minutes." See previous comment. Please double-check, is it 2, 3, 3.5, or 4 minutes?*

AC: We apologize for the confusion. All presented data and figures have been corrected for a broadening of w=7 images (or 3.5 minutes), and the locations in the text have been modified accordingly. The comment referring to figure 8 is a little confusing. The broadening w does not vary by day. The lowest bin (0 to 5 min) appears to be correctly labeled, since the bin size in the histogram is 5 minutes. The fact that this bin is influenced by the broadening is discussed in the text.

*2. Linear classification:*

*a. P10, L22 "An image IHS and STS are assigned as the average over all scoring quadrants." How were the results calculated for each individual quadrant in Tab. 6? By a linear combination as for the Linear Discriminant Analysis?*

AC: All scores are computed for quadrants, thus each image first receives four individual quadrant scores. The image score is the average of the quadrant scores. The results in Table 6 simply use the quadrant scores themselves which are computed using the method described in detail in this manuscript. The word "scoring" in this quote refers to the fact that some quadrants may be excluded from the averaging since they may not have had a valid score in a class (low sun, over exposure, bird on mirror, too low value for F, etc). No changes were made to the manuscript in response to this question.

*b. P16, L7: "The algorithm presented here for TSI data […] does not characterize halos in a binary decision, but rather assigns a continuous ice halo score to an image [...]" The presented algorithm does*

*classify halos in a binary decision after computing the score. This is true for other classification algorithms as well, e.g. for the random forest classifier. Please correct this statement.*

AC: The statement is correct as it is written. Some of the changes made in response to earlier comments do elaborate on this.

*3. **Sky type classification**:*

*a. P15, L18-21: "For example, in January we found that 9 % of all cirrostratus skies were accompanied by a 22deg halo. In the data for April, this fraction increased to 22% of all cirrostratus skies. We also have registered halos for a portion of partly cloudy skies, and for cloudy skies. No halos have been registered in any of the clear skies. This is certainly consistent with the observations of Forster et al (Forster et al., 2017)." The reference of the last sentence "this is certainly consistent with the observations of Forster et al" is not quite clear. Does it refer to "No halos have been registered in any of the clear skies" or 22% of all cirrostratus skies show halos? If the comparison refers to the fraction of cirrostratus skies accompanied by a 22deg halo, this statement is not correct. Sassen et al. 2003 and Forster et al. 2017 use Lidar data (and a temperature threshold) to identify clouds dominated by ice crystals. So even though the resulting numbers are similar for April, the population is different.*

AC: the direct numbers to which this sentence referred were placed a couple of lines above. They have been moved to a better position, and the sentence has been modified accordingly (P15 L28ff). The reviewer correctly reiterates that we are comparing a photographic record to a Lidar-verified record. Language to that extend has been inserted into the manuscript.

*b. A quantity that could be directly compared is the overall frequency of all 22deg halos with >=1/4. Could you provide a number?*

AC: The overall frequency of halos is given in table 6, for all sky types. The overall halo frequency varies between 3.9% of all images in January 2018 to 9.4% of all images in April 2018. Sassen et al. gives a number of 6% of time with a clear and bright 22° halo for the 10-year FARS record, but also indicates a fraction of 37.3% of time with any partial or weak indication of 22° halo. That is a very wide range to compare to. In addition, Forster noted that this particular statistics is sensitive to the binning interval. If 1-h intervals are used, then the halo fraction may increase to 50%. A few sentences describing this were inserted into the manuscript P16 L5ff.

***Minor comments:***

*• P15, L28-30: "One of the conclusion to be made from the relation between STS and IHS concerns the confidence in the presence of smooth crystalline habits among the cloud particles, as shown only in a one-fifth fraction of all cirrostratus." Please clarify this sentence: what is the conclusion here? The average fraction of 22deg in CS sky types amounts to 16.25%, i.e. rather 1/6 than 1/5.*

AC: This sentence seems to be from an earlier version of the manuscript. The described location contains a significantly more detailed description of the observed halos in CS skies, as given by reviewer comment 3a above.

*• AC: "[…]I have not been able to visually and reliably discriminate parhelia in any TSI image. An algorithm specifically for parhelia was therefore not attempted. With the separation into quadrants, any existing parhelia would form right on the boundary between top and bottom quadrant, and basically*

*average into the radial intensity of this quadrant. […] The top quadrants, if not overexposed, may give halo signals. But again – parhelia can not be visually distinguished in those images. It would be worth adding a short sentence to the manuscript, describing that 22deg parhelia could, in principle be mis-classified as 22deg halo, but due to the coarse image resolution and low brightness, they have not been detected so far in the TSI images.*

AC: Inserted a comment to this effect on P3,L29

*• P7, L17 ff: Was the image distortion accounted for in addition to the coordinate transformation in Eq. 9? If it was assumed negligible, this should be stated in the text and expected errors could be cited from Long et al. (cf. Fig. 5)*

AC: This correction was found to be small, and not to significantly disturb the finding of the solar position. It has been omitted. Text of the manuscript was adjusted accordingly (P7 L30)

*• Long et al. appears twice in the references.*

AC: corrected.

*• P7, L17: plain -> plane*

AC: Typo was corrected. Thanks.

*• P12, L27: 44,026 images vs. Tab. 6 44,057 images*

AC: The record of month March 2018 contains 44057 images. P12,L27 was corrected.

*• Table 5: It seems that the attempt was made here to combine two tables into one. I would suggest limiting the table to the assessment of the algorithm compared to the visual inspection, which has to be assumed as "ground truth" here. That means the table should express only the second part of the sentence: "86 % of all algorithm CS skies also identify as CS if inspected visually". The first part of the sentence would focus on assessing the visual classification of the images against the algorithm, which is not the primary interest here. If the authors consider both results equally important, I would suggest separating the results into 2 tables.*

For example:

| Algorithm | Visual CS | PCL | CLD | CLR |
|-----------|-----------|-----|-----|-----|
| CS | 86% | 3% | 1% | 6% |
| PCL | ... | 91% | ... | |
| CLD | | | 98% | |
| CLR | | | | 93% |

AC: The table has been split into two segments, according to reviewer suggestion

[revised manuscript text omitted]

---

## Author Response (AR3)

Dear Associate Editor,

Thank you very much for accepting the manuscript for publication.

We have made the requested correction to the reference in the "Code Availability" section, as well as to the reference Boyd et al itself. The change items are highlighted in red below.

The text, tables, and figure captions are contained in the upload manuscript07112019-text.docx. A few minor corrections were made to the captions: bold face was removed from the text, and symbol explanations were included as a legend in the figures, instead of the captions.

Since the manuscript was written in MS Word, no separate reference file appears to be required. If that is in error, please let us know.

The figures were prepared as pdf files if applicable, with the exception of Figure 2. Please let us know if the submitted formats do not comply with the requirements for any reason.

Thank you again, and we are looking forward to the next steps.

Sylke Boyd